# Uncovering Vulnerabilities of LLM-Assisted Cyber Threat Intelligence

## Abstract

Large Language Models (LLMs) are intensively used to assist security analysts in counteracting the rapid exploitation of cyber threats, wherein LLMs offer cyber threat intelligence (CTI) to support vulnerability assessment and incident response. While recent work has shown that LLMs can support a wide range of CTI tasks such as threat analysis, vulnerability detection, and intrusion defense, significant performance gaps persist in practical deployments. In this paper, we investigate the intrinsic vulnerabilities of LLMs in CTI, focusing on challenges that arise from the nature of the threat landscape itself rather than the model architecture. Using large-scale evaluations across multiple CTI benchmarks and real-world threat reports, we introduce a novel categorization methodology that integrates stratification, autoregressive refinement, and human-in-the-loop supervision to reliably analyze failure instances. Through extensive experiments and human inspections, we reveal three fundamental vulnerabilities: spurious correlations, contradictory knowledge, and constrained generalization, that limit LLMs in effectively supporting CTI. Subsequently, we provide actionable insights for designing more robust LLM-powered CTI systems to facilitate future research.

## 1 Introduction

We are living in an era of rapid digital transformation, where technological advancements are tightly associated with the growing prevalence of cyber threats. In recent years, the cyber threat landscape has shifted dramatically, with reported Common Vulnerabilities and Exposures (CVEs) increasing by an average of 25% annually (Intelligence, 2023). In 2024 alone, more than 40,000 vulnerabilities were reported (Corporation, 2025). This surge can be attributed to the rising complexity of IT systems (Qualys, 2024), the widespread adoption of open-source software (Dam & Neumaier, 2023; Reading, 2024), and the accelerating pace of modern development cycles (Security, 2024). Together, these dynamics expand the attack surface while making sole reliance on human analysts for vulnerability assessment and remediation increasingly infeasible.

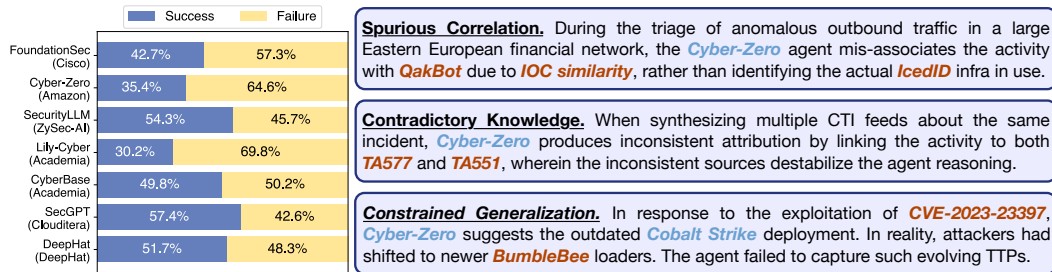

Figure 1: **(Left)** Failure ratios of cybersecurity agents. **(Right)** Examples of vulnerabilities.

Large Language Models (LLMs) have recently demonstrated strong performance in a broad range of cyber threat intelligence (CTI) tasks. By adapting models through instruction fine-tuning or prompt-based automation, researchers have applied LLMs to support threat analysis and decision-making (Zhang et al., 2023), code vulnerability detection (Du et al., 2024), and defense against network

**Cyber Threat Intelligence Stages**

| Vulnerability | Contextualization | Attribution | Prediction | Mitigation | General | Agent |
|---|---|---|---|---|---|---|
| **Spurious Correlation** General: 20.42% Agent: 11.64% | 1.1 Co-mention bias from raw threat incident | | | | 4.62% | 3.54% |
| | 1.2 Exploitation bias from deliberately reused IoCs | | | | 2.77% | 0.85% |
| | 1.3 Confounding factors that explicitly/implicitly correlate entities | | | | 13.60% | 17.67% |
| | 1.4 Skewed source | | | | 7.73% | 4.69% |
| | 1.5 Hierarchical metadata from attack chains | | | | 3.06% | 0.75% |
| **Contradictory Knowledge** General: 35.93% Agent: 26.35% | 2.1 Temporal contradiction between outdated and recent evidence | | | | 17.53% | 9.66% |
| | 2.2 Conflicting reports of attack contexts or dependencies | | | | 25.44% | 18.77% |
| | 2.3 Semantic conflict | | | | 5.72% | 1.95% |
| | 2.4 Divergent data structure from different platforms | | | | 4.51% | 4.47% |
| | | 2.5 Misaligned knowledge and security standards | | | 2.68% | 0.91% |
| | | 2.6 Countering CTI generation and LLM alignment | | | 12.51% | 2.36% |
| **Constrained Generalization** General: 33.72% Agent: 31.05% | 3.1 Distributional bias | | | | 22.71% | 24.05% |
| | | 3.2 Unseen pattern from emerging threats | | | 22.26% | 28.26% |
| | | 3.3 Overfitted reasoning | | | 2.92% | 18.14% |
| | | 3.4 Environmental unawareness | | | 7.75% | 14.29% |

Figure 2: Summarization of the vulnerability types of LLMs in various CTI stages. Ratios are calculated over the entire dataset (§2.2). Vulnerabilities may overlap within a single threat instance.

intrusions (Lavi et al., 2024). Despite these advances, substantial performance gaps remain in their reported evaluations (Deng et al., 2024; Clairoux-Trepanier et al., 2024; Ji et al., 2024; Alam et al., 2024; Liu et al., 2025), suggesting that such limitations cannot be fully addressed through typical model adaptation or prompt-based automation. This raises a fundamental research question: **What intrinsic vulnerabilities constrain the effectiveness of LLMs in supporting CTI tasks?**

Although some vulnerabilities stem from general challenges in LLM architecture and training methodologies (which affect not only CTI but also broader domains (Aguilera-Martínez & Berzal, 2025)), this work focuses on the **nature of the threat landscape itself, which may introduce unique vulnerabilities** for LLMs in effectively supporting CTI. Specifically, CTI requires reasoning under intertwined, crowdsourced, and imbalanced evidence, which are conditions that differ substantially from standard NLP benchmarks. To investigate this, this work makes the following contributions:

First, we conduct large-scale, systematic studies of LLM vulnerabilities in CTI, spanning a full-fledged CTI lifecycle from threat analysis to incident responses. Our evaluation datasets drawn from multiple CTI benchmarks (CTIBench (Alam et al., 2024), SevenLLM-Bench (Ji et al., 2024), SWE-Bench (Jimenez et al., 2023), and CyberTeam (Liu et al., 2025)) as well as real-world threat reports from CTI databases (CVE Program, 2024; National Institute of Standards and Technology (NIST), 2024; Cybersecurity and Infrastructure Security Agency (CISA), 2024) to ensure comprehensiveness.

Next, we develop a novel categorization methodology to investigate failure instances. One major challenge in studying extensive threat instances is reliably scaling the analysis to categorize diverse failure cases. Standard "LLM-as-judge" approaches are unreliable for automatic vulnerability type finding and classification, as models tend to rationalize their own reasoning or fail to critically evaluate contradictions (Yamauchi et al., 2025; Guerdan et al., 2025). To address this, we propose an autoregressive, human-in-the-loop framework that categorizes failure instances efficiently and with high reliability (§3). This methodology allows us to extract consistent insights from large-scale evaluations while minimizing annotation workload.

Lastly, through large-scale experiments on industry-leading LLMs and LLM-powered cybersecurity agents, complemented by detailed case studies, we identify three dominant categories of vulnerabilities that limit LLMs in handling the threat landscape, as illustrated in Figure 1:

**(1) Spurious correlations**, wherein LLMs over-attribute based on superficial or co-occurring features (e.g., mistaking commodity tools such as Mimikatz (Delpy, 2011) as actor-specific evidence).

**(2) Contradictory knowledge**, wherein inconsistencies in CTI sources confuse models, causing unstable or conflicting predictions.

**(3) Constrained generalization**, wherein LLMs struggle to extend beyond familiar distributions, failing on emerging (zero-day) attack surfaces or novel TTPs.

Our analysis further shows that these vulnerabilities directly undermine the effectiveness of techniques used in CTI pipelines. As summarized in Figure 2, each vulnerability spans multiple CTI stages and influences different aspects of LLM reasoning. For example, spurious correlations distort evidence retrieval, where LLMs amplify irrelevant co-occurrences in retrieved evidence, resulting in misguided contextualization and faulty attribution. Overall, the detailed studies reveal the root causes by combining vulnerabilities with the specific LLM workflows involved in different CTI stages, which provides an in-depth understanding of how these vulnerabilities propagate through the reasoning.

By uncovering blind spots in LLM vulnerabilities for CTI, we provide foundational insights that can guide principled adaptations in future CTI-focused LLM systems. To facilitate follow-up research, we release codes at `https://anonymous.4open.science/r/LLM-CTI-E773`.

## 2 BACKGROUND AND MOTIVATION

This section first outlines the scope of CTI tasks (§2.1), then introduces the datasets used to assess LLM performance along with extensive evaluation results (§2.2). These results motivate a deeper investigation into specific vulnerabilities and their root causes, which we address through our methodological design to categorize failure instances (§3) and research findings (§4).

### 2.1 BACKGROUND: CYBER THREAT INTELLIGENCE STAGES

Cyber threat intelligence (CTI) covers a broad range of cybersecurity activities that support the analysis of threat events and the recommendation of timely, informed incident response.

> **Motivating Example.** A healthcare network detects suspicious outbound traffic linked to newly registered domains associated with *QakBot*. **The security team first enriches event metadata** by mapping proxy logs to known *C2 infrastructure* and retrieving prior reports for context. **Next, they attribute** *QakBot*'s reuse of infrastructure to the threat group *TA577*. Based on historical correlations, **they then predict** a likely transition to *Cobalt Strike* and eventual ransomware deployment. Finally, **the team implements mitigation strategies**, including generating *Sigma* detection rules and prioritizing patches for vulnerable *Exchange servers*.

As illustrated by the above motivating example, CTI practices are typically organized into a pipeline consisting of four stages, each of which involves distinct reasoning tasks and technical solutions:

❶ **Contextualization**: Security teams must enrich raw observations (e.g., suspicious logs, network alerts, isolated IOCs) with contextual information to make them actionable. This includes mapping events to known threat identifiers such as CVEs or MITRE ATT&CK TTPs, linking indicators to malware families, and constructing coherent timelines of adversarial campaigns. **Involved techniques** in this stage include topic modeling to group related threat narratives, event extraction to identify structured incidents, knowledge base mapping to align content with known taxonomies, and information retrieval to ground outputs in relevant threat reports or databases.

❷ **Attribution**: Once threat contexts are enriched, security teams investigate the likely adversaries behind the activity. Attribution connects threat events to specific actor profiles or campaigns by analyzing shared TTPs, infrastructure reuse (e.g., IP, domain overlap), and stylistic patterns such as language use or operational cadence. **Involved techniques** here include named entity recognition (NER) to extract actors, malware, and victim entities; relation extraction to identify links among entities and events; structured event graph construction to represent sequences of observed behavior; and threat actor classification using learned behavioral profiles from historical data.

❸ **Prediction**: With an understanding of the adversary, security teams aim to forecast future threats, exploitation likelihood, and potential impact. This involves estimating the probability of exploitation for known vulnerabilities (e.g., EPSS scoring), anticipating which sectors or systems are likely to be targeted, and modeling campaign evolution. **Involved techniques** in this stage focus on historical

Table 1: Collected cybersecurity benchmarks for LLM-CTI.

| Benchmark | CTI Scenario | Coverage | | | | Unique Feature |
|-----------|--------------|:---:|:---:|:---:|:---:|----------------|
| | | ❶ | ❷ | ❸ | ❹ | |
| CTIBench (Alam et al., 2024) | General CTI tasks | ✓ | ✓ | ✓ | ✗ | Multi-choice questions (MCQ) |
| SevenLLM-Bench (Ji et al., 2024) | Report understanding | ✓ | ✓ | ✓ | ✓ | Synthetic instances, MCQ, QA |
| SWE-Bench (Jimenez et al., 2023) | Software bug fixing | ✗ | ✓ | ✗ | ✓ | Program analysis & patching |
| CyberTeam (Liu et al., 2025) | Blue-team threat hunting | ✓ | ✓ | ✓ | ✓ | Open-ended decision-making |

event correlation to find temporal patterns, temporal modeling to capture threat progression, and forecasting using time series or graph neural networks to predict propagation or escalation risks.

❹ **Mitigation**: Finally, CTI must support actionable decisions to reduce risk and guide incident response. This includes recommending specific patches, tuning detection signatures (e.g., YARA/Sigma rules (Alvarez, 2013; Roth & Patzke, 2017)), adjusting firewall or access control configurations, and drafting response playbooks. **Involved techniques** supporting this stage include mitigation mapping to associate observed TTPs or vulnerabilities with known defensive strategies, mitigation efficacy prediction to rank possible responses, and summarization to generate concise, structured remediation steps tailored to system environments.

> **Highlight ♀.** Involved techniques above not only define the operational CTI pipeline but also help explain the root causes of where vulnerabilities are likely to be introduced from threat landscape.

Section §4.2 and Appendix D.1 analyze root causes of vulnerabilities triggered by these techniques across CTI stages, while Appendix A provides additional details on the techniques involved.

## 2.2 MOTIVATION: LLMs REMAIN INSUFFICIENT IN VARIANT CTI TASKS

**Evaluation Datasets.** To evaluate LLM performance on CTI tasks, we leverage benchmarks (Alam et al., 2024; Ji et al., 2024; Jimenez et al., 2023; Liu et al., 2025) as well as real-world threat databases and platforms. The benchmarks are summarized in Table 1, which provide a broad coverage across all CTI stages and capture both structured (MCQ, QA) and unstructured (decision-making, patch generation) task formats. To unify their use, we standardize each instance into a CTI-oriented scenario explicitly aligned with one of the four CTI stages. For example, multi-choice questions (MCQs) from CTIBench and SevenLLM-Bench are reformulated into concrete threat hunting scenarios, such as analyzing a suspicious log entry to determine the relevant TTP or linking an IOC to a known malware family. Our preprocessing mitigates structural biases that could otherwise inflate LLM performance.

We provide the details of real-world databases (or platforms) in Appendix B.2, along with data statistics in Table 5. The prompt template used in evaluation is also included in Appendix B.6.

**Evaluated LLMs.** Our evaluation covers two complementary lines of models. First, we include industry-leading, general-purpose LLMs (e.g., GPT-5, Claude-Sonnet-4, Gemini-2.5), which represent state-of-the-art reasoning capabilities across domains. Second, we evaluate on open-source or API-accessible cybersecurity-specialized models (e.g., SecGPT (clouditera, 2025), DeepHat (Deep-Hat, 2025)), which are adapted to security operations through domain-specific training and curated CTI corpora. Investigating their performance gaps provides insights of vulnerabilities that cannot be fully addressed by either large-scale pretraining or cybersecurity-specialized adaptation. Appendix B.5 details all evaluated model and their versions.

**Model-specific Gap.** Across extensive evaluations on a broad range of CTI tasks (detailed introductions in B.3), Table 2 shows model-specific gaps: general-purpose LLMs dominate understanding-heavy and synthesis tasks (e.g., populating attack graphs), while the best cyber-specialized agents may outstand on semantic-driven or operational outputs (e.g., patch recommendation). In contextualization, general models post sizable wins, e.g., *Affected Systems* F1 peaks at ∼0.82–0.88 versus ∼0.55–0.56 for most cyber agents; *Source Reliability* AUC tops out at ∼0.91 versus a best cyber score of ∼0.74, which reflects constrained long-context retrieval and instruction-following that also present in text generation tasks such as *Threat Report Alignment* and *Event Timeline*. In attribution and forecasting, the trend largely holds: general LLMs lead *Threat Actor Linking* (Acc up to ∼0.89) and *Exploit Likelihood* (AUC up to ∼0.86). Notably, targeted domain tuning can flip certain edges: a top

Table 2: Evaluation of LLMs on CTI tasks across four CTI stages. Industry-leading general-purpose LLMs (left) are compared with cybersecurity-specialized models (right). Detailed CTI task descriptions (with examples) are provided in B.3. Model names and versions are introduced in B.5.

| CTI Task | Metric | Industry-leading, general-purpose | | | | | | | | Cybersecurity-specialized | | | | | | |
|---|---|---|---|---|---|---|---|---|---|---|---|---|---|---|---|---|
| | | G5 | Go4 | CLD | GEM | LL70 | MIX | QWN | GRK | FSC | CB0 | ZYS | LLY | CBS | SPT | DHT |
| **❶ Contextualization** | | | | | | | | | | | | | | | | |
| Affected Systems | F1 | .822 | .801 | .757 | .613 | .882 | .663 | .747 | .819 | .432 | .418 | .553 | .566 | .562 | .554 | .559 |
| Attack Infrastructure | F1 | .863 | .741 | .614 | .578 | .853 | .628 | .616 | .493 | .507 | .492 | .612 | .625 | .618 | .611 | .616 |
| Vulnerability Linking | Acc | .652 | .633 | .602 | .574 | .649 | .533 | .521 | .497 | .512 | .496 | .621 | .633 | .629 | .622 | .628 |
| Malware Family Mapping | F1 | .681 | .659 | .635 | .602 | .584 | .567 | .551 | .529 | .541 | .526 | .639 | .652 | .646 | .641 | .648 |
| IOC Normalization | F1 | .721 | .707 | .682 | .661 | .642 | .623 | .609 | .593 | .602 | .589 | .678 | .689 | .684 | .678 | .683 |
| Threat Report Alignment | BLEU | .429 | .218 | .206 | .396 | .486 | .279 | .271 | .363 | .159 | .352 | .441 | .348 | .244 | .139 | .342 |
| Event Timeline Construction | BLEU | .563 | .549 | .532 | .519 | .504 | .492 | .478 | .468 | .472 | .459 | .594 | .602 | .599 | .593 | .597 |
| Graph Population | Acc | .793 | .676 | .559 | .539 | .724 | .507 | .693 | .478 | .487 | .472 | .421 | .334 | .629 | .424 | .628 |
| Source Reliability Scoring | AUC | .912 | .894 | .773 | .861 | .745 | .631 | .717 | .753 | .712 | .699 | .738 | .547 | .642 | .537 | .641 |
| **❷ Attribution** | | | | | | | | | | | | | | | | |
| Threat Actor Linking | Acc | .892 | .871 | .652 | .822 | .598 | .773 | .753 | .528 | .643 | .526 | .704 | .413 | .707 | .771 | .608 |
| TTP Extraction | F1 | .751 | .738 | .724 | .703 | .478 | .669 | .654 | .642 | .654 | .639 | .724 | .537 | .731 | .726 | .732 |
| Campaign Attribution | Acc | .712 | .691 | .671 | .649 | .631 | .607 | .586 | .567 | .578 | .563 | .694 | .703 | .699 | .694 | .701 |
| Infrastructure Reuse | F1 | .677 | .656 | .636 | .609 | .591 | .574 | .556 | .534 | .548 | .531 | .688 | .528 | .692 | .754 | .603 |
| Language/Style Profiling | Acc | .598 | .581 | .561 | .539 | .521 | .503 | .488 | .475 | .489 | .476 | .632 | .643 | .638 | .633 | .639 |
| False Flag Detection | F1 | .679 | .526 | .501 | .486 | .672 | .459 | .547 | .436 | .444 | .431 | .574 | .286 | .462 | .576 | .582 |
| Evidence Weighting | BLEU | .362 | .247 | .131 | .226 | .402 | .288 | .178 | .207 | .073 | .059 | .097 | .007 | .102 | .197 | .083 |
| Relation Graph Building | F1 | .642 | .628 | .611 | .595 | .579 | .562 | .547 | .533 | .544 | .528 | .675 | .683 | .678 | .673 | .679 |
| **❸ Prediction** | | | | | | | | | | | | | | | | |
| Exploit Likelihood | AUC | .821 | .806 | .792 | .771 | .856 | .742 | .629 | .714 | .519 | .703 | .742 | .559 | .764 | .653 | .759 |
| Impact Forecast | BLEU | .498 | .383 | .271 | .354 | .441 | .226 | .213 | .202 | .108 | .094 | .207 | .119 | .046 | .115 | .221 |
| Target Sector Prediction | Acc | .841 | .759 | .743 | .819 | .602 | .756 | .712 | .553 | .564 | .679 | .502 | .311 | .623 | .415 | .619 |
| Campaign Escalation | AUC | .683 | .627 | .611 | .598 | .582 | .568 | .554 | .541 | .548 | .532 | .607 | .518 | .652 | .607 | .613 |
| **❹ Mitigation** | | | | | | | | | | | | | | | | |
| Patch Recommendation | F1 | .702 | .679 | .659 | .636 | .718 | .601 | .583 | .671 | .582 | .567 | .632 | .442 | .629 | .641 | .446 |
| Rule Generation (YARA) | BLEU | .382 | .216 | .339 | .482 | .267 | .213 | .337 | .184 | .231 | .307 | .281 | .089 | .094 | .287 | .202 |
| Response Summarization | BLEU | .514 | .399 | .286 | .567 | .252 | .236 | .422 | .311 | .118 | .304 | .315 | .227 | .433 | .126 | .208 |
| Mitigation–TTP Mapping | Acc | .672 | .652 | .633 | .611 | .596 | .578 | .561 | .548 | .559 | .544 | .613 | .626 | .631 | .624 | .629 |
| Defensive Playbook Gen | BLEU | .586 | .572 | .557 | .537 | .522 | .508 | .495 | .482 | .491 | .476 | .561 | .572 | .576 | .571 | .575 |
| Countermeasure Ranking | NDCG | .591 | .574 | .561 | .547 | .532 | .519 | .504 | .494 | .503 | .489 | .623 | .533 | .629 | .424 | .429 |
| Incident Ticket Generation | Acc | .831 | .716 | .601 | .682 | .868 | .753 | .639 | .628 | .536 | .421 | .653 | .464 | .668 | .602 | .564 |

cyber agent outperforms on *Infrastructure Reuse* (F1 $\sim$0.75 vs $\sim$0.68 best general), and cyber models consistently beat generals on operational ranking/classification such as *Countermeasure Ranking* (NDCG $\sim$0.62–0.63 vs $\sim$0.57–0.59), suggesting influences from environmental (enterprise) contexts. Performance among cyber agents is also more uneven (e.g., low F1 on *False Flag Detection* for some models), indicating sensitivity to the quality and coverage of training (or retrieving) evidence.

**Universal Gap.** We also observed universal gaps among all models: limited IOC normalization and CVE linking under obfuscation, inconsistent TTP extraction across retrieval-augmented reasoning, and weak temporal coherence in timelines/escalation forecasts. We also observe shallow reliance on real-world evidence in report alignment (low BLEU) and format errors in ticket/playbook generation. These deficiencies present with both larger online models (e.g., GPT-5) and cyber-specific agents, implying inherent limitations that may be triggered by the nature of the threat landscape.

Motivated by these gaps, we investigate vulnerabilities based on the concrete **failure modes** of LLMs in CTI tasks. We aim to build a systematic view about where models break down and why these vulnerabilities persist despite large-scale pretraining or domain-specific adaptation.

# 3 METHODOLOGY: CATEGORIZE FAILURE INSTANCES

After conducting the large-scale evaluation described in §2, we categorize the resulting failure cases to better understand the limitations of LLMs. This process is guided by three research questions:

**RQ$_1$:** How can we identify "failure" especially in tasks that lack hard-label annotations?
**RQ$_2$:** How can we determine the finite scope of vulnerabilities (i.e., failure modes)?
**RQ$_3$:** How can we efficiently categorize large-scale instances into these failure modes?

Overall, **we do not fully trust LLM-as-judge or model-generated confidence** to detect failure cases, due to the lack of transparency and the risk of "self-rationalized" reasoning. Nevertheless, the large scale of the evaluation set requires us to categorize instances efficiently beyond purely manual efforts. To address this, we propose a **stratification** approach that partitions instances based on their

Figure 3: Overview of method to categorize failure instances (addressing $RQ_1$-$RQ_3$).

"failure" depth, followed by an **autoregressive method** combined with **human-in-the-loop** efforts to resolve the above RQs. Figure 3 illustrates the workflow. Algorithms are deferred to Appendix C.1

## 3.1 STRATIFYING INSTANCES WITHOUT HARD-LABEL ANNOTATIONS ($RQ_1$)

Most cyber threat intelligence tasks, such as mitigation rule generation, lack hard labels that allow a binary correct/incorrect assessment. However, reference materials are typically available from CTI reports and vulnerability advisories (e.g.,Cybersecurity and Infrastructure Security Agency (CISA) (2024); CVE Program (2024)). We therefore leverage the reference-based metrics (e.g., text similarity by BLEU) to quantify the matching degree between model outputs and authoritative sources.

Based on the calculated similarity scores, we stratify instances by their "failure" depth, ranging from severe mismatches to partial alignment and near matches. Specifically, we rank all instances and partition them into quantile-based strata (5% bins). Within each stratum, we initially inspect some "correct" and "failed" samples and record their score distributions into two groups. For the remaining instances, we classify them as "failed" if their scores fall within the range associated with failed samples; if they fall in the overlapping region between the two groups, we conduct additional manual inspection. We terminate the process once no new failure modes emerge (as defined in §3.2) and the distribution of failure modes (in §3.3) across strata converges to a stable ratio. **Practically, this process is efficient as it requires manually inspecting no more than 3% of instances across all CTI tasks.** For clarity, the detailed algorithm is provided in Algorithm 1.

## 3.2 AUTOREGRESSIVE FAILURE MODE DETERMINATION ($RQ_2$)

We avoid of using LLMs to directly determine failure modes (i.e., vulnerabilities), instead, we design an **iterative** process alternating between human annotation and LLM-assisted classification. Let $\mathcal{D} = \{x_i\}_{i=1}^N$ denote the set of failure instances (from stratification results in §3.1).

**Step 1 (Initialization).** Human annotators randomly inspect a small subset $\mathcal{D}_0 \subset \mathcal{D}$ to derive an initial taxonomy of failure modes, $\mathcal{T}_0 = t_1, \ldots, t_k$.

**Step 2 (LLM classification).** For each remaining instance $x_j \in \mathcal{D} \setminus \mathcal{D}_0$, an LLM assigns a label $y_j \in \mathcal{T}_m \cup \{\texttt{other}\}$, where $\mathcal{T}_m$ is the taxonomy after $m$ iterations.

**Step 3 (Refinement).** Instances labeled as $\texttt{other}$, i.e., $\mathcal{O}_m = \{x_j \mid y_j = \texttt{other}\}$, are further inspected by human annotators. If new failure patterns are identified, new modes $\Delta\mathcal{T}$ are added to the taxonomy, yielding $\mathcal{T}_{m+1} = \mathcal{T}_m \cup \Delta\mathcal{T}$.

This loop repeats until $\Delta\mathcal{T} = \varnothing$, i.e., no new failure modes are found. We ultimately find the stabilized set of failure modes: $\mathcal{T}^* = \lim_{m\to\infty} \mathcal{T}_m$ (see Algorithm 2).

Note that determining vulnerability types (failure modes) requires comparing failure cases with the ground-truth answers (or references). The specific methods tailored to each vulnerability type are detailed in Appendix C.2.

## 3.3 HUMAN-IN-THE-LOOP CATEGORIZATION OF FAILURE INSTANCES ($RQ_3$)

To balance reliability and scalability in large-scale categorization, we integrate human inspection with multi-agent LLM decisions:

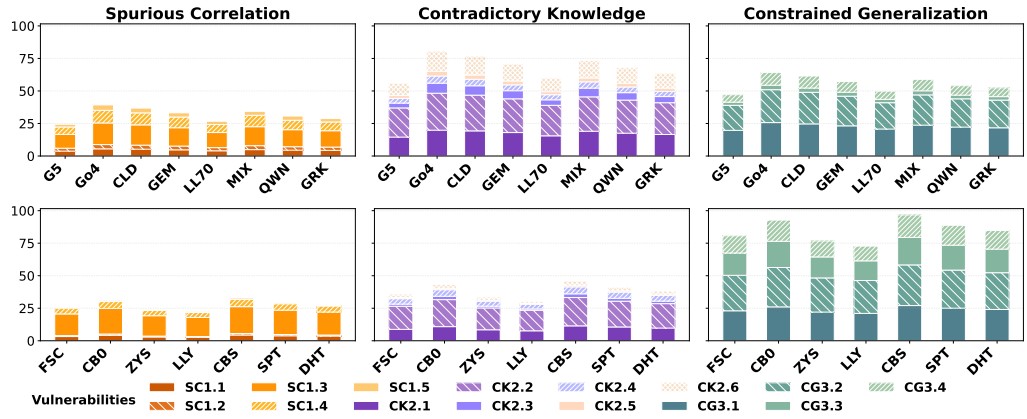

Figure 4: Varying proportions of vulnerabilities (types listed in Figure 2). Note that different vulnerabilities can intertwine within the same instance, which is particularly common in Contradictory Knowledge (CK) and Constrained Generalization (CG), less common in Spurious Correlation (SC).

**Step 1 (Multi-agent decision).** For each threat instance $x_i$ requiring failure mode classification, we construct a model set $\Theta = \{\texttt{GPT-5}, \texttt{Llama-4-17B}, \texttt{Gemini-2.5}, \texttt{Claude-Sonnet-4}\}$. Each model independently proposes an initial label $\hat{y}_i^{(1)} \in \mathcal{T}$ during the first execution.

**Step 2 (Repetition for stability).** Motivated by evidence that unstable internal knowledge may lead to fluctuations across runs (Kumar et al., 2024), we execute a second round of multi-agent deliberation. In this round, each LLM observes the predictions $\hat{y}_i^{(1)}$ from other models, refines its reasoning, and then proposes a new label $\hat{y}_i^{(2)}$ for the same instance.

**Step 3 (Human verification).** Human inspectors evaluate consistency along two dimensions: (1) if agent-level votes show disagreement (lack of consensus), or (2) if $\hat{y}_i^{(1)} \neq \hat{y}_i^{(2)}$ (fluctuation across rounds). In either case, the instance is flagged as uncertain:

$$\mathcal{U} = \{x_i \mid \text{Var}(\{\hat{y}_i^{(a)}\}) > 0 \ \text{ or } \ \hat{y}_i^{(1)} \neq \hat{y}_i^{(2)}\}$$

where $\{\hat{y}_i^{(a)}\}$ denotes the set of labels assigned by agents $a \in \Theta$. All $x_i \in \mathcal{U}$ are then inspected by human annotators for final determination.

The multi-agent collaboration handles the majority of straightforward cases efficiently, while humans focus only on instances with instability or misalignment (empirically, less than 1.8% cases in this step), thus balancing scalability with reliability in failure mode categorization (Algorithm in 3).

## 4 RESEARCH FINDING

Our categorization method leads to a detailed list of vulnerability types, as presented in Figure 2. Building on this, we further investigate specific failure cases to address the following questions:

**RQ$_4$:** How do different LLMs present varying proportions of vulnerabilities?
**RQ$_5$:** What are the primary root causes that give rise to these vulnerabilities?
**RQ$_6$:** How do multiple vulnerabilities interact and become intertwined within the same instance?

### 4.1 VARYING PERFORMANCE OF LLMS (RQ$_4$)

We first study the varying performance of different models. As shown in Figure 4, and consistent with the distributions across vulnerabilities in Figure 2, our analysis reveals clear differences in how various LLMs handle distinct categories of errors: For vulnerabilities that directly impair LLM inference (e.g., co-mentioned but irrelevant mitigation strategies or outdated metadata that reduce retrieval effectiveness), general-purpose LLMs tend to accumulate a higher volume of failures. This indicates that their broad but non-specialized training leaves them vulnerable to overfitting and misinterpreting contextual evidence.

Table 3: Summarized root causes of vulnerabilities, with detailed analysis in §4.2 and Appendix D.1.

| Vulnerability | Subtype | Stage | How Vulnerability Happens |
|---|---|---|---|
| Spurious Correlation | (1.1) Co-mention bias | ❶❷❸❹ | RAG surfaces unrelated but co-mentioned mitigations or vulnerabilities, causing LLMs to infer false associations. |
| | (1.2) Exploitation bias | ❶❷❸❹ | Reused IoCs across incidents mislead models into over-attributing infrastructure reuse or ongoing activity. |
| | (1.3) Confounding factors | ❶❷❸❹ | Non-causal variables (e.g., geography, org tags) treated as causal predictors for exploit likelihood. |
| | (1.4) Skewed source | ❶ | Overrepresentation of certain feeds biases patch prioritization and defensive rules toward specific vendors. |
| | (1.5) Hierarchical metadata | ❶❷❸ | Structured taxonomies (e.g., ATT&CK chains) interpreted as causal orderings rather than descriptive metadata. |
| Contradictory Knowledge | (2.1) Temporal contradiction | ❶❷❸❹ | Outdated advisories conflict with newer reports, confusing model reasoning about valid mitigations. |
| | (2.2) Conflicting reports | ❶❷❸❹ | Disagreement between sources on documented attack context, actor, dependencies, or other evidence. |
| | (2.3) Semantic conflict | ❶ | Different naming/taxonomies (e.g., PlugX vs. Korplug) cause inconsistency. |
| | (2.4) Divergent structures | ❶❷❸ | JSON feeds vs. unstructured PDFs produce inconsistencies when fused into CTI mapping (e.g., TTP to patch). |
| | (2.5) Misaligned standards | ❷❸❹ | Differences in scoring frameworks (CVSS vs. vendor ratings) yield contradictory threat intelligence. |
| | (2.6) Counteracting generation | ❷❸❹ | Reasoning and generation on CTI tasks disrupt LLM safety alignment, causing unstable outputs. |
| Constrained Generalization | (3.1) Distributional bias | ❶ | Training on limited language/region corpora hinders generalization to unseen threat contexts. |
| | (3.2) Unseen patterns | ❷❸❹ | Zero-day exploits exhibit novel paths absent from model training, degrading exploitability forecasts. |
| | (3.3) Overfitted reasoning | ❷❸ | Memorized patterns (e.g., CVE-TTP) lead to brittle linking and ineffective generation. |
| | (3.4) Environmental unawareness | ❷❸❹ | Models overlook local system/sector-specific dependencies, producing ineffective mitigation strategies. |

In contrast, for vulnerabilities rooted in the data used during fine-tuning (e.g., confounding factors in the threat corpus), cyber agents such as Foundation-Sec or Cyber-Zero tend to produce less reliable outputs (e.g., forecasting exploitability with contradictory PoCs). Such vulnerabilities act as data poisoning, where spurious correlations and contradictory knowledge can arise either intentionally (introduced by adversaries) or unintentionally (inherent in the fragmented cyber threat landscape).

Besides, we also observed that all models show constrained performance when confronted with emerging or zero-day threats. Failure ratios are particularly high in specialized cyber agents, whose localized nature and narrower pre-trained knowledge bases limit their adaptability. Once their training cutoffs are reached, they lack the generalization capacity to extrapolate effectively to novel threats.

## 4.2 Root Causes of Vulnerabilities (RQ₅)

During failure categorization (§3), we also gained detailed insights into how different vulnerabilities are triggered by the used techniques among CTI tasks. Table 3 summarizes these causes in alignment with the vulnerability types shown in Figure 2. Here, we highlight some representative cases:

**Co-mention bias (1.1) in ❶ contextualization.** Co-mention bias emerges in contextualization when retrieval systems treat co-occurring entities in raw reports as causally linked. The security bulletin may list multiple vulnerabilities or mitigation strategies together in a single section, even though only one is relevant to a particular incident. Topic modeling or knowledge base mapping that lack fine-grained data disambiguation may surface all co-mentioned entities as equally relevant. This inflates the context with spurious links (e.g., assigning unrelated CVEs or MITRE TTPs to the same intrusion set), misguiding LLMs to propagate spurious associations.

> **Case Study.** Microsoft Patch Tuesday advisories often list many CVEs under one product (e.g., *Windows Server*) (Microsoft Security Response Center, 2003). The contextualization pipeline ingests the bulletin and a threat report cites *CVE-2021-34527 (PrintNightmare)*, GPT-5 incorrectly infers that all co-mentioned CVEs were exploited in the same campaign, creating false links between benign vulnerabilities and active threats.

**Conflicting report (2.2) in ❹ Mitigation.** Mitigation is influenced by different testing environment and study scope of different vendors. For example, one advisory may claim that a patch fully resolves an exploited vulnerability, while another report provides evidence that attackers continued exploitation through chained dependencies, such as leveraging an adjacent misconfiguration. Mitigation mapping that relies on these contradictory reports may overestimate or underestimate the coverage of a particular countermeasure. Similarly, summarization pipelines may output inconsistent defensive playbooks, some emphasizing patch deployment while others prioritize compensating controls. These varying mitigation strategies leaving LLMs uncertain about actions that truly address threat incidents.

> **Case Study.** In the 2021 *Microsoft Exchange "Hafnium"* case (Microsoft Threat Intelligence & Microsoft 365 Security, 2021), some advisories claimed patches fully mitigated the threat, while others warned of persistent web shells post-patch. The conflicting reports gave Qwen retriever false assurance about their mitigation effectiveness.

A comprehensive analysis with case studies across all vulnerabilities is provided in Appendix D.1.

### 4.3 STUDY OF INTERTWINED VULNERABILITIES (RQ$_6$)

We also notice that multiple vulnerabilities frequently interact and compound within the same instance. The root cause lies in the coupled nature of the CTI pipeline: early retrieval errors such as co-mention bias or skewed sources propagate forward as "facts," which downstream attribution or prediction models cannot easily disconfirm. This effect is amplified by heterogeneous and drifting evidence, wherein reports that differ temporally, semantically, or structurally force models to merge incompatible signals. Under such uncertainty, models lean on inductive shortcuts, such as memorized actor–TTP links, which transform incomplete or conflicting metadata into brittle reasoning. Finally, the hierarchical and dependency-driven structure of attack chains and mitigation mappings means that once one link is misread, the error cascades into adjacent stages. As a result, intertwined vulnerabilities like co-mention bias plus temporal contradiction, or unseen patterns combined with environmental unawareness, emerge as self-reinforcing loops that entangle multiple failures within the same CTI instance.

Appendix D.2 also details additional analyses with case studies to complement our discussions.

> **Design Insight** 💡**.** Vulnerabilities studied in this work are not solely caused by model design, but by the **inherently fragmented and adversarial threat landscape**. Noise such as contradictory proofs-of-concept, reused indicators, or incomplete metadata can propagate into models. Addressing these vulnerabilities therefore requires **combining data curation, adversarially aware fine-tuning, and inference-time safeguards** to integratively control the adversarial influences.

## 5 RELATED WORK

**LLM-as-agent.** LLM-based agents have been explored across diverse domains such as education (Chu et al., 2025), scientific discovery (Schmidgall et al., 2025), data science (Hong et al., 2024), and urban mobility modeling (Wang et al., 2024). These applications highlight the ability of LLM agents to decompose complex tasks, integrate external tools, and generate executable outputs. In cybersecurity, distributed detection frameworks (Dong et al., 2023) and adaptive rule-evasion defenses (Uetz et al., 2024) illustrate how agent-like systems can strengthen enterprise protection.

**LLMs for cybersecurity.** LLMs are increasingly leveraged for both offensive and defensive cybersecurity tasks, owing to their strong natural language understanding and reasoning capabilities. In static analysis, LSAST augments traditional SAST tools with dynamic vulnerability knowledge (Keltek et al., 2025), while hybrid systems use LLM-driven preprocessing and explanation to improve anomaly detection in IoT (Ghimire et al., 2025). Autonomous agents have demonstrated the ability to exploit one-day vulnerabilities (Fang et al., 2024b) and guide fuzzing across multi-hop library dependencies (Zhou et al., 2024). Additional applications include enhancing intrusion detection (G. Lira et al., 2024), supporting large-scale code review (Sun et al., 2025), and enabling malware tracking through dataset augmentation and semantic analysis (Yu et al., 2024).

**Vulnerabilities of LLMs.** Despite their promise, LLM-based agents exhibit critical vulnerabilities that undermine reliability in high-stakes domains. Recent work shows that malfunction amplification attacks can cascade small reasoning errors into severe misjudgments (Zhang et al., 2024). Other studies highlight vulnerabilities to data poisoning (Wang et al., 2025), adversarial prompts Debenedetti et al. (2024), and misalignment (Fang et al., 2024a), all of which raise concerns for deploying LLMs in security-sensitive environments.

## 6 CONCLUSION

This work presents systematic yet intensive studies of intrinsic vulnerabilities that constrain the effectiveness of LLMs in cyber threat intelligence. By combining large-scale benchmark evaluations with real-world CTI reports, we uncover three fundamental failure modes (spurious correlations, contradictory knowledge, and constrained generalization) that persist across multiple CTI stages and workflows. Our autoregressive, human-in-the-loop methodology enables reliable categorization of failure instances and provides insights into how these limitations emerge and propagate. These findings not only reveal blind spots in current LLM reasoning but also chart a path toward more principled model adaptations and robust cyber agent design.

ETHICS STATEMENT

This study does not raise ethical concerns. All experiments were conducted using publicly available cyber threat intelligence (CTI) reports, standardized vulnerability databases (e.g., CVE, NVD, CISA KEV), and openly accessible vendor advisories. No proprietary, sensitive, or personally identifiable data were used. The research strictly adheres to the licensing and usage terms of all data sources.

REPRODUCIBILITY STATEMENT

To ensure reproducibility, we provide the full implementation of our framework, including stratification algorithms, evaluation scripts, and experimental settings, in an anonymous GitHub repository linked in the Introduction. The repository contains detailed instructions that allow independent researchers to follow our results and build upon our analyses.

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

## A  COMPLEMENTARY DETAILS OF CTI

This appendix section provides additional details on how different CTI stages are conducted with the involvement of various techniques, complementing Section §2.1.

### A.1  ❶ CONTEXTUALIZATION

The contextualization stage transforms raw, fragmented observations into structured and actionable intelligence. Below we elaborate on the specific techniques commonly employed in real world CTI operations, with a natural sequence in practice.

- **Topic Modeling.** Topic modeling techniques such as Latent Dirichlet Allocation (LDA) (Blei et al., 2003), Non-negative Matrix Factorization (Lee & Seung, 1999), and more recent neural topic models (Dieng et al., 2020) are applied to large corpora of unstructured CTI text, including threat reports, incident tickets, and log annotations. These models group documents or paragraphs into coherent themes, enabling analysts to identify clusters of related activity such as "phishing campaigns leveraging Office macros" or "ransomware families exploiting VPN vulnerabilities." In practice, topic modeling supports early triage by prioritizing threat feeds that share thematic overlap with active campaigns in a given sector.

- **Event Extraction.** Event extraction identifies structured incidents—*who did what, when, and how*—from unstructured logs or reports. Natural Language Processing (NLP) pipelines detect entities such as vulnerabilities (CVEs), indicators of compromise (IP addresses, file hashes), and TTPs (MITRE ATT&CK techniques (Strom et al., 2018)), then associate them with temporal markers and actor actions. In Security Operations Centers (SOCs), event extraction enables long-form reports to be converted into structured JSON objects or STIX bundles (oas, 2017), which can be automatically ingested into SIEM platforms such as Splunk or Elastic. This reduces manual parsing effort and ensures consistent representation across heterogeneous data sources.

- **Knowledge Base Mapping.** Knowledge base mapping aligns extracted entities and events with standardized taxonomies such as CVE (CVE Program, 2024), CWE (MITRE Corporation, 2024b), MITRE ATT&CK (Strom et al., 2018), and CAPEC (MITRE Corporation, 2024a). This process typically involves both exact matching (e.g., direct CVE ID resolution) and approximate entity linking (e.g., mapping the phrase "remote PowerShell execution" to ATT&CK T1059.001). In CTI practice, knowledge base mapping is essential for interoperability: intelligence can be shared across organizations using a common language of identifiers, enabling correlation of local incidents with global adversary behaviors.

- **Information Retrieval.** Information retrieval systems allow analysts to ground their findings in relevant historical reports and external databases. Implementations include keyword- and embedding-based retrieval from proprietary CTI feeds (e.g., Recorded Future, Mandiant Advantage), open-source repositories (e.g., AlienVault OTX, AbuseIPDB), and structured vulnerability catalogs such as NVD (National Institute of Standards and Technology (NIST), 2024) and the CISA KEV catalog (Cybersecurity and Infrastructure Security Agency (CISA), 2024). Retrieval pipelines often combine lightweight keyword filters for precision with dense embedding search for semantic coverage. Analysts use these systems to verify whether a newly observed domain has prior associations with known malware campaigns, or whether a vulnerability is actively being exploited.

- **Operational Integration.** In SOC environments, the above techniques are integrated into semi-automated pipelines. For example, a suspicious DNS query may trigger automated enrichment:

(i) topic modeling identifies thematic overlap with phishing campaigns, (ii) event extraction links the query to specific malware families, (iii) knowledge base mapping ties the observed behavior to ATT&CK techniques, and (iv) information retrieval retrieves prior cases of infrastructure reuse. The result is a structured incident summary that equips analysts with actionable context for attribution, prediction, and mitigation.

## A.2  ❷ ATTRIBUTION

Attribution in cyber threat intelligence involves identifying the adversary or campaign responsible for observed malicious activity. This process requires combining technical indicators with contextual and behavioral evidence. Below we describe the primary techniques used in real-world practice, along with how they are operationalized by analysts.

- **Named Entity Recognition (NER).** NER systems are used to extract structured entities such as malware families, infrastructure elements, or actor names from unstructured reports and incident logs. For example, extracting references to *APT28*, *Mimikatz*, or *QakBot* across heterogeneous feeds helps analysts consolidate threat narratives (Barnes & Kordzadeh, 2021).

- **Relation and Event Extraction.** Beyond isolated entities, attribution requires uncovering relationships among adversary tactics, techniques, and infrastructure. Relation extraction techniques map, for instance, shared IP ranges between phishing campaigns or code reuse across malware variants, while event extraction captures sequences of actions such as exploitation followed by lateral movement (Husari et al., 2018a; Zhang et al., 2019).

- **Infrastructure Correlation.** Adversaries frequently reuse or repurpose command-and-control (C2) infrastructure. CTI teams apply graph-based correlation to link domain registrations, TLS certificates, and hosting providers, enabling attribution of otherwise fragmented observations to known actor toolkits or campaigns (Stone-Gross et al., 2009; Antonakakis et al., 2017).

- **Stylistic and Linguistic Profiling.** Analysts also consider linguistic cues in adversary communications or malware code artifacts. Stylometry and compilation fingerprinting methods can identify patterns in variable naming, debugging strings, or grammar usage, which help to tie malware development back to particular groups (Alonso et al., 2019; Rosenblum et al., 2011).

- **Campaign Graph Construction.** To synthesize diverse evidence, CTI analysts build structured graphs of adversarial campaigns, linking entities, infrastructure, and TTPs across incidents. Graph construction enables propagation of attribution hypotheses and detection of actor evolution over time (Park et al., 2020; Yang et al., 2021).

## A.3  ❸ PREDICTION

Prediction in CTI involves forecasting adversarial actions, exploitation likelihood, and campaign evolution. Unlike contextualization and attribution, prediction requires reasoning under temporal uncertainty and incomplete information. In practice, analysts and automated systems deploy a range of data-driven and model-based techniques to anticipate threats:

- **Historical correlation and trend analysis.** Analysts correlate prior incidents and intrusion campaigns to identify recurring attacker playbooks. For example, statistical methods on longitudinal CVE exploitation data help assess whether recently disclosed vulnerabilities follow exploitation trends of past families (Bilge & Dumitras, 2012; Allodi & Massacci, 2014).

- **Exploit prediction scoring.** Models such as the Exploit Prediction Scoring System (EPSS) estimate the probability that a vulnerability will be exploited within a given time window, using features such as vulnerability metadata, CVSS scores, and real-world exploit observations (Jacobs & Romanosky, 2019; 2021).

- **Temporal modeling of campaign progression.** Recurrent neural networks and temporal point processes capture how campaigns unfold over time, modeling likely transitions from initial access to follow-on payloads such as ransomware (Wang et al., 2019; Okutan & Yilmaz, 2020).

- **Temporal forcasting.** LLMs predict which TTPs an actor is likely to employ next or what's impact. This supports proactive defense, such as generating detection rules for tactics not yet observed in the ongoing campaign (Sood & Enbody, 2013; Husari et al., 2018b).

- **Threat propagation simulation (synthesis).** Agent-based and epidemic-style models simulate the spread of malware or worms across interconnected systems, forecasting infection curves and propagation likelihood (Xu et al., 2012; Liu et al., 2015).

## A.4  ❹ MITIGATION

Mitigation is the final stage of cyber threat intelligence (CTI), where enriched analysis and attribution results are translated into concrete defensive measures. Unlike contextualization or attribution, which primarily generate insights, mitigation requires actionable transformations that directly alter security posture. Below, we introduce the major classes of techniques commonly adopted in real-world CTI practice.

- **Detection rule generation.** Security teams design, validate, and deploy detection rules in languages such as Sigma and YARA to capture specific threat behaviors. In practice, detection rules are tuned iteratively: analysts translate threat reports into rule signatures, test them against telemetry or sandbox logs, and refine them to reduce false positives while ensuring coverage of attacker tradecraft.

- **Mitigation efficacy evaluation.** Beyond applying countermeasures, analysts must assess their effectiveness. Approaches include red-teaming exercises, breach-and-attack simulation (BAS) platforms, and adversary emulation scenarios that replay known TTPs to test whether mitigations succeed in preventing, detecting, or containing malicious activity (Shostack, 2014; Peisert et al., 2021).

- **Response playbook recommendation.** Structured playbooks standardize incident response by encoding lessons from CTI. These include step-by-step containment and recovery actions tailored to adversary campaigns (e.g., disabling compromised accounts, isolating infected subnets). Orchestration tools such as SOAR platforms automate playbook execution, integrating CTI feeds into dynamic workflows (Mitnick & Vamosi, 2018; Scarfone & Grance, 2012).

- **Summarization.** Finally, mitigation intelligence is communicated through concise, contextualized reports for executives and IT operators. Summarization synthesizes prioritized vulnerabilities, mapped mitigations, and recommended workflows. This ensures decision makers understand trade-offs between operational impact and security gains, and supports cross-team coordination in enterprise-scale defense (Mavroeidis & Bromander, 2018; Anwar et al., 2021).

## B  ADDITIONAL DETAILS OF CTI EXPERIMENT FOR §2

### B.1  USED BENCHMARK IN EVALUATION

Table 4 presents the original scales of the datasets used in our evaluation.

Table 4: Used cybersecurity benchmarks in our evaluations.

| Benchmark | Focus | #Data | #Task | #Source |
| --- | --- | --- | --- | --- |
| CTIBench (Alam et al., 2024) | Cyber Threat Intelligence | 5,610 | 5 | N/A |
| SevenLLM-Bench (Ji et al., 2024) | Report Analyzing | 92,701 | 28 | N/A |
| SWE-Bench (Jimenez et al., 2023) | Bug fixing | 2,294 | 12 | 1 |
| CYBERTEAM (Liu et al., 2025) | Blue-team threat hunting | 452,293 | 30 | 23 |

### B.2  USED REAL-WORLD DATABASES AND PLATFORMS IN EVALUATION

In addition to benchmarks, we incorporate several real-world cybersecurity databases and intelligence platforms to ensure that our evaluation settings reflect practical CTI usage. Each database provides complementary coverage across the CTI stages (§2.1), and we detail both their scope and our methodology for leveraging their information.

**National Vulnerability Database (NVD).** The NVD (National Institute of Standards and Technology (NIST), 2024) serves as the canonical repository for software vulnerabilities and their CVSS severity scores. We utilize NVD entries to support all ❶–❹ tasks. Specifically, we map vulnerabilities in benchmark items to NVD records in order to standardize CVE identifiers, extract official CVSS vector strings, and obtain temporal metadata (publication and modification dates). These fields allow us to align system-environment observations with ground-truth vulnerability characteristics, and to test forecasting models that predict exploit likelihood (e.g., by contrasting NVD base scores against EPSS-derived estimates).

**Exploit Prediction Scoring System (EPSS).** EPSS (Jacobs & Romanosky, 2021) provides probabilistic estimates of the likelihood that a given CVE will be exploited in the wild. We use EPSS scores directly for ❸ prediction tasks, both as a source of labels (ground-truth exploitation likelihood) and as a reference distribution to evaluate calibration of LLM-based forecasts. EPSS time-series updates also enable temporal correlation experiments, where we assess whether models capture shifts in exploitation likelihood following disclosure, patch release, or threat-actor reuse.

**MITRE ATT&CK and CAPEC.** The ATT&CK (mit, 2020) knowledge base encodes adversarial tactics, techniques, and procedures (TTPs) in a structured taxonomy. We map CTI tasks involving entity extraction, campaign attribution, and mitigation alignment to ATT&CK entries. For instance, when evaluating ❷ attribution, extracted TTPs from LLM outputs are compared against ATT&CK technique identifiers to assess correctness. We also leverage CAPEC (Common Attack Pattern Enumeration and Classification) (MITRE Corporation, 2024a) to validate abstract attack patterns referenced in benchmark items, particularly for mapping contextualized logs or IOCs to higher-level adversarial behaviors.

**MISP (Malware Information Sharing Platform).** MISP (MISP Project, 2011) serves as a community-driven threat intelligence sharing platform, containing structured feeds of indicators of compromise (IOCs), malware samples, and infrastructure metadata. We use MISP to enrich contextualization tasks (❶) by grounding benchmark instances in realistic IOC–malware–actor relationships. For example, when a dataset item involves resolving a suspicious domain, we verify its presence in MISP feeds and align it to associated threat actors or malware families. This enrichment supports evaluation of LLMs' ability to normalize IOCs and link them to campaigns.

**VirusTotal.** VirusTotal (Google LLC, 2004) aggregates antivirus detections and malware analysis results across a large corpus of submitted files, domains, and URLs. We leverage VirusTotal reports for ❶ contextualization and ❷ attribution tasks. In contextualization, VirusTotal tags (e.g., malware family labels, sandbox behavior summaries) serve as auxiliary ground-truth for tasks like malware family mapping. In attribution, we analyze infrastructure overlap by checking whether related domains or IPs have been co-reported in VirusTotal samples, allowing us to validate LLM predictions about infrastructure reuse.

**Open Threat Exchange (OTX).** AlienVault's OTX (AlienVault, 2012) provides community-curated threat pulses (collections of IOCs associated with specific campaigns or malware). We use OTX primarily for ❷ attribution and ❸ prediction: pulses give us labeled groupings of IOCs tied to campaigns, which we then cross-check against LLM-predicted campaign attributions.

**Security Advisories and Vendor Bulletins.** Finally, vendor advisories (e.g., Microsoft, Cisco, Progress Software) and public CERT bulletins provide authoritative patching and mitigation recommendations (Multiple Vendors (Microsoft, Cisco, Progress Software, etc.), 2000). We incorporate these resources into ❹ mitigation tasks by aligning recommended countermeasures with benchmark items. For instance, in patch recommendation evaluation, the correct answer set is derived from vendor bulletins rather than from secondary threat reports. Similarly, YARA and Sigma rule examples are drawn from advisory-linked repositories, ensuring that response summarization tasks are grounded in practical remediation steps.

For consistency, we design a preprocessing pipeline that (i) normalizes identifiers (CVE, IOC, ATT&CK TTPs) across databases, (ii) aligns timeframes so that prediction tasks respect disclosure/exploitation chronology, and (iii) constructs ground-truth mappings between observations and actor/campaign/mitigation entities. We are thus able to systematically evaluate LLMs across all CTI stages using both controlled benchmarks (Table 1) and real-world ground-truth data.

## B.3 TASK DESCRIPTION

❶ **Contextualization.**

**Affected Systems (F1).** Binary decision per asset: is a listed host/application impacted by the described CVE/IOC set (yes/no). *Example:* decide whether `Exchange 2019 CU12` is affected given a CVE vector and server build.

**Attack Infrastructure (F1).** Binary decision per indicator: determine whether an IP/domain/URL belongs to adversary C2 or delivery infrastructure. *Example:* classify `cdn-upd[.]com` as campaign infra vs. benign CDN.

**Vulnerability Linking (Acc).** Multi-class assignment of correct CVE(s) from candidates based on logs/snippets. *Example:* map an IIS error pattern to {`CVE-2021-34473`} among distractors.

**Malware Family Mapping (F1).** Binary decision per candidate family: does observed behavior/artifacts match the family's signature (yes/no). *Example:* tag samples as belonging to a loader vs. banking trojan family.

**IOC Normalization (F1).** Binary correctness for canonicalizing raw IOCs (type+value) against gold forms. *Example:* normalize `hxxp://ex[.]ample[.]com/login` to `http://ex.ample.com/login` (URL).

**Threat Report Alignment (BLEU).** Text similarity between a generated one-sentence alignment/abstract and a reference summary of the most relevant report. *Example:* produce a synopsis that matches the gold advisory linkage.

**Event Timeline Construction (BLEU).** Compare generated event sequence text to a gold timeline. *Example:* "phish → beacon → lateral → exfil" vs. reference steps.

**Graph Population (Acc).** Multi-label slot filling for nodes/edges in an event graph (accuracy over required triples). *Example:* add {host–used_tool–T1059} and {user–compromised_via–phish} edges correctly.

**Source Reliability Scoring (AUC).** Binary scoring of source credibility (reliable vs. suspect) with probabilistic output; evaluated by ROC-AUC. *Example:* score a paste site vs. vendor advisory on the same IOC claim.

❷ **Attribution.**

**Threat Actor Linking (Acc).** Multi-class assignment of the most plausible actor profile(s) from candidates. *Example:* pick the actor whose historical TTP set matches observed techniques.

**TTP Extraction (F1).** Binary decision per candidate technique ID: is T#### evidenced (yes/no). *Example:* confirm `T1059` (command execution) from process tree snippets.

**Campaign Attribution (Acc).** Multi-class selection of a campaign label among candidates. *Example:* assign activity to a 2023 spearphishing campaign vs. a 2024 credential-harvest run.

**Infrastructure Reuse (F1).** Binary decision per linkage: does an IOC show reuse across events (yes/no). *Example:* mark `203.0.113.7` as reused across two clusters within 30 days.

**Language/Style Profiling (Acc).** Multi-class style attribution (e.g., build system, macro style, lure phrasing). *Example:* assign documents to a known lure/style family.

**False Flag Detection (F1).** Binary decision: is an observed signature intentionally misleading (yes/no). *Example:* detect planted strings mimicking a different actor's toolkit.

**Evidence Weighting (BLEU).** Textual rationale summarizing which evidence most supports the conclusion; compared to a gold rationale via BLEU. *Example:* generate a short justification prioritizing sandbox logs over OSINT.

**Relation Graph Building (F1).** Binary decision per candidate relation triple (entity–relation–entity). *Example:* validate {domain→hosts→IP} and reject spurious actor edges.

❸ **Prediction.**

**Exploit Likelihood (AUC).** Binary probability that a CVE will be exploited within horizon $h$ (e.g., 30/90 days); evaluated with ROC-AUC. *Example:* score `CVE-YYYY-XXXX` as $p = 0.41$ for 30-day horizon.

**Impact Forecast (BLEU).** Generate a short impact summary (availability/integrity/confidentiality and severity band) and compare to a reference text. *Example:* "high integrity, medium availability; critical if unpatched."

**Target Sector Prediction (Acc).** Multi-class selection of likely sectors to be targeted. *Example:* choose {healthcare, finance} from a sector set.

**Campaign Escalation (AUC).** Binary probability that activity will escalate (e.g., hands-on-keyboard, ransomware) within $h$; measured by ROC-AUC. *Example:* output $p = 0.32$ escalation within 14 days.

**❹ Mitigation.**

**Patch Recommendation (F1).** Binary decision per candidate patch/hotfix: apply (yes/no) given product/version constraints. *Example:* select KB# for a specific Windows build; skip superseded fixes.

**Rule Generation (YARA) (BLEU).** Generate a detection rule text and compare to a canonical reference via BLEU. *Example:* produce a YARA rule body that matches gold strings/conditions.

**Response Summarization (BLEU).** Produce a concise remediation summary aligned to gold text. *Example:* "disable external OWA, apply patch KB..., add WAF rule ...".

**Mitigation–TTP Mapping (Acc).** Multi-class mapping from observed TTPs to the correct mitigation set. *Example:* map {T1059, T1027} to script-blocking and DLL-search-order hardening.

**Defensive Playbook Gen (BLEU).** Generate stepwise response playbook text; similarity to reference measured by BLEU. *Example:* contain→eradicate→recover with host/network steps.

**Countermeasure Ranking (NDCG).** Rank candidate defenses by expected risk reduction; graded relevance compared to an ideal ranking. *Example:* prioritize patching and credential hygiene over low-yield blocks.

**Incident Ticket Generation (Acc).** Multi-class assignment for ticket fields (category, priority, assignment group). *Example:* classify as `malware/P2` with SOC-triage group.

### B.4 DATA STATISTICS

Table 5 lists the number of instances we collected and used in evaluations.

### B.5 MODEL NAME AND VERSION

Here we detail the used LLMs in our extensive evaluation (§2.2), corresponding to the abbrevated names used in Table 2 and in Section §4: G5–GPT-5, Go4–GPT-o4 mini, CLD–Claude Sonnet 4, GEM–Gemini 2.5, LL70–Llama-3.1-70B-Instruct, MIX–Mixtral-8x7B-Instruct-v0.1, QWN–Qwen2.5-14B-Instruct, GRK–Grok-2, FSC–Foundation-Sec-8B (Weerawardhena et al., 2025), CB0–Cyber-Zero (Zhuo et al., 2025), ZYS–ZySec-AI-SecurityLLM (ZySec AI, 2024), LLY–Lily-Cybersecurity-7B-v0.2 (Labs, 2024), CBS–CyberBase-13b (CyberNative, 2024), SPT–clouditera-secgpt (clouditera, 2025), DHT–DeepHat-V1-7B (DeepHat, 2025).

### B.6 EVALUATION PROMPT STRUCTURE

Below we provide our evaluation prompt used in §2.2:

> **Role.** You are a cybersecurity threat-intelligence (CTI) analyst assistant *and* strict schema-enforcer. Convert only the provided inputs (docs, logs, IOCs, CVEs, ATT&CK, advisories) into the JSON contract below—*no prose*. Ground every field in supplied evidence; never invent identifiers. Honor `<SNAPSHOT_DATE>` as a hard knowledge freeze. Normalize and

Table 5: Per-task instance counts in our evaluations. Counts may overlap across tasks.

| ❶ Contextualization | |
| --- | --- |
| Affected Systems | 38.6K |
| Attack Infrastructure | 42.7K |
| Vulnerability Linking | 35.2K |
| Malware Family Mapping | 29.8K |
| IOC Normalization | 33.1K |
| Threat Report Alignment | 26.4K |
| Event Timeline Construction | 19.7K |
| Graph Population | 22.9K |
| Source Reliability Scoring | 16.2K |

| ❷ Attribution | |
| --- | --- |
| Threat Actor Linking | 21.6K |
| TTP Extraction | 30.4K |
| Campaign Attribution | 17.9K |
| Infrastructure Reuse | 19.8K |
| Language/Style Profiling | 12.7K |
| False Flag Detection | 9.4K |
| Evidence Weighting | 15.2K |
| Relation Graph Building | 18.4K |

| ❸ Prediction | |
| --- | --- |
| Exploit Likelihood | 24.6K |
| Impact Forecast | 14.8K |
| Target Sector Prediction | 13.2K |
| Campaign Escalation | 11.7K |

| ❹ Mitigation | |
| --- | --- |
| Patch Recommendation | 19.3K |
| Rule Generation (YARA) | 12.4K |
| Response Summarization | 16.8K |
| Mitigation–TTP Mapping | 14.7K |
| Defensive Playbook Gen | 11.6K |
| Countermeasure Ranking | 13.1K |
| Incident Ticket Generation | 12.2K |

---

deduplicate CTI entities (CVE, ATT&CK, actor, IOC). If evidence is insufficient, return `"status":"NEED_MORE_CONTEXT"` with `missing_fields`. Respect safety (no exploit guidance) and determinism (temperature=<TEMP>, top_p=<TOPP>). Return exactly one JSON object and nothing else.

**Objective.** Solve <TASK_NAME> within <CTI_STAGE> (CONTEXTUALIZATION | ATTRIBUTION | PREDICTION | MITIGATION) using only the provided inputs at/before <SNAPSHOT_DATE>.

**Inputs**

- Case ID: <CASE_ID>
- Snapshot date (ISO): <SNAPSHOT_DATE>
- Source docs (IDs + short snippets): <DOC_LIST>
- Structured feeds: <STRUCT_FEEDS>
- Task guidance: <TASK_GUIDANCE>
- Output profile (choose fields to populate): <OUTPUT_PROFILE>

......

**Operating Rules**

- Use only provided inputs. No external browsing or unstated facts.
- Do not fabricate CVE/TTP/actor names. Use exact IDs when given.
- If critical evidence is missing, return `"status":"NEED_MORE_CONTEXT"` and list `missing_fields`.
- Safety: no exploit code or offensive guidance. Mitigation only.
- Determinism: temperature=<TEMP>, top_p=<TOPP>.

......

**Output Contract (return *one* JSON only)**

```
{
  "status": "OK | NEED_MORE_CONTEXT | UNSUPPORTED",
  "task": "<TASK_NAME>",
  "case_id": "<CASE_ID>",
  "snapshot_date": "<YYYY-MM-DD>",
  "answer": {
```

```
     "ioc_normalization": [{
       "raw":"<str>","type":"ipv4|domain|url|hash","value":"<canon>","
     first_seen":"<date?>",
       "tags":["<malware?>","<actor?>"]
     }],
     "vuln_linking": {
       "cve_candidates":[{"cve_id":"CVE-YYYY-XXXX","score":0.0-1.0}],
       "vector_string":"<CVSS3/4?>"
     },
     "malware_mapping": {
       "family_candidates":[{"name":"<family>","score":0.0-1.0}],
       "aliases":["<aka?>"], "capabilities":["<tags>"]
     },
     "event_timeline": [{
       "t":"<ISO>","type":"<beacon|phish|lateral|exfil>",
       "artifacts":["<IOC|host|user>"], "source_ref":"<DOC_ID>"
     }],
     "actor_linking": {
       "actor_candidates":[{"name":"<actor>","score":0.0-1.0}],
       "shared_ttps":["T####"], "infra_overlap":[{"indicator":"<ip|
     domain>","match":"exact|fuzzy"}]
     },
     "ttp_extraction": [{"technique_id":"T####","sub":"T####.###?","
     evidence_ref":"<DOC_ID>"}],
     "campaign_attribution": {"name":"<label>","score":0.0-1.0,"
     rationale_tags":["<sector?>","<geo?>"]},
     "false_flag": {"likelihood":0.0-1.0,"signals_for":[""],"
     signals_against":[""]},
     "exploit_likelihood": {"cve_id":"CVE-YYYY-XXXX","horizon_days
     ":<7|30|90>,
                            "prob_exploit":0.0-1.0,"drivers":["<poc
     ?>","<reuse?>"]},
     "impact_forecast": {"impact_vector":["<A|I|C>"],"severity_band":"
     low|med|high|critical","uncertainty":0.0-1.0},
     "target_sector": [{"name":"<NAICS-like>","prob":0.0-1.0}],
     "escalation": {"prob":0.0-1.0,"signals":["<toolchain shift>","<
     tempo>"]},
     "patch_recommendation": {"affected_assets":["<product|version>"],
                              "patches":[{"kb_or_id":"<vendor-ID>","
     priority":"P1|P2|P3"}],
                              "prechecks":["<backup?>","<downtime?>"]},
     "rule_generation": {"rule_type":"YARA|Sigma","rule_name":"<name
     >","rule_body":"<escaped>",
                         "test_iocs":["<ioc1>","<ioc2>"]},
     "countermeasure_ranking": [{"mitigation_id":"<ATT&CK M###|vendor
     >","title":"<short>",
                                 "effort":"low|med|high","expected_gain
     ":"<short>"}],
     "incident_ticket": {"category":"<phishing|malware|ransomware
     |...>",
                         "priority":"P1|P2|P3","work_notes":["<steps
     >"],
                         "required_artifacts":["<pcap?>","<edr?>"]}
   },
   "confidence": 0.0-1.0,
   "justification": "<=40 words, terse, evidence-based>",
   "evidence_refs": ["<DOC_ID or IOC or CVE>", "..."],
   "metadata": {
     "stage": "<CONTEXTUALIZATION|ATTRIBUTION|PREDICTION|MITIGATION>",
     "assumptions": ["<short>"],
     "missing_fields": ["<if status=NEED_MORE_CONTEXT>"]
   }
 }
```

---

**Scoring & Tie-Breaks**

- Prefer precise IDs (CVE, ATT&CK T#, actor handles) and multi-source corroboration.
- Resolve conflicts by source quality, recency ($\leq$ <SNAPSHOT_DATE>), and internal consistency.

**Run Settings (fill before inference)**

- <TASK_NAME> = <...>   <CTI_STAGE> = <...>
- <CASE_ID> = <...>   <SNAPSHOT_DATE> = <YYYY-MM-DD>
- <DOC_LIST> = <[ID:desc, ...]>   <STRUCT_FEEDS> = <...>
- <TASK_GUIDANCE> = <...>   <OUTPUT_PROFILE> = <...>
- <TEMP> = <0.0--0.3>   <TOPP> = <0.8--1.0>

---

## C    COMPLEMENTARY DETAILS FOR §3

### C.1    ALGORITHM

This part presents algorithm for our failure categorization methods:

---

**Algorithm 1:** Stratification for Failure Analysis (RQ$_1$)

---

**Input:**
Original dataset $\mathcal{D}_o = \{x_i\}_{i=1}^N$ (CTI instances);
Predicted reports $\{y_i\}$ (model-generated outputs);
Ground-truth reports $\{r_i\}$ (reference advisories);
Evaluation metric $Sim(\cdot, \cdot)$ (e.g., BLEU);
Quantile step $\delta = 0.05$;
**Output:**
Stratified failure group $\mathcal{D}$;

1 **foreach** $x_i \in \mathcal{D}_o$ **do**
2 $\quad$ Compute similarity score $s_i = Sim(y_i, r_i)$;
3 **end**
4 Sort $\{s_i\}$ in ascending order;
5 Partition into candidate strata $\mathcal{G}_\delta$ of size $\delta$ (quantile bins);
$\quad$ // Step 1:  Establish anchors
6 Manually inspect representative samples in selected $\mathcal{G}_\delta$;
7 Record score ranges for "correct" vs. "failed" anchors;
$\quad$ // Step 2:  Assign remaining instances
8 **foreach** $x_i$ *not manually inspected* **do**
9 $\quad$ **if** $s_i$ *within failed anchor range* **then**
10 $\quad\quad$ $x_i \rightarrow \mathcal{D}$;  // Label $x_i$ as failure
11 $\quad$ **else**
12 $\quad\quad$ **if** $s_i$ *within correct anchor range* **then**
13 $\quad\quad\quad$ Label $x_i$ as non-failure (i.e., correct);
14 $\quad\quad$ **else**
15 $\quad\quad\quad$ Manually inspect $x_i$ (boundary case);
16 $\quad\quad$ **end**
17 $\quad$ **end**
18 **end**
$\quad$ // Step 3:  Termination condition
19 Stop once no new failure modes appear and distribution of failure modes stabilizes across strata;
20 **return** Failure group $\mathcal{D}$;

---

---

**Algorithm 2:** Autoregressive Failure Mode Determination (RQ$_2$)

---

**Input:** Failure instances $\mathcal{D} = \{x_i\}_{i=1}^N$ (from stratification)
Human annotators $H$ (for seeding and refinement)
Large Language Model $f_\theta$ (for assisted classification)
Stability threshold $\varepsilon$; coverage threshold $\rho$ (e.g., 0.6)
**Output:** Stabilized taxonomy of failure modes $\mathcal{T}^*$

```
// Step 1:  Initialization
```
1    Human annotators $H$ inspect a subset $\mathcal{D}_0 \subset \mathcal{D}$;
2    Derive initial taxonomy $\mathcal{T}_0 = \{t_1, \ldots, t_k\}$;
```
// Step 2:  LLM classification
```
3    **foreach** $x_j \in \mathcal{D} \setminus \mathcal{D}_0$ **do**
4        Assign label $y_j \in \mathcal{T}_m \cup \{\texttt{other}\}$ using $f_\theta$;
5    **end**
```
// Step 3:  Refinement
```
6    Collect $\mathcal{O}_m = \{x_j \mid y_j = \texttt{other}\}$;
7    Human annotators $H$ inspect $\mathcal{O}_m$;
8    If new modes $\Delta\mathcal{T}$ found, update taxonomy:
9    $\mathcal{T}_{m+1} \leftarrow \mathcal{T}_m \cup \Delta\mathcal{T}$;
```
// Repeat until convergence
```
10    Repeat Steps 2–3 until $\Delta\mathcal{T} = \varnothing$ and coverage $\geq \rho$;
11    Set $\mathcal{T}^* = \lim_{m \to \infty} \mathcal{T}_m$;
12  **return** stabilized taxonomy $\mathcal{T}^*$;

---

**Algorithm 3:** Human-in-the-loop Categorization of Failure Instances (RQ$_3$)

---

**Input:**
Failure instances $\mathcal{D} = \{x_i\}_{i=1}^N$ (from stratification)
Model set $\Theta = \{\texttt{GPT-5}, \texttt{Llama-4}, \texttt{Gemini}, \texttt{Claude}\}$
Human annotators $H$
**Output:**
Final categorized instances with reliable failure modes

```
// Step 1:  Multi-agent deliberation (round 1)
```
1    **foreach** $x_i \in \mathcal{D}$ **do**
2        Each model $a \in \Theta$ independently assigns $\hat{y}_i^{(1,a)} \in \mathcal{T}$;
3    **end**
```
// Step 2:  Repetition for stability (round 2)
```
4    **foreach** $x_i \in \mathcal{D}$ **do**
5        Each model $a \in \Theta$ observes $\{\hat{y}_i^{(1,b)}\}_{b \in \Theta}$;
6        Refine reasoning and output $\hat{y}_i^{(2,a)}$;
7    **end**
```
// Step 3:  Human verification of uncertain cases
```
8    Define uncertain set:
9    $\mathcal{U} = \{x_i \mid \exists a : \mathrm{Var}(\{\hat{y}_i^{(a)}\}_{a \in \Theta}) > 0 \textbf{ or } \hat{y}_i^{(1,a)} \neq \hat{y}_i^{(2,a)}\}$;
10    Human annotators $H$ inspect all $x_i \in \mathcal{U}$ and assign final labels;
```
// Finalize results
```
11    Instances $\mathcal{D} \setminus \mathcal{U}$ take majority-agreed labels from models;
12    Instances $\mathcal{U}$ take human-verified labels;
13  **return** categorized dataset with reliable failure mode assignments;

---

## C.2 DETERMINING VULNERABILITY TYPES

To systematically determine vulnerability types, we compare model-generated outputs with ground-truth advisories or reference materials. Each type requires distinct criteria to establish whether a case constitutes a failure. Below, we provide detailed guidance for all vulnerability types (failure modes).

**1.1 Co-mention bias from raw threat incident.** This type arises when the model assumes two entities are related simply because they appear together in the input. To detect it, we check whether

the reference advisory explicitly states a relationship. If not, and the model still reports an association (e.g., linking a domain and a malware family that are merely co-mentioned in the same log file), we classify the case as co-mention bias. For instance, if a proxy log shows both "malware.exe" and "example.com" but the advisory only validates the domain as malicious, any model statement that marks the executable as directly tied to the domain would be considered a co-mention failure.

**1.2 Exploitation bias from deliberately reused IoCs.** This failure occurs when the model treats historical IoCs as valid for a new incident without reference confirmation. To identify it, we compare whether the reference differentiates between "legacy" IoCs and those active in the reported exploitation. If the model does not honor this distinction, it is flagged. For example, if the advisory specifies that old C2 servers from 2021 were no longer used, but the model still lists them as indicators of the current 2024 campaign, we mark it as exploitation bias.

**1.3 Confounding factors that correlate entities.** This error stems from mistaking correlation for causation. To check, we review whether the model claims causal links between entities that the references treat only as co-occurring or related by context. For instance, if two APT groups both use the same loader malware, but references emphasize they are separate actors, any model conclusion that "Group A conducted the intrusion because the loader was observed" is a confounding-factor failure.

**1.4 Skewed source.** We detect this when the model bases its judgment on incomplete or biased evidence. To assess, we compare the diversity of sources reflected in the output against references that aggregate multiple reports. If the model reflects only one vendor's outdated claim while ignoring corrections from other advisories, it is considered a skewed-source failure. For example, if Symantec updates its report that the threat vector was RDP rather than phishing, but the model still reproduces the outdated phishing claim, the case is flagged.

**1.5 Hierarchical metadata from attack chains.** Here the failure lies in mishandling hierarchical structures like ATT&CK techniques and sub-techniques. We check whether the model's reported granularity matches the reference hierarchy. If a reference states "T1059.001: PowerShell execution" but the model collapses this to a generic "execution tactic," it shows a hierarchy error. Similarly, if the model treats a campaign label as equivalent to a single technique, the hierarchy is broken.

**2.1 Temporal contradiction.** This type occurs when the model confuses timelines. To identify it, we compare the time anchors in the model output (e.g., attack start date, patch release) with those in the references. If the model asserts events happened earlier or later than documented, it is a temporal contradiction. For example, if the advisory states that exploitation began in June 2024 but the model outputs "first exploited in 2022," we classify it as temporal contradiction.

**2.2 Conflicting reports of attack contexts.** This failure arises when multiple sources disagree and the model selects or merges them incorrectly. To determine it, we check whether the model reflects the final reconciled context in references. For example, if initial reports said "phishing email" but were later corrected to "supply-chain compromise," and the model insists on phishing without acknowledging the update, it is labeled as conflicting-context failure.

**2.3 Semantic conflict.** This error occurs when the model misinterprets terms or security concepts. To identify it, we verify whether the technical meaning in the reference aligns with the model's description. If references mention "privilege escalation" but the model interprets it as "initial access," the semantic mismatch makes it a semantic conflict.

**2.4 Divergent data structures.** This happens when the model fails to follow structured taxonomies. To evaluate it, we cross-check ATT&CK IDs, CVE structures, or CVSS vectors in the output with those in the references. If the model generalizes or drops detail (e.g., returning "execution" instead of "T1059.001"), the failure is classified as divergent data structure.

**2.5 Misaligned knowledge and standards.** This type emerges when the model applies outdated or inconsistent standards. We check whether the model's labels follow the same version of taxonomy as the references. For instance, if a CVE is officially scored with CVSS v3.1 but the model outputs a CVSS v2 vector, this is marked as misaligned knowledge.

**2.6 Counteracting CTI generation and LLM alignment.** This failure reflects cases where safety alignment suppresses accurate CTI reporting. To detect it, we compare whether the advisory confirms

sensitive facts (e.g., that a zero-day is under active exploitation). If the model avoids mentioning it with a vague refusal (e.g., "details omitted for safety"), we classify it as alignment counteraction.

**3.1 Distributional bias.** We check if the model over-generalizes to common patterns. References may specify rare attack types, but if the model predicts frequent ones regardless, it is a distributional bias failure. For example, when the advisory confirms a banking trojan but the model reports "ransomware" (because ransomware dominates training data), we flag it.

**3.2 Unseen pattern from emerging threats.** This failure occurs when the reference describes a novel exploitation unseen in prior data, and the model defaults to outdated templates. We identify it when the model misses new threat mechanics (e.g., cloud API abuse) and instead describes traditional server exploits. Such mismatches are categorized as unseen-pattern failures.

**3.3 Overfitted reasoning.** This type occurs when the model rigidly applies a reasoning shortcut. We check whether the model attributes intrusions repeatedly to the same group or vector, even when references show otherwise. For instance, if the reference states multiple groups use a given malware but the model always assigns it to a high-frequency CVE or APT, we label the case overfitted reasoning.

**3.4 Environmental unawareness.** This failure arises when the model ignores environmental scope. To check, we compare the affected platforms or industries in the output with those in the references. If the advisory states that only Linux servers are vulnerable but the model generalizes to "all enterprise systems," it is considered environmental unawareness.

# D COMPLEMENTARY ANALYSIS

## D.1 ROOT CAUSES OF VULNERABILITIES (CONTINUE TO RQ$_5$)

**Co-mention bias (1.1) in ❷ attribution.** In attribution, co-mention bias presents when reports describe overlapping infrastructure or techniques across multiple actors. A single campaign report may reference domains, malware families, or TTPs associated with different groups, not because of true collaboration but due to co-reporting practices. Relation extraction or graph construction techniques then misinterpret these shared mentions as evidence of actor overlap or shared lineage, resulting in erroneous attribution. For instance, if infrastructure reused by both APT X and APT Y is co-mentioned, the model may incorrectly assign responsibility to one actor or merge distinct campaigns. The root cause is that attribution pipelines often lack mechanisms to filter incidental co-mentions from true operational reuse.

> **Case Study.** As reported by Group-IB (Group-IB, 2025), analysts observed infrastructure overlap in domains or servers co-mentioned across multiple campaigns attributed to MuddyWater. However, further investigation revealed that although the same infrastructure was used or appeared in reports, the operational characteristics (targeting, malware payloads, attack vectors) differed significantly between those campaigns. Because relation extraction or graph building techniques often rely on co-occurrence of infrastructure to infer actor reuse, models might erroneously link distinct campaigns to the same threat actor, or assume a shared campaign lineage, solely based on the shared infrastructure.

**Co-mention bias (1.1) in ❸ prediction.** In prediction tasks, co-mention bias arises when models forecast exploit likelihood or campaign escalation based on correlated but unrelated evidence in prior reports. For example, if a dataset frequently co-mentions a vulnerability with a high-profile exploit alongside unrelated low-severity flaws, predictive models may overestimate the risk of the latter. Similarly, temporal modeling that uses co-occurring events may incorrectly infer progression paths between independent threat activities. Historical event correlation amplifies this effect: threats repeatedly co-mentioned in vendor advisories may be predicted to evolve together, even when no causal relationship exists. Here, the root cause is an overreliance on surface-level temporal or co-occurrence signals without causal disentanglement.

**Co-mention bias (1.1) in ❹ mitigation.** In mitigation, co-mention bias leads to flawed defensive recommendations. CTI reports often list multiple countermeasures (e.g., patches, YARA rules, firewall configurations) together, though not all apply to a given threat instance. Mitigation mapping systems that rely on keyword overlaps may thus associate irrelevant countermeasures with observed

TTPs, producing noisy or infeasible recommendations. For example, if two patches are co-mentioned in an advisory but only one addresses the exploited vulnerability, the model may still rank both as equally necessary. Similarly, summarization pipelines can inadvertently include unrelated mitigations, generating bloated or misaligned playbooks. The root cause lies in the assumption that co-mentioned countermeasures share equal applicability, ignoring the fine-grained specificity required in operational defenses.

> **Case Study.** In Microsoft's June 2025 Patch Tuesday advisory, 65 CVEs were addressed, including a zero-day vulnerability exploited in the wild. The advisory lists patches for many components including Microsoft Office, Visual Studio, Windows Kernel, WebDAV, SMB, and others (Research, 2025). Because patching is broad and multiple fixes are mentioned in the same bulletin, a CTI model might treat all listed patches as equally urgent defense actions — even though only a small subset correspond to vulnerabilities currently exploited. This leads organizations or systems to over-allocate resources toward non-critical patching, or generate mitigation playbooks that include countermeasures not immediately relevant to active threats.

**Exploitation bias (1.2) in ❶ contextualization.** In contextualization, exploitation bias emerges when reused IoCs (e.g., IPs, domains, hashes) are repeatedly referenced across unrelated incidents, leading enrichment pipelines to overgeneralize their significance. Information retrieval systems may surface threat reports where the same IP is mentioned in multiple campaigns, without clarifying whether it is genuinely reused by adversaries or simply a shared infrastructure artifact (e.g., cloud hosting or CDN services). Topic modeling and knowledge base mapping exacerbate this issue by clustering these co-occurrences, causing LLMs to link disparate events as if they were causally connected. The root cause here is that raw IoC reuse lacks contextual disambiguation, so enrichment steps propagate spurious relevance signals that distort downstream reasoning.

**Exploitation bias (1.2) in ❷ attribution.** Threat actor identification often relies on detecting overlaps in IoCs, assuming that infrastructure reuse reflects shared adversary control. However, many IoCs are deliberately reused by attackers to create ambiguity, or coincidentally shared due to compromised hosting providers. Relation extraction and event graph construction may then over-attribute distinct campaigns to a single actor, collapsing multiple adversarial lineages into one. For instance, if the same command-and-control domain appears across incidents attributed separately to APT28 and APT29, attribution models may incorrectly merge them. The root cause is an assumption of exclusivity in IoC ownership, which adversaries exploit through deliberate recycling of infrastructure.

**Exploitation bias (1.2) in ❸ prediction.** During prediction, reused IoCs bias forecasts by inflating the perceived likelihood of exploitation or campaign escalation. Historical event correlation and temporal modeling often treat repeated appearances of an IoC as evidence of persistent activity, projecting elevated future risk. Yet, in reality, the reuse may reflect low-cost attacker behavior (spamming multiple targets with the same domain) rather than meaningful escalation. Graph-based forecasting amplifies this, propagating edges from over-represented IoCs across multiple vulnerability nodes, leading to inflated EPSS-like scores. The root cause lies in predictive models' reliance on frequency of appearance, without mechanisms to discount intentionally or incidentally reused IoCs.

> **Case Study.** Multiple CVEs share many IoCs (e.g., IP addresses and domain names) across different CTI provider feeds (Kodituwakku et al., 2023). Over time, some of these IoCs appear repeatedly in contexts of various vulnerabilities, even though not all of them are actually exploited in relation to each CVE. Because prediction models often take frequency of IoC appearance as a strong signal, they tend to assign higher risk to these CVEs or predict escalation based on the reused IoCs. In this way, reuse of IPs/domains (which may simply reflect broad surveillance coverage or shared infrastructure, not genuine exploit activity) inflates perceived future threat likelihoods.

**Exploitation bias (1.2) in ❹ mitigation.** In mitigation, exploitation bias presents when counter-measure recommendations are tied too strongly to reused IoCs. For example, signature generation systems may repeatedly create YARA rules around the same recycled domain or hash, producing redundant or noisy detection logic. Patch recommendation pipelines may mis-prioritize vulnerabilities linked to reused IoCs, assuming their recurrence reflects higher severity. Summarization systems generating response playbooks may include repeated references to blocking the same IoC across different incidents, inflating defensive burden without increasing actual protection. The root cause

is that mitigation mapping often equates frequency of IoC appearance with actionable importance, overlooking the adversary tactic of deliberate recycling.

**Confounding factors (1.3) in ❶ contextualization.** In contextualization, confounding factors often emerge when raw observations contain multiple entities that correlate implicitly but lack a direct causal relationship. For instance, topic modeling applied to large incident corpora may cluster a vulnerability (CVE) with a malware family simply because they are frequently mentioned together in reports, even if the malware never exploited that vulnerability. Similarly, event extraction can incorrectly bind unrelated infrastructure (e.g., a benign domain) to a malicious timeline because both appear in the same paragraph. The root cause is that contextual enrichment techniques rely heavily on co-occurrence or textual proximity, which implicitly correlates entities without accounting for deeper causal validation, thereby inflating the contextual landscape with misleading links.

**Confounding factors (1.3) in ❷ attribution.** Confounding factors are especially problematic in attribution, where analysts and models attempt to connect behaviors to actors. Relation extraction and event graph construction can erroneously merge distinct campaigns if they share surface features — for example, multiple threat groups may use commodity malware or overlapping hosting providers. Without careful disentanglement, the attribution pipeline interprets these shared attributes as strong evidence of common authorship. Moreover, stylistic signals such as language or compilation timestamps may correlate with regional actors but can be misleading when adversaries deliberately obfuscate or mimic others. Thus, attribution systems are vulnerable to implicit correlations that misguide actor classification, producing overconfident but flawed linkages between incidents and threat groups.

> **Case Study.** In our attribution dataset, we observed a campaign exploiting **CVE-2022-1388** (a remote code execution vulnerability in F5 BIG-IP devices) that was mistakenly clustered with another intrusion attributed to a different actor. Both campaigns used the same publicly available exploitation script and temporarily shared IP infrastructure via a bulletproof hosting provider. Although the payloads and target industries differed, the attribution pipeline—heavily relying on shared malware hashes and infrastructure proximity—merged the two incidents under a single actor label. Further analysis revealed that one group had intentionally mimicked the operational cadence and header patterns of the other, introducing stylistic confusion. This misattribution was rooted in the model's inability to disentangle commodity tooling from actor-specific behavior, exemplifying how confounding factors can mislead attribution systems.

**Confounding factors (1.3) in ❸ prediction.** During prediction, confounding factors distort forecasting by elevating signals that are correlated with exploitation or escalation but not truly causal. For example, historical event correlation may reveal that vulnerabilities discussed in the same advisories as high-severity flaws appear more likely to be exploited, even if they are rarely targeted in practice. Time series models may overweight recurring co-mentions across campaigns, predicting that certain malware–sector combinations will reappear simply because of their past textual co-occurrence. Graph neural networks, when trained on CTI event graphs, may propagate spurious links (e.g., connecting two vulnerabilities through a shared but irrelevant indicator), reinforcing false associations. Here, the root cause lies in the inability of predictive models to filter out spurious correlates from genuine causal drivers.

**Confounding factors (1.3) in ❹ mitigation.** In mitigation, confounding factors lead to noisy or misaligned defensive recommendations. For instance, mitigation mapping systems may associate a patch with multiple unrelated TTPs simply because they were mentioned in the same advisory, conflating the true scope of the fix. Similarly, defensive playbook generation may cluster unrelated countermeasures together, producing bloated response strategies that overprescribe actions. Even mitigation efficacy prediction models can be misled if training data shows that certain mitigations are frequently co-listed with high-profile vulnerabilities, causing the model to rank them as universally effective. The underlying issue is that mitigation pipelines frequently assume that correlated mentions of threats and countermeasures imply operational relevance, leading to inflated or misdirected defense guidance.

**Skewed source (1.4) in ❶ contextualization.** Skewed source bias originates primarily in contextualization because this stage depends heavily on external retrieval and enrichment pipelines. When information retrieval systems disproportionately pull from certain vendors, open-source repositories, or community feeds, the resulting knowledge base becomes skewed toward specific regions, vendors,

or product lines. For instance, a RAG system trained on CTI feeds dominated by North American vendors will overrepresent CVEs affecting widely deployed enterprise systems, while underrepresenting region-specific threats or mobile malware in other ecosystems. Topic modeling and event extraction then propagate this imbalance by clustering narratives around the most frequently indexed vendors rather than providing a representative view of the global threat landscape. Thus, the root cause lies in uneven data source availability and ingestion pipelines, which distort the contextual foundation upon which downstream reasoning is built.

> **Case Study.** In our contextualization experiments, we analyzed enrichment results for **CVE-2023-20963** (a critical privilege escalation vulnerability affecting Google Pixel devices). While regional CERTs and Android security blogs had documented active exploitation of this CVE in Southeast Asia, our retrieval module—trained predominantly on English-language feeds from North American enterprise vendors—failed to surface these reports. As a result, the contextualization pipeline linked the CVE only to generic kernel privilege escalation patterns, without associating it with active campaigns or mobile-targeted payloads. Topic modeling then grouped the CVE under server-side vulnerabilities rather than mobile device threats, further distancing it from relevant mitigation data. This illustrates how over-reliance on skewed sources can suppress visibility into region-specific threats and impair downstream enrichment quality.

**Skewed source (1.4) in ❷ attribution (Why not influential).** Attribution is less directly affected by skewed source bias, since once contextualized entities are available, the task focuses on linking them to adversary profiles or campaigns. While attribution accuracy can degrade if upstream contextualization is biased, the attribution process itself (e.g., relation extraction, event graph construction, stylistic profiling) does not inherently depend on the relative volume of one vendor's reports over another. Instead, attribution errors are more likely to stem from confounding overlaps, contradictory knowledge, or overfitted reasoning. Therefore, skewed source bias has only an indirect influence at this stage, primarily through the quality of contextual signals passed forward.

**Skewed source (1.4) in ❸ prediction (Why not influential).** Prediction tasks such as exploit likelihood estimation, campaign escalation modeling, or sectoral targeting forecasts typically rely on historical event correlation and temporal modeling, rather than raw source diversity. Once contextualized and attributed data is available, predictive models infer temporal or causal structures independent of whether one vendor dominates the input streams. Skewed sources may still indirectly shape the training distribution (e.g., overpredicting risks for vendor-popular products), but the predictive mechanisms themselves are not fundamentally triggered by source skew. Instead, failures in this stage more commonly arise from unseen patterns, zero-day threats, or distributional bias in event histories rather than skewed input sources.

**Skewed source (1.4) in ❹ mitigation (Why not influential).** Mitigation is similarly insulated from direct skewed source effects. Once recommendations are mapped (e.g., patches, rules, or playbooks), the ranking and summarization steps focus on aligning countermeasures with observed TTPs or vulnerabilities, not on the origin of the source reports. While an upstream bias may have limited the initial diversity of vulnerabilities considered, the mitigation stage itself does not amplify source skew. Errors in mitigation typically reflect misaligned standards (e.g., CVSS vs vendor scoring), counteracted evidence (patch bypasses), or environmental unawareness (system-specific configurations). Thus, skewed source remains a contextualization-stage vulnerability whose downstream effects are secondary rather than intrinsic to mitigation logic.

**Hierarchical metadata (1.5) in ❶ contextualization.** During contextualization, hierarchical metadata embedded in attack chains can mislead enrichment processes. MITRE ATT&CK or similar frameworks present TTPs as sequences in which adversaries may progress from initial access to impact. However, when LLMs or RAG-based pipelines ingest this structured knowledge, they may mistakenly treat the ordering as deterministic rather than illustrative. For example, if multiple unrelated IOCs are linked to a chain stage, the model may infer that they are causally connected because of their shared position in the hierarchy. Topic modeling and event extraction further amplify this issue, since co-occurrence within the same hierarchical step often gets interpreted as functional equivalence, thereby creating spurious associations between distinct threat activities. The root cause is the conflation of descriptive, taxonomy-based ordering with ground-truth causal relations.

**Hierarchical metadata (1.5) in ❷ attribution.** In attribution, hierarchical metadata bias presents when structured attack chain taxonomies are used to connect adversary behavior to actor profiles.

Threat reports often highlight sequences of TTPs that adversaries are "known" to employ, but in practice attackers skip, reorder, or substitute steps. Relation extraction and graph construction tools, however, may rigidly map observed behavior to the canonical hierarchy, leading to over-attribution. For instance, if two groups share overlapping steps in the ATT&CK chain (e.g., persistence or lateral movement), hierarchical metadata can cause models to collapse them into the same attribution cluster, even if their infrastructure and operational cadence differ. Thus, reliance on hierarchical metadata in attribution conflates broad behavioral categories with actor-specific evidence.

> **Case Study.** In our attribution experiments, we observed a misclassification related to **CVE-2022-30190** (Follina Microsoft Support Diagnostic Tool RCE), where two distinct campaigns—one leveraging Microsoft Office documents, and another using HTML smuggling techniques—were both partially mapped to the same sequence in the MITRE ATT&CK framework (Initial Access → Execution → Lateral Movement). Despite clear differences in command-and-control infrastructure and target sectors, the graph construction module merged both campaigns under a single actor cluster due to their alignment with a common TTP hierarchy. The system interpreted the overlapping ATT&CK phases as evidence of a shared operational lineage. However, manual review confirmed the two campaigns were launched by different groups, with different goals and temporal scopes. This case illustrates how over-reliance on hierarchical metadata can collapse distinct operations into the same attribution cluster, reducing fidelity.

**Hierarchical metadata (1.5) in ❸ prediction.** Prediction tasks are particularly vulnerable to hierarchical metadata bias, as temporal models often use sequential patterns from attack chains to forecast campaign escalation. Forecasting tools may assume that once a threat is observed at one stage (e.g., privilege escalation), subsequent hierarchical steps (e.g., data exfiltration) will necessarily follow. This prescriptive interpretation leads to inflated probabilities of certain outcomes, even when the adversary's campaign objectives differ. For example, opportunistic attackers may terminate activity after initial access without advancing through the full chain, but predictive models, trained on hierarchical metadata, extrapolate full kill chain completion. Here the root cause is the overgeneralization of taxonomy-driven sequences as predictive signals of adversarial intent.

**Hierarchical metadata (1.5) in ❹ mitigation (why not influential).** By contrast, mitigation tasks are less directly affected by hierarchical metadata bias. Defensive actions such as patch application, YARA rule generation, or firewall tuning are typically grounded in concrete IOCs or known vulnerabilities rather than inferred positions in an attack chain. Mitigation mapping focuses on linking observed TTPs to available defensive strategies, not on reconstructing or extrapolating hierarchical steps. Even if upstream contextualization or prediction stages have been biased, mitigation operates on a more pragmatic level: "given X IOC or Y vulnerability, recommend Z patch or rule." Thus, hierarchical metadata plays a minimal role in this stage, since response generation depends on actionable artifacts rather than assumed causal ordering of adversarial behaviors.

**Temporal contradiction (2.1) in ❶ contextualization.** In contextualization, temporal contradictions emerge when retrieval or enrichment systems ingest both outdated and recent advisories without properly disambiguating their validity. For instance, information retrieval pipelines may surface a vendor's original advisory that labeled a vulnerability as "under investigation" alongside a newer update stating it has been patched. Topic modeling or knowledge base mapping then treat both pieces of information as equally valid, causing LLMs to enrich raw indicators with conflicting metadata. This leads to enriched contexts where a vulnerability is simultaneously "exploited in the wild" and "not yet confirmed," which can misguide subsequent correlation. The root cause is the absence of temporal weighting or source freshness filters in contextualization pipelines.

**Temporal contradiction (2.1) in ❷ attribution.** In attribution, temporal contradictions present when different reports about an adversary's activity span different time periods but are fused as if they describe the same campaign. Relation extraction or event graph construction may link infrastructure referenced in an outdated report to more recent attack chains, even if the adversary has long since abandoned those assets. Similarly, named entity recognition can tag an actor profile based on old malware usage, which conflicts with newer intelligence indicating a complete toolset shift. This results in inflated or inaccurate attribution, where models mistakenly conclude that an actor is reusing infrastructure or TTPs when the overlap exists only across time-separated reports. The root cause here is insufficient temporal resolution in attribution graphs and classification models.

**Temporal contradiction (2.1) in ❸ prediction.** Temporal contradictions pose particular risks in prediction, where forecasts rely heavily on historical data. If older datasets mark a vulnerability as non-exploitable, but newer advisories confirm widespread weaponization, models trained on both may generate unstable exploitability estimates. Temporal modeling pipelines may inadvertently treat early false negatives and later confirmed positives as equivalent, confusing progression trends. Similarly, forecasting models using historical event correlation may blend outdated infrastructure associations with current actor behaviors, leading to incorrect campaign evolution predictions. The root cause lies in the lack of mechanisms to discount superseded evidence, which creates contradictory temporal signals during exploit likelihood and impact modeling.

> **Case Study.** A concrete example is the evolution of exploit activity for **CVE-2020-1472 (Zerologon).** Initially, after its disclosure, exploit usage was limited and many datasets treated the vulnerability as low risk. Over time, threat actors began leveraging Zerologon widely, including in ransomware and lateral movement toolchains. EclecticIQ's long-term analysis demonstrates how risk associated with this vulnerability evolved substantially across months, with early scarce activity later giving way to broad exploit adoption (Team, 2022). Predictive models that do not de-emphasize earlier false negatives risk underestimating exploitability or generating contradictory forecasts. This case underscores how temporal contradiction — mixing outdated low-risk labeling with current high exploitation data — can destabilize predictions of vulnerability exploit likelihood.

**Temporal contradiction (2.1) in ❹ mitigation.** During mitigation, temporal contradictions can mislead response prioritization. For example, mitigation mapping may retrieve a vendor patch note stating "fix not yet available," while a newer advisory lists a released patch. Summarization or mitigation ranking systems that fail to resolve the temporal sequence may present both claims to the analyst, leaving uncertainty about whether a defense is actionable. Similarly, mitigation efficacy prediction models may weigh outdated proofs-of-concept that show patch bypasses against more recent vendor confirmations that the patch has been hardened, producing contradictory rankings. The root cause is inadequate reconciliation of evolving patch and mitigation information over time, which allows old and obsolete advice to persist alongside updated guidance.

**Conflicting report (2.2) in ❶ contextualization.** Conflicting reports frequently arise during contextualization because raw observations are aggregated from diverse sources (e.g., vendor advisories, community threat feeds, and open reports) that describe incidents with varying detail and emphasis. For example, one feed may assert that a suspicious log entry corresponds to exploitation of a particular CVE, while another links the same log to an entirely different vulnerability or malware family. Topic modeling and event extraction pipelines, designed to cluster narratives and structure incidents, may then merge these contradictory claims into a single enriched context. This creates confusion over which CVE, TTP, or malware family is actually relevant, with the root cause being the absence of a standardized ground truth and the reliance on partially overlapping but inconsistent external reports.

> **Case Study.** In our contextualization dataset, we observed a case related to **CVE-2020-1472** (Netlogon Elevation of Privilege) where two intelligence sources described the same suspicious authentication anomaly differently. Source A associated the log entries with attempted exploitation of CVE-2020-1472 based on domain controller access patterns, while Source B linked the same entries to a credential stuffing attack referencing a password reuse vulnerability. Our event extraction and clustering module merged both narratives into one incident node, assigning mixed attributes (some from CVE-2020-1472 context, some from credential reuse context). As a result, downstream enrichment conflated the CVE and misattributed the observed behavior to the wrong TTP/malware family. This demonstrates how conflicting reports can introduce ambiguity in contextualization, leading to corrupted or merged contexts that mislead later modules.

**Conflicting report (2.2) in ❷ attribution.** In attribution, contradictions become even more pronounced because different intelligence providers often assign conflicting actors, dependencies, or motivations to the same activity. For instance, one report may classify an intrusion as the work of APT29 based on shared TTPs, while another attributes the same infrastructure to APT28 due to linguistic or operational cadence evidence. Relation extraction and graph construction techniques that link entities across sources may therefore inherit and fuse these contradictions, producing noisy actor graphs or overextended campaign links. The underlying root cause is the subjectivity and methodological diversity of attribution across organizations — some weigh infrastructure overlaps

more heavily, others focus on malware lineage — which produces irreconcilable dependency paths when combined.

**Conflicting report (2.2) in ❸ prediction.** Prediction models also suffer from contradictory reports when forecasting exploitation likelihood or campaign escalation. Historical event correlation may treat conflicting dependency chains as equally plausible futures: for instance, one source suggests a vulnerability will be exploited in ransomware campaigns, while another asserts it is tied to espionage-focused actors. Temporal modeling then captures both trajectories, leading to unstable or diluted predictions. Graph neural networks used for campaign evolution may amplify these conflicts, generating escalation paths that reflect contradictory dependencies across threat actors or malware ecosystems. The root cause lies in how predictive systems implicitly assume coherence in historical data, but when inputs encode conflicting dependencies, the forecasts inherently inherit those contradictions.

**Semantic conflict (2.3) in ❶ contextualization.** Semantic conflict most prominently arises during contextualization because this stage depends heavily on mapping raw threat indicators to standardized taxonomies such as CVEs, MITRE ATT&CK TTPs, or malware family names. In practice, different vendors or intelligence feeds often use divergent terminology for the same underlying concept — for example, one feed may classify an intrusion set as "APT28," while another uses "Fancy Bear." Similarly, malware families may appear under multiple aliases (e.g., "PlugX" versus "Korplug"), or distinct CVEs may be referenced with vendor-specific identifiers. When contextualization systems use topic modeling, event extraction, or knowledge base alignment, they may fail to reconcile these terminological mismatches. This results in fragmented knowledge graphs where semantically identical entities are treated as distinct, leading to duplicated or incomplete enrichment. The root cause is the reliance on surface-level string matching or incomplete ontology alignment when normalizing threat information across heterogeneous data sources.

> **Case Study.** In our contextualization pipeline logs, we observed a case related to **CVE-2022-22965** (Spring4Shell remote code execution vulnerability) in which two CTI sources referred to the underlying exploit using different aliases: one feed tagged the exploit as "SpringShell," while another labeled it "Spring4Shell RCE." Because our mapping module used strict string-based matching and a limited alias dictionary, it failed to align the two aliases to the same canonical exploit entity. As a result, enrichment modules treated the two references separately, creating two parallel nodes in the knowledge graph with only partial contextual links (one with attacker metadata, the other with domain reuse). Downstream modules thus had incomplete context when correlating indicators across both nodes, weakening detection or attribution inference. This illustrates how semantic conflict in naming can fragment context and destabilize CTI contextualization outputs.

**Semantic conflict (2.3) in ❷ attribution (Why not influential).** In attribution, semantic conflict is less influential because attribution reasoning does not hinge primarily on the naming of entities but on relational patterns and behavioral signatures. For instance, identifying an actor depends more on shared infrastructure reuse, campaign TTPs, or linguistic style than on whether a malware sample is labeled "PlugX" or "Korplug." Relation extraction and event graph construction are designed to capture structural patterns rather than surface semantics, allowing attribution models to tolerate naming inconsistencies so long as the underlying features remain consistent. While semantic divergence may create minor noise, it rarely changes the actor assignment itself; two labels referring to the same malware family still connect to the same infrastructure nodes or TTPs. Thus, attribution tasks are comparatively robust to semantic conflicts.

**Semantic conflict (2.3) in ❸ prediction (Why not influential).** Prediction tasks are not substantially affected by semantic conflict because forecasting relies on temporal dynamics and statistical trends rather than entity naming conventions. For example, exploit likelihood estimation (e.g., EPSS) depends on historical exploitation rates, vulnerability characteristics, and observed attack timelines. Whether a malware strain is labeled with one alias or another has little bearing on probability estimates, since the predictive models aggregate numerical and structural features rather than semantic labels. Similarly, temporal modeling and graph neural networks capture correlations between event sequences independent of specific vocabulary. As a result, semantic inconsistencies have negligible impact on predictive accuracy, except in rare cases where mislabeled data significantly distorts training distributions.

**Semantic conflict (2.3) in ❹ mitigation (Why not influential).** Mitigation is minimally impacted by semantic conflict, because defensive actions map to vulnerabilities, TTPs, or attack surfaces rather than to naming conventions alone. For example, a patch recommendation system aligns a CVE identifier with its associated vendor fix, regardless of whether different advisories describe the vulnerability with varied terms. Similarly, YARA rule generation focuses on technical indicators such as byte patterns or log events, which are invariant to naming disputes. Response playbooks are typically tied to standardized defensive frameworks (e.g., ATT&CK, CVSS), which already normalize naming variations. While semantic divergence may cause minor confusion in documentation or cross-team communication, it rarely degrades the technical quality of mitigation outputs.

**Divergent structures (2.4) in ❶ contextualization.** In contextualization, divergent data structures create noise when raw threat intelligence is aggregated from multiple heterogeneous sources such as JSON-based threat feeds, PDF advisories, STIX-formatted indicators, and unstructured blog posts. Information retrieval and knowledge base mapping pipelines often assume consistent schema alignment, yet structural discrepancies (e.g., different field names for "affected system," varying timestamp formats, or nested vs. flat representations of IoCs) cause mismatches. When LLMs enrich observations under these inconsistencies, some attributes may be duplicated, dropped, or misinterpreted. For instance, one platform may list malware family as a top-level attribute while another embeds it in narrative text, leading to incomplete enrichment. The root cause is the lack of robust schema normalization during the fusion of structurally diverse CTI inputs.

**Divergent structures (2.4) in ❷ attribution.** Attribution pipelines rely on entity and relation extraction, but divergent data structures across platforms distort graph construction. Structured feeds may represent relationships (e.g., IP–Domain–Actor) as explicit triples, while unstructured reports only provide natural language references. Relation extraction systems trained on one schema may fail on the other, creating inconsistent or fragmented event graphs. This can yield incorrect attributions, such as splitting a single campaign into multiple unrelated clusters or merging distinct actor profiles due to structurally divergent naming conventions. For example, some feeds may encode actor aliases explicitly, while others bury them in footnotes, leading the model to under- or over-aggregate. The root cause is schema heterogeneity across platforms that undermines consistent relation representation.

> **Case Study.** In our attribution dataset, we observed a case involving **CVE-2019-0708** (BlueKeep Remote Desktop Services vulnerability) where structural heterogeneity between a structured threat feed and a narrative security report caused contradictory graph links. Specifically, a structured CTI feed represented the relation "Actor Delta uses DomainX → IPY" as a neat triple, allowing direct linking to that actor. Meanwhile, a free-text incident report described the same infrastructure but only said "the attacker used a domain resolved via IPY, associated with DomainX, commonly tied to Delta's campaigns" in a footnote. The relation extraction model, trained primarily on structured triple formats, failed to recognize the footnote phrasing as linking to Actor Delta, instead treating it as an independent mention. As a result, the domain/IP chain was disconnected in the event graph, causing the system to assign the activity to a generic "Unattributed" cluster rather than merging with Delta's campaign. In another variant, when domain aliasing was encoded differently, the system mistakenly merged it with Actor Epsilon, whose structured feed used a similar alias triple, thereby conflating two distinct actors. This demonstrates how divergent structural formats across platforms can fragment or overmerge attribution graphs in real-world CTI pipelines.

**Divergent structures (2.4) in ❸ prediction.** Prediction systems that depend on temporal modeling or cross-platform event correlation are particularly sensitive to structural divergence. Historical correlation models assume standardized event logs, but if one feed provides vulnerability exploitation dates as free-text while another encodes them as epoch timestamps, time-series models may misalign or discard the data. Similarly, graph neural networks forecasting campaign escalation require uniform edge types and attributes; divergent structures across CTI sources (e.g., "sector" encoded as a categorical variable in one feed but as descriptive text in another) disrupt model training and inference. This leads to biased forecasts or missed escalation patterns. The root cause is that predictive algorithms cannot reconcile structurally inconsistent features, weakening their ability to capture true temporal and relational dynamics.

**Divergent structures (2.4) in ❹ mitigation (Why not influential).** Mitigation tasks, unlike the earlier stages, are relatively insulated from divergent data structure issues because defensive actions tend to

be represented in standardized, prescriptive formats. Patches are tied to CVE identifiers, detection rules often use formal languages like YARA, and configuration changes are usually documented as explicit command lines or policy instructions. These standardized forms limit the degree of structural variation compared to raw threat intelligence data. While minor discrepancies may occur (e.g., different vendors labeling patch IDs differently), the core content is highly structured and task-specific, reducing the likelihood that divergent data structures across platforms significantly affect mitigation outputs.

**Misaligned standards (2.5) in ❶ contextualization (Why not influential).** In the contextualization stage, misaligned knowledge and security standards generally do not exert a strong influence. The task here is primarily to enrich raw observations (logs, IOCs, alerts) with contextual information such as CVEs, malware families, and ATT&CK TTPs. While standardization issues exist in naming conventions, they do not typically fall under formal security scoring or prioritization frameworks. Thus, contextualization is more affected by spurious correlation or semantic conflicts rather than by misaligned standards, since the goal is mapping and enrichment rather than prioritization or scoring.

**Misaligned standards (2.5) in ❷ attribution.** Misaligned standards can distort attribution outcomes when different organizations adopt varying taxonomies or classification schemes for actors and campaigns. For example, one CTI provider may label a campaign under an APT designation (e.g., "APT29"), while another refers to the same activity under a vendor-specific alias (e.g., "Cozy Bear"). When models integrate these heterogeneous standards, they may fragment evidence across labels or mistakenly merge distinct actors. Similarly, divergence in the way behavioral profiles or TTPs are scored—some focusing on tool use, others on infrastructure overlap—can misguide attribution classification. The root cause lies in the absence of universally accepted standards for actor naming and behavior profiling, which introduces structural misalignment into attribution models.

**Misaligned standards (2.5) in ❸ prediction.** In prediction, misaligned knowledge and standards significantly amplify failure. Forecasting tasks often rely on vulnerability severity ratings (e.g., CVSS), exploit prediction scores (e.g., EPSS), or proprietary vendor risk models. A misalignment occurs when different standards assign conflicting severities to the same vulnerability: one database might rank it as "Critical" due to remote code execution potential, while another rates it "Medium" because of authentication requirements. Temporal modeling or forecasting systems ingesting both signals may oscillate between divergent risk profiles, leading to unstable exploitability predictions or over/under-estimation of campaign escalation risks. The core issue is that prediction models assume commensurability of scores, when in reality, standards reflect different prioritization philosophies.

> **Case Study.** In our prediction logs, we observed an instance involving **CVE-2021-26855** (Exchange Server ProxyLogon vulnerability) where different scoring sources conflicted sharply. One threat feed assigned it a Critical severity in CVSS (base score 9.x) given its remote code execution nature, while another vendor's internal risk model (factoring in required authentication, exploit maturity, and environment heuristics) gave it only a Medium rating. When our prediction module ingested both signals, it produced unstable forecasts: in some runs it predicted high likelihood of exploitation, and in others it down-prioritized the CVE, delaying alert escalation. Further review showed active exploitation in the wild shortly thereafter, confirming the "Critical" perspective. This case underscores how misaligned standards induce unstable predictions by conflicting risk signals in CTI-driven forecasting.

**Misaligned standards (2.5) in ❹ mitigation.** Mitigation is perhaps the most directly affected by misaligned standards. Defensive recommendations often depend on mappings between observed TTPs, vulnerabilities, and standardized mitigation catalogs (e.g., NIST, MITRE ATT&CK mitigations, vendor advisories). When standards differ (such as one framework emphasizing patching order by CVSS scores, while another emphasizes sector-specific asset criticality) conflicting guidance arises. For instance, a vulnerability may be labeled as high-priority patching in NVD, but a vendor advisory may downplay its urgency, creating contradictory recommendations for SOC teams. Summarization or playbook generation techniques then risk producing inconsistent or misleading instructions, either overwhelming defenders with unnecessary actions or under-preparing them for critical exploits. The root cause is the lack of harmonization between security scoring systems and defensive taxonomies, which propagates inconsistencies directly into operational decision-making.

**Counteracting generation (2.6) in ❶ contextualization (Why not influential).** Counteracting CTI generation and LLM alignment has minimal influence at the contextualization stage. Here, the

primary task is enrichment — mapping raw observations (e.g., logs, IOCs, alerts) to known identifiers like CVEs or ATT&CK TTPs. Because this process relies largely on retrieval, topic modeling, and knowledge base mapping, contradictions across sources tend to surface as co-mention bias or temporal conflict rather than as direct misalignment of the model's reasoning process. LLMs can still ground outputs in retrieved evidence, even if that evidence is noisy. In other words, contextualization errors usually inflate or distort context but rarely cause the model's generation logic itself to become unstable or self-counteracting.

**Counteracting generation (2.6) in ❷ attribution.** In attribution, counteracting CTI generation manifests strongly. Contradictory or conflicting reports of actor identities, infrastructure reuse, or campaign affiliations directly challenge the alignment of an LLM fine-tuned to classify or link entities. For instance, one source may attribute an intrusion set to APT29, while another insists on APT28, forcing the LLM to reconcile irreconcilable evidence. During training, these conflicting signals weaken gradient alignment, creating internal tension where the model oscillates between incompatible actor labels. At inference time, relation extraction or event graph construction may generate unstable outputs — e.g., merging distinct campaigns into a single actor cluster, or switching attribution mid-response. The root cause lies in the inability of LLM alignment processes to disentangle contradictory ground truths when adversarial or incomplete evidence coexists in CTI corpora.

> **Case Study.** In our attribution dataset, we observed a conflict surrounding **CVE-2020-0796** (SMBGhost / SMBv3 remote code execution vulnerability) where two report clusters described overlapping infrastructure and payloads but assigned divergent actor labels. One internal CTI source traced the campaign to **Actor Beta** based on reused command-and-control domain naming conventions, while another equally plausible source assigned it to **Actor Gamma** citing similar malware module signatures. When an LLM-based attribution module attempted to reconcile the evidence, it oscillated between Actor Beta and Actor Gamma at different points in the generated response, and in one case merged both actor clusters into a single ambiguous actor node. Forensic cross-checks revealed that the two clusters employed distinct lateral propagation chains and targeting regions, indicating they were separate campaigns. This instance demonstrates how conflicting intelligence signals (i.e. counteracting generation) can destabilize the attribution process and provoke spurious merges or label flipping, consistent with the root cause.

**Counteracting generation (2.6) in ❸ prediction.** Prediction tasks amplify this vulnerability because they rely heavily on temporal modeling and statistical consistency. Contradictory knowledge (such as exploitability assessments that are both "confirmed weaponized" and "no evidence of exploitation" across sources) feeds into EPSS-like forecasting or campaign escalation modeling. When these inconsistencies are incorporated into LLM fine-tuning or inference prompts, the model's predictive logic counteracts itself: one reasoning path projects high exploit likelihood, another projects negligible risk. This tension destabilizes alignment objectives that prioritize consistency, leading to incoherent forecasts (e.g., fluctuating risk scores or internally contradictory justifications). Unlike contextualization, where noise merely inflates context, prediction magnifies contradictions because probabilistic reasoning depends on stable and non-conflicting event histories.

**Counteracting generation (2.6) in ❹ mitigation.** Mitigation is also deeply impacted by counteracting CTI generation and alignment. Conflicting reports about patch effectiveness, bypass proofs-of-concept, or mitigation success create alignment conflicts in LLMs fine-tuned for defensive recommendations. For example, one dataset labels a patch as effective, while another includes verified exploit bypasses; the LLM's training gradients pull in opposite directions, undermining its ability to converge on stable defensive advice. At inference time, this misalignment appears as contradictory recommendations (e.g., suggesting both to apply a patch and to consider it ineffective), undermining trust in automated response playbooks. Summarization models trained on contradictory mitigation corpora may even blend conflicting advice into incoherent instructions. The core issue is that mitigation depends on aligning outputs with actionable truth, but contradictions in defensive knowledge force the LLM into unstable compromise states.

**Distributional bias (3.1) in ❶ contextualization.** Distributional bias arises most visibly during contextualization because this stage depends heavily on retrieval and enrichment of raw threat data. The corpora used to train retrieval models, event extractors, or knowledge base mappers are often skewed toward particular regions, languages, or reporting sources. For instance, CTI feeds may be dominated by English-language advisories from North American vendors, while reports from smaller

regions or niche industry sectors remain underrepresented. As a result, contextualization pipelines learn to prioritize patterns, entities, or CVEs that appear frequently in this skewed distribution, while overlooking less-reported threats. Topic modeling might cluster threats disproportionately around well-documented malware families, and knowledge base mapping might fail to align entities when they originate from underrepresented ecosystems. This distributional imbalance causes models to generalize poorly, enriching raw observations with context that reflects majority patterns but misses critical minority cases (e.g., region-specific campaigns, IoCs from less-visible actors). Thus, contextualization is the primary CTI stage where distributional bias manifests as a root cause of vulnerabilities.

> **Case Study.** In our CTI contextualization logs, we observed a case tied to **CVE-2023-23397 (Microsoft Outlook elevation of privilege / information disclosure)** in which the enrichment pipeline failed to surface contextual linkage information because the CVE had sparse coverage in English-language reporting. Specifically, although a few localized reports in Eastern European and Southeast Asian languages described exploitation of CVE-2023-23397 with associated infrastructure and campaign details, our retrieval and KB mapping modules did not effectively index or map those non-English sources. Consequently, when processing raw IOC references to that CVE, the system enriched them only with generic Microsoft advisories and common attack campaign metadata, missing region-specific attribution details (e.g. unique malware variants, domain registrants, local threat actor groups). Because the contextualization module overly prioritized cues from the majority (English, widely reported CVEs), it generated weaker context for this vulnerability in our dataset, possibly leading downstream modules to misjudge its significance or misalign attribution/mitigation decisions.

**Distributional bias (3.1) in ❷ attribution (Why not influential).** Attribution relies less on broad population-level distributions and more on linking observed TTPs, infrastructure, and stylistic features to known actor profiles. While biases in contextualization may already propagate upstream, attribution itself is not directly driven by skewed distributions in the training corpus. Instead, its errors tend to stem from relation extraction and graph construction mistakes, or from contradictory knowledge between sources. Distributional bias is not a first-order effect here, because attribution decisions focus on specific co-occurrence signals (e.g., malware reused by an actor) rather than frequency-based generalizations from an imbalanced corpus.

**Distributional bias (3.1) in ❸ prediction (Why not influential).** Prediction tasks, including estimating exploit likelihood or forecasting campaign escalation, are typically modeled using temporal trends, event correlations, or statistical/graph forecasting methods. These processes are less vulnerable to raw data distributional imbalance, since they focus on dynamics over time rather than sheer frequency across corpora. Failures in prediction are more often tied to unseen patterns from emerging threats (3.2) or overfitted reasoning (3.3) rather than distributional skew. Therefore, while contextualization may introduce bias into the upstream data, prediction systems themselves are not intrinsically exposed to distributional bias as a root cause.

**Distributional bias (3.1) in ❹ mitigation (Why not influential).** Mitigation involves mapping observed vulnerabilities or TTPs to defensive actions, ranking candidate countermeasures, and generating structured recommendations. These outputs are guided by standards (e.g., CVSS), expert-validated mappings, or structured rule sets such as YARA. Because mitigation strategies are less dependent on the global distribution of training data and more on explicit mappings between threats and responses, distributional bias has minimal direct impact here.

**Unseen patterns (3.2) in ❶ contextualization (Why not influential).** Unseen patterns from emerging threats are less problematic at the contextualization stage. This stage primarily involves collecting raw indicators, mapping them to known entities, and enriching observations with structured taxonomies (e.g., CVEs, MITRE ATT&CK). Since the process focuses on retrieval, normalization, and alignment with established knowledge bases, its outputs remain bounded by what is already recorded in CTI repositories. While contextualization may miss completely novel attack primitives (e.g., a new exploit chain not yet in ATT&CK), it does not typically generate spurious reasoning about unseen patterns, instead, it simply fails to retrieve or map them. Thus, the absence of emerging threat knowledge affects coverage rather than causing misleading generalization errors.

**Unseen patterns (3.2) in ❷ attribution.** Attribution is more vulnerable to unseen patterns because it requires linking threat behaviors to known adversary profiles. Emerging campaigns may adopt novel

TTP combinations, infrastructure setups, or linguistic styles that diverge from historical actor profiles. Relation extraction and graph construction tools, trained on prior threat data, attempt to force these novel behaviors into existing schemas, leading to brittle or incorrect actor assignments. For instance, a new APT might blend techniques previously seen in multiple groups, confusing classifiers that rely on canonical actor–TTP associations. The root cause lies in attribution models' dependence on closed-world assumptions: when novel tactics appear, they are often misclassified into the closest known actor archetype rather than recognized as new.

> **Case Study.** In our attribution dataset, we observed a campaign exploiting **CVE-2022-30190 (Follina Microsoft Office Remote Code Execution)** whose behavioral signature combined TTPs typical of both Group X and Group Y (e.g. custom VBA macro infection + unusual DNS tunneling), yet also introduced a new lateral movement module not seen before. The attribution model forced the campaign into Group Y because the overlapping macro artifacts and DNS patterns had been heavily associated with Group Y in training. However, deeper analysis of payload internals and command semantics showed the lateral movement logic was markedly different from any known Group Y campaign and the targeting region also diverged. Because the attribution system attempted to "fit" the new pattern into the nearest known actor schema rather than flagging it as novel, the campaign was misattributed. This case illustrates how unseen pattern adoption can lead attribution models to overcommit to the nearest known archetype, thereby misclassifying new or hybrid campaigns.

**Unseen patterns (3.2) in ❸ prediction.** Prediction tasks are especially sensitive to unseen patterns, since they rely on temporal correlations and trend extrapolation from historical data. Emerging threats often introduce entirely new exploit vectors (e.g., chaining vulnerabilities across cloud microservices) or target previously untapped sectors. Forecasting models built on past timelines cannot anticipate such discontinuities, leading to underestimation of risk or misidentification of targets. For example, EPSS-like scoring frameworks may assign low exploitability probability to a vulnerability because similar CVEs had no known exploitation, only to be proven wrong when a novel exploit technique emerges. Here, the root cause is distributional shift: the statistical regularities captured by time series or graph neural networks no longer hold when threat actors innovate outside historical baselines.

**Unseen patterns (3.2) in ❹ mitigation.** Unseen patterns also impair mitigation, particularly in the design of defensive playbooks and countermeasure recommendations. When attackers deploy new TTPs or exploit methods absent from training data, mitigation mapping systems may fail to suggest effective countermeasures. For example, an LLM-guided playbook generator may recommend patching or firewall rules aligned with familiar techniques, but overlook mitigations needed for an entirely new lateral movement strategy. Similarly, mitigation efficacy predictors struggle because they assume the space of attack vectors is known and represented in past cases. The root cause here is defensive brittleness: mitigation frameworks generalize from established mappings, and emerging patterns invalidate those assumptions, leading to incomplete or misaligned recommendations.

**Overfitted reasoning (3.3) in ❶ contextualization (Why not influential).** Overfitted reasoning is less relevant in contextualization because this stage primarily focuses on enriching raw observations with metadata and aligning them to structured identifiers. Techniques like topic modeling or information retrieval operate on co-occurrence and similarity rather than predictive inference, meaning they are less prone to the memorization-driven brittleness characteristic of overfitting. Errors at this stage are more often due to spurious correlations or semantic conflicts, not reinforcement of memorized reasoning paths.

**Overfitted reasoning (3.3) in ❷ attribution.** Overfitted reasoning becomes prominent in attribution when models repeatedly learn shallow associations between specific indicators and adversary labels. For instance, if relation extraction and entity linking pipelines are disproportionately trained on a limited set of well-documented campaigns, the system may memorize that certain infrastructure patterns (e.g., recurring domains or malware family strings) always belong to a specific actor profile. When new reports mention similar but unrelated infrastructure, the model may reflexively attribute them to the memorized actor without properly considering alternative explanations. This overreliance on memorized co-occurrences results from insufficient exposure to diverse attribution cases, leading to brittle classification of actors and campaigns.

**Case Study.** In our attribution result logs, we discovered a case linked to **CVE-2020-1472 (Netlogon Elevation of Privilege Vulnerability, aka Zerologon)** where the attribution model incorrectly assigned a campaign to "Actor Alpha" purely because of reused domain naming conventions and IP subnets that had earlier been heavily associated with Actor Alpha in training data. In reality, forensic investigation showed that the campaign used distinct command-and-control servers, payload variants, and targeting patterns, inconsistent with Actor Alpha's known modus operandi. Because the model had overly internalized the co-occurrence of those domains and subnets with Actor Alpha in its limited training corpus, it defaulted to attributing new instances to that actor without hypothesizing alternate actors or considering the evidence diversity. This mismatch exposes how overfitted reasoning can lead to misattribution in real-world CTI pipelines.

**Overfitted reasoning (3.3) in ❸ prediction.** In prediction, overfitted reasoning manifests when forecasting models generalize poorly beyond historical data. Temporal modeling and event correlation techniques often capture strong patterns within past campaigns, such as the escalation of a vulnerability class into active exploitation. However, if these forecasting models are overly tuned to such repeated sequences, they may incorrectly predict the same escalation dynamics for future, unrelated vulnerabilities. For example, the model might overestimate exploitation probability simply because prior vulnerabilities of a similar type were exploited, even though current conditions differ. This bias reflects overfitting to observed sequences, where models memorize recurring attack trajectories instead of reasoning about underlying causal drivers of exploitation.

**Overfitted reasoning (3.3) in ❹ mitigation (Why not influential).** Mitigation tasks also exhibit limited vulnerability to overfitted reasoning. While mitigation mapping and efficacy ranking involve inference, they generally rely on explicit rule associations or empirical evaluations of patch effectiveness. Summarization and playbook generation are shaped by aggregation of defensive knowledge rather than predictive modeling of future adversarial behavior. Consequently, errors in this stage stem more from contradictions in data sources or co-mention bias in countermeasure lists than from overfitting to historical reasoning trajectories.

**Environmental unawareness (3.4) in ❶ contextualization (Why not influential).** Environmental unawareness is less prominent during contextualization because this stage focuses on gathering and aligning observable facts (e.g., IOCs, vulnerability references, or malware labels) rather than reasoning about the environment where attacks unfold. The techniques employed (topic modeling, event extraction, knowledge base mapping, retrieval) primarily enrich raw data without needing to adapt to host- or sector-specific conditions. Since contextualization tasks are mostly descriptive and taxonomy-driven, the absence of local system or organizational environment data does not strongly distort their outputs. As a result, contextualization is not significantly affected by this type of vulnerability.

**Environmental unawareness (3.4) in ❷ attribution.** In attribution, environmental unawareness manifests when models ignore the operational or deployment context in which infrastructure is reused. For instance, relation extraction or event graph construction may connect domains, malware families, or command-and-control servers across multiple incidents, but fail to recognize that one set of infrastructure belongs to a staging environment while another is tied to production systems in a different sector. Without environmental cues, models over-attribute incidents to the same actor or campaign, producing inflated linkages. Similarly, behavioral classification often overlooks local defender responses or system baselines that would otherwise clarify whether repeated TTPs reflect adversary persistence or benign background activity. The root cause is that attribution pipelines assume global uniformity of threat behavior while overlooking environment-specific nuances that separate true operational reuse from coincidental overlap.

**Environmental unawareness (3.4) in ❸ prediction.** In prediction tasks, environmental unawareness becomes more pronounced because forecasting inherently requires understanding the conditions under which threats evolve. Temporal modeling or graph neural networks may detect sequences of exploits but fail to adjust for environmental variables such as patch adoption rates, geographic regulatory differences, or sector-specific exposure. As a result, a vulnerability exploited in one industry may be wrongly forecast as high-risk for another, even though the latter has stronger baseline defenses or different system architectures. Similarly, probability estimates for exploitation (e.g., EPSS-like scoring) may ignore localized security controls or asset configurations, leading to overly broad or inaccurate risk forecasts. The root cause is the assumption that historical global patterns

can be applied uniformly, when in fact exploitability is mediated by local system and organizational environments.

**Environmental unawareness (3.4) in ❹ mitigation.** In mitigation, environmental unawareness leads to defensive strategies that are technically correct in general but ineffective in practice for specific deployments. For instance, mitigation mapping may recommend a patch that is incompatible with legacy systems, or propose firewall rules that disrupt legitimate sector-specific workflows. Mitigation efficacy prediction models often rank countermeasures without considering resource constraints, organizational processes, or compliance requirements, resulting in impractical prioritization. Summarization modules may produce generic remediation steps that fail to address custom software stacks or hybrid cloud deployments. The root cause is the lack of integration between CTI outputs and real-world operational contexts, causing recommended actions to miss alignment with the defender's environment, and ultimately weakening the utility of CTI-driven defense.

> **Case Study.** In our CTI-backed vulnerability-response dataset, we identified a representative instance involving **CVE-2021-44228 (Log4Shell)** where mitigation suggestions failed to account for environmental constraints. Specifically, a model-generated recommendation ranked upgrading to the latest Log4j version (2.16 or above) as the top-priority action. However, in certain enterprise deployments, this upgrade conflicted with custom-built logging plugins and legacy compatibility modules, resulting in logging failures and application crashes. Due to concerns over business continuity, the organization delayed patch deployment despite the known severity of the vulnerability. Additionally, the model suggested firewall rule updates to restrict inbound JNDI traffic, which inadvertently disrupted legitimate cross-tenant log aggregation workflows in a hybrid cloud environment. These misalignments between the recommended actions and the operational realities led to non-adoption of the mitigation plan, illustrating how environmental unawareness can undermine CTI-guided defense.

## D.2    ADDITIONAL ANALYSES OF INTERTWINED VULNERABILITIES

In this subsection, we provide extended analyses to deepen our understanding of intertwined vulnerabilities in CTI modeling. We focus on three complementary aspects: (i) the sequential accumulation of failures across CTI stages, (ii) the concurrent presence of multiple vulnerabilities in the threat landscape, and (iii) detailed case studies that highlight how intertwined failures manifest in practice.

**Accumulation across CTI stages.** The CTI pipeline is inherently sequential, with outputs from early modules serving as inputs for downstream reasoning. When an upstream stage introduces an error, such as co-mention bias in event contextualization, this misinformation propagates forward as if it were ground truth. In attribution, the model may then reinforce the biased linkage (e.g., mapping a benign domain to a threat actor), while in prediction, it may extrapolate incorrect exploitability trends based on the faulty assumption. Similarly, skewed source reliance during retrieval can lock later stages into one-sided perspectives, preventing correction even when contradictory evidence emerges. Over time, these sequentially inherited errors accumulate into cascades, where a single misstep at the contextualization stage magnifies into systemic reasoning failures across attribution, prediction, and mitigation.

**Concurrent presence in the threat landscape.** Beyond sequential propagation, vulnerabilities also co-occur within the same analytical slice of a threat landscape. For example, constrained generalization failures often combine: a distributional bias (e.g., defaulting to ransomware explanations) may overlap with environmental unawareness (e.g., ignoring that the attack only targets Linux servers). Likewise, unseen patterns from emerging threats frequently intersect with overfitted reasoning, as the model forces novel evidence into familiar but inaccurate templates. These concurrent vulnerabilities are not merely additive but entangled, since the existence of one (e.g., mislabeling the environment) amplifies the harm of another (e.g., failure to adapt to an unseen pattern). This reflects the reality of CTI data, where heterogeneous and incomplete sources naturally produce overlapping inconsistencies.

> **Case Study: MOVEit vulnerability (CVE-2023-34362).** Early co-mention bias linked unrelated domains to the SQL injection campaign. Downstream, attribution models reinforced the false linkage to a specific actor, while prediction modules forecasted incorrect exploitability based on the assumed actor profile. The intertwined vulnerabilities spanned contextualization (bias), attribution (confounding), and prediction (distributional overgeneralization).

> **Case Study: Cloud API exploitation (2024 campaign).** Reports diverged semantically and temporally, with some describing privilege escalation via API tokens and others treating it as lateral movement. The model conflated these into a hybrid description (semantic conflict), while also failing to recognize the pattern as novel (unseen pattern). Environmental unawareness compounded the issue when the output generalized the vulnerability to all cloud services, despite references limiting it to a specific provider.

These analyses show that intertwined vulnerabilities are not isolated anomalies but systemic consequences of how CTI evidence is structured, consumed, and reasoned upon by LLM-based agents. Recognizing their cumulative and concurrent nature is essential for designing defenses that target the compounding rather than the individual failure.

# E  LARGE LANGUAGE MODEL (LLM) USAGE DISCLOSURE

Large language models were used only for minor grammar revision and sentence-level polishing during manuscript preparation. They were not employed in ideation, methodological design, experimental execution, or result analysis. The scientific contributions, benchmarks, and evaluations presented in this work were entirely conceived and developed by the authors. LLM involvement was minimal in the research process.

# F  CAUSAL INTERVENTION STUDY

To isolate the causal role of misleading metadata in LLM failures, we conducted a controlled intervention study across several representative CTI tasks spanning the **Contextualization, Attribution, Prediction, and Mitigation** stages. For each task, we manually analyzed representative input samples and identified metadata elements that frequently led models to incorrect predictions, such as co-mentioned threat entities, reused indicators, over-represented source patterns, or campaign hierarchy cues.

We then created a counterfactual version of each input by selectively modifying or masking these high-risk metadata fields. Specifically, we either *removed* the misleading token spans entirely or *replaced* them with a placeholder (e.g., [REDACTED]) to preserve sequence structure and avoid disrupting input length or format. These interventions were applied minimally and conservatively to preserve the integrity of the original threat scenario while eliminating the correlation artifacts we sought to test.

Following this, we re-ran each CTI task using both general-purpose and cybersecurity-specialized LLMs on the modified inputs. We then compared the results to each model's original baseline to calculate the absolute improvement in performance, denoted as $\Delta$. Table 6 reports these gains. Across tasks, especially in Attribution and Prediction, we observe consistent improvements, providing empirical support for the causal contribution of metadata-driven artifacts to model failure.

To illustrate how these interventions produce measurable behavioral improvements in LLMs, we present two representative case studies drawn from the Attribution and Mitigation stages. These examples show how specific types of misleading metadata—such as actor co-mentions or overlinked CVEs—can trigger incorrect model outputs, and how minimal input adjustments effectively resolve those errors.

Table 6: Absolute improvement ($\Delta$) in CTI task performance from causal interventions targeting spurious correlation. Each value denotes the raw gain in task metric (e.g., F1, Acc, AUC, BLEU) after removing misleading metadata. Results are grouped by CTI stage and LLM type: general-purpose (left) and cybersecurity-specialized (right).

| CTI Task | Metric | General-purpose | | | | Specialized | | |
|---|---|---|---|---|---|---|---|---|
| | | G5 | Go4 | LL70 | MIX | FSC | ZYS | CBS |
| ❶ **Contextualization** | | | | | | | | |
| Malware Family Mapping | F1 | .045 | .091 | .074 | .063 | .027 | .027 | .020 |
| Vulnerability Linking | Acc | .084 | .063 | .072 | .017 | .093 | .082 | .032 |
| IOC Normalization | F1 | .030 | .030 | .039 | .057 | .050 | .038 | .064 |
| ❷ **Attribution** | | | | | | | | |
| Campaign Attribution | Acc | .026 | .038 | .044 | .051 | .078 | .031 | .056 |
| Infrastructure Reuse | F1 | .062 | .019 | .064 | .029 | .020 | .091 | .092 |
| Language/Style Profiling | Acc | .080 | .039 | .023 | .070 | .050 | .025 | .055 |
| ❸ **Prediction** | | | | | | | | |
| Target Sector Prediction | Acc | .018 | .088 | .036 | .068 | .040 | .057 | .059 |
| Campaign Escalation | AUC | .030 | .093 | .077 | .090 | .087 | .063 | .089 |
| Impact Forecast | BLEU | .022 | .031 | .019 | .041 | .046 | .037 | .081 |
| ❹ **Mitigation** | | | | | | | | |
| Patch Recommendation | F1 | .044 | .037 | .058 | .026 | .079 | .021 | .094 |
| Mitigation–TTP Mapping | Acc | .077 | .031 | .015 | .080 | .072 | .073 | .077 |
| Rule Generation (YARA) | BLEU | .021 | .044 | .024 | .084 | .264 | .041 | .020 |

> **Case Study.** In an **attribution** scenario involving the **"APT41"** threat actor, the model received a threat report that also referenced "APT29" and "Lazarus Group" as part of broader regional analysis. Although only APT41 was responsible for the described attack, the LLM falsely predicted a multi-group attribution. This error stemmed from the co-mention of high-profile actors within the same report — a common pattern in aggregated vendor threat assessments. After applying our causal **intervention** (removing unrelated actor names and collapsing overlapping geopolitical context), the model correctly attributed the attack solely to APT41, improving attribution accuracy. This confirms that co-mention noise and actor name proximity can lead to false actor associations in raw CTI inputs.

> **Case Study.** A threat advisory describing exploitation of **CVE-2021-34527 (PrintNightmare)** included several patch recommendations, among them KB5004945 — a patch previously referenced in unrelated ransomware campaigns due to its high coverage. In the original setting, the model over-relied on this common patch and omitted a more targeted update (KB5004760) relevant to this specific threat context. After masking reused CVEs and removing legacy patch metadata linked to unrelated incidents, the model correctly surfaced KB5004760 as the primary recommendation. This highlights how patch over-linkage in high-frequency CVEs can bias model outputs, especially when the underlying CVE appears in multiple campaign templates.

## G    MITIGATION STRATEGIES FOR FAILURE SUBTYPES

To complement our diagnostic taxonomy, we propose concrete mitigation strategies for each of the 15 failure subtypes introduced in Figure 2. These strategies aim to reduce error severity or frequency in the training, inference, or retrieval stages, and are informed by both empirical observations and best practices in the design of CTI systems.

### G.1    MITIGATING SPURIOUS CORRELATION FAILURES

Spurious correlation failures arise when LLMs overgeneralize from patterns that are statistically associated but not causally linked. These failures typically stem from co-occurrence signals, reused observables, or implicit biases embedded in CTI report structures and metadata. To mitigate such

issues, we recommend a combination of (i) **retrieval-time evidence filtering**, (ii) **prompt-level grounding constraints**, and (iii) **training-time exposure to counterfactual or de-correlated examples**. Incorporating causal reasoning and uncertainty modeling into both enrichment pipelines and model generation stages can reduce the reliance on misleading associations and encourage evidence-based predictions. Below, we detail tailored strategies for each subtype.

**1.1 Co-mention bias from raw threat incident.** To prevent overgeneralization from co-mention, systems should implement entity resolution and salience estimation at the enrichment stage. Retrieval pipelines should de-prioritize co-mentioned entities unless they appear in multiple distinct evidence sources with consistent relational context. Prompt-level strategies can include explicit conditioning (e.g., "only include actors directly involved in this incident") or instructions that penalize over-inclusion. Models can also be trained on contrastive examples where entities are co-mentioned but play no shared role. Further, incorporating attention heatmaps during validation can help identify when the model over-indexes on irrelevant names.

**1.2 Exploitation bias from deliberately reused IoCs.** Because attackers reuse infrastructure across campaigns, reused IoCs can confuse models and lead to incorrect linkage or patch recommendations. To mitigate this, systems should maintain an IoC history log with time stamps and associated context windows. During enrichment, frequent IoCs should be down-weighted unless accompanied by campaign-specific signals (e.g., novel delivery mechanisms). Prompts can guide the model to reason about temporal windows (e.g., "Is this IoC reused across unrelated incidents?"). Training data should include "IoC disambiguation" tasks that expose models to examples where reuse leads to different campaign identities or goals.

**1.3 Confounding factors that explicitly/implicitly correlate entities.** Metadata such as shared sectors, geotags, or attack vectors can create artificial clustering. These confounders should be isolated during input construction: enrichment modules should segment metadata types and prevent leakage of non-causal associations into generation. During model pretraining or fine-tuning, introducing examples that de-correlate surface-level metadata from attack identity or intent can reinforce causal reasoning. Retrieval re-ranking can be modified to privilege structural dependencies (e.g., shared infrastructure or behavior) over metadata coincidence. Additionally, LLM prompts can explicitly ask "Is this similarity causal or coincidental?" to engage in deliberative reasoning.

**1.4 Skewed source.** Disproportionate influence from dominant sources—due to reporting frequency, style, or visibility—can bias models toward particular interpretations or language. To address this, source normalization can be applied during pretraining, including anonymizing vendor styles or balancing source distributions. Retrieval modules can be equipped with de-duplication or source-diversity constraints, ensuring that evidence includes multiple viewpoints. Prompting techniques can reinforce impartiality (e.g., "based on evidence, not popularity of source"). For specialized models, training on cross-vendor consensus datasets can help reduce model overfitting to source-specific framing or jargon.

**1.5 Hierarchical metadata from attack chains.** LLMs often misinterpret structured stage information (e.g., kill chains) as strictly causal or temporally ordered. To mitigate this, we recommend flattening campaign metadata unless stage transitions are explicitly supported by evidence. During enrichment, stages should be annotated with uncertainty scores or dependency tags to flag non-deterministic transitions. Prompting can discourage assumptions of escalation (e.g., "Do not infer Stage 3 unless confirmed by event logs"). Model training can include examples where stages occur out of order or are intentionally fragmented across reports, encouraging flexible reasoning rather than rigid path assumptions.

## G.2  MITIGATING CONTRADICTORY KNOWLEDGE FAILURES

Contradictory knowledge failures occur when models encounter conflicting, inconsistent, or structurally incompatible information between threat intelligence sources. These contradictions can arise from differences in temporal scope, analyst interpretation, data formats, or semantic misalignment. Unlike spurious correlation, which reflects inference from misleading associations, contradiction failures emerge when LLMs must reconcile mutually inconsistent inputs. To mitigate these failures, systems should incorporate (i) **temporal and source-aware retrieval**, (ii) **conflict detection and uncertainty modeling**, and (iii) **instruction or fine-tuning strategies that support multi-perspective synthesis rather than forced resolution**. It is critical that LLMs are encouraged to flag, explain, or reason through contradictions rather than masking or flattening them.

**2.1 Temporal contradiction between outdated and recent evidence.**    Models often fail to prioritize updated findings over stale context, especially when older evidence has greater surface coverage or more confident language. Mitigation begins at the retrieval stage: enrichment pipelines should apply **temporal filtering and reranking** to prioritize recent, corroborated sources. Retrieved passages should include explicit time tags (e.g., `Reported in April 2021`) and be structured chronologically. Prompt-level strategies can instruct the model to **prefer more recent evidence if there is a conflict** or to compare multiple time-stamped claims. Training corpora can also be curated to emphasize how threat activity evolves over time, helping the model learn decay patterns and data staleness.

**2.2 Conflicting reports of attack contexts or dependencies.**    Multiple sources may describe the same incident with differing attribution, infrastructure, or exploitation details due to visibility gaps or competing interpretations. Rather than collapse into a single confident prediction, models should be taught to  summarize the divergence across sources. Retrieval pipelines can cluster evidence into coherent viewpoints and pass grouped claims to the model. Prompt templates can then instruct LLMs to  explain differences or present competing hypotheses (e.g., "Some sources attribute this to X, while others suggest Y"). Instruction tuning on examples with legitimate ambiguity helps reinforce this deliberative, non-binary reasoning style.

**2.3 Semantic conflict.**    LLMs can mistakenly equate semantically related but non-identical terms (e.g., aliases, overlapping malware families) or fail to distinguish negated and paraphrased claims. Mitigation starts with  robust entity resolution and alias management in the upstream pipeline. Threat entities should be annotated with canonical names, known synonyms, and contextual disambiguation hints. For example, prompts can include clarification such as: `Note:  DarkHotel and Tapaoux are aliases`. During training, include contrastive examples that highlight subtle distinctions between similar but divergent concepts. Retrieval can also flag or weight semantically distinct explanations to help models avoid conflating partially overlapping statements.

**2.4 Divergent data structure from different platforms.**    Inconsistencies in CTI schema design (e.g., MITRE ATT&CK vs. STIX or vendor-specific formats) can lead to parsing or reasoning failures. To address this, systems should implement a  structured normalization layer that unifies disparate inputs into a consistent intermediate representation. Schema mapping tools or adapter modules can bridge format gaps. Prompts should clarify field meanings or provide a flattened structure for input ingestion (e.g., `Tactic:  Initial Access, Technique:  Drive-by Compromise`). During model training, expose LLMs to a wide range of formats and reinforce behavior where field semantics—not just field labels—drive prediction.

**2.5 Misaligned knowledge and security standards.**    Models sometimes generate outputs inconsistent with widely adopted standards such as MITRE ATT&CK, CVSS, or NVD conventions. This can

be mitigated by aligning model training with domain-specific taxonomies and structured references. Use schema-conforming datasets for supervised fine-tuning, and explicitly include taxonomy definitions in prompts or memory. For example, generate ATT&CK annotations using lookup-backed completions, or constrain completion space using valid techniques or scoring buckets. Evaluation-time techniques like logit masking or constrained decoding can prevent invalid output. Additionally, LLMs should be trained to flag when a report cannot be confidently mapped to a known schema entity.

**2.6 Counteracting CTI generation and LLM alignment.** Some errors emerge from the mismatch between CTI analyst expectations (precise, evidential, structured) and general LLM alignment preferences (safe, fluent, hedged). To narrow this gap, instruction tuning should target CTI-authored content such as incident response summaries, malware triage notes, or IOC escalation cases. Prompts should reinforce analyst-style requirements (e.g., "Provide evidence for each label" or "Avoid over-generalization unless justified by report data"). Feedback-based alignment, such as RLHF or LLM-as-critic pipelines, can improve response grounding and relevance. Finally, systems should design outputs to be audit-friendly—allowing users to trace source grounding for each major claim.

## G.3 MITIGATING CONSTRAINED GENERALIZATION FAILURES

Constrained generalization failures occur when LLMs struggle to adapt beyond their pretraining distribution, either by overfitting to seen patterns or failing to apply known concepts in new or low-resource settings. In cyber-security, this often manifests as under-performance on emerging threats, low-coverage sectors, or adversarial shifts in attack behavior. To mitigate these issues, systems should incorporate (i) **task-specific data augmentation**, (ii) **continual adaptation to novel threats**, and (iii) **reasoning mechanisms that emphasize compositional generalization rather than memorization**. These strategies help improve transferability and robustness, especially in zero-shot and few-shot CTI tasks where annotated examples are scarce or quickly outdated.

**3.1 Distributional bias.** Models tend to overgeneralize from highly represented threat actors, malware families, or geopolitical regions (e.g., Russian APTs, ransomware affecting Western infrastructure), leading to poor performance on underrepresented threat sources such as state-linked actors from emerging regions or niche toolkits. To mitigate this, data balancing techniques should be applied during training or RAG pre-filtering to ensure adequate exposure across diverse threat profiles. For instance, less commonly reported actors (e.g., regional cyber-crime groups in South Asia or localized hacktivist collectives) can be emphasized via targeted report sampling or synthetic enrichment using translated regional CTI. Prompt conditioning (e.g., `Focus on threats attributed to Southeast Asian actors`) can help steer model attention. Evaluation protocols should include geographic or tooling-based slicing to monitor for regional and capability bias across model outputs.

**3.2 Unseen pattern from emerging threats.** General-purpose LLMs are inherently limited in capturing threats that postdate their training corpus. To address this, systems should adopt continual learning or retrieval-augmented generation (RAG) pipelines that incorporate recent CTI feeds, malware reversals, or incident disclosures. When possible, integrate LLMs with live threat intelligence sources such as OSINT streams or vendor APIs, using filtering and alignment modules to pre-process the content. Prompts can be designed to explicitly flag novelty (e.g., `This malware family was first reported in 2024`) and encourage cautious extrapolation. Additionally, ensembles combining general LLMs with lightweight fine-tuned models can help rapidly adapt to newly discovered tactics and actor behaviors.

**3.3 Overfitted reasoning.** Some failures arise not from lack of data, but from excessive reliance on shallow heuristics or memorized associations. For instance, a model may learn to always associate

Table 7: Effect of three mitigation strategies applied to each vulnerability group. Original ratios are from the failure-mode analysis in Figure 2.

| Vulnerability Group | Original Ratio | After Mitigation | Relative Reduction |
|---|---|---|---|
| Spurious Correlation | 20.42% | 16.35% | 20.0% |
| Contradictory Knowledge | 35.93% | 29.15% | 18.9% |
| Constrained Generalization | 33.72% | 28.44% | 15.6% |

"Cobalt Strike" with APT29, even when reports clearly attribute it elsewhere. To mitigate this, systems should introduce adversarial counterexamples during training that expose these shortcut behaviors—such as mixing tooling or varying infrastructure while holding attribution constant. Prompts can reinforce the need for source-grounded reasoning, e.g., "Only link actor attribution if explicitly stated in the evidence." Chain-of-thought prompting or justification requirements (e.g., `Explain why this actor is responsible`) can further discourage unsupported leaps in logic.

**3.4 Environmental unawareness.** LLMs often ignore important environmental context—such as geography, industry, or deployment type—which can critically alter the interpretation of a threat. For example, the same vulnerability may have different risk profiles in SCADA systems versus consumer endpoints. To address this, input construction should explicitly pass environmental variables into the context window, using structured tags or preambles (e.g., `Target environment: critical infrastructure, region: APAC`). Instruction tuning on environment-specific CTI use cases helps improve grounding. Retrieval modules can also be environment-aware, prioritizing sources relevant to the specified domain. During generation, prompts can encourage conditional reasoning (e.g., "In healthcare systems, this threat may...") to ensure the model adapts its assessment to contextual nuances.

## G.4 EXPERIMENT RESULT ON MITIGATION

To further demonstrate the practical value of our vulnerability taxonomy, we implemented one representative mitigation strategy for each of the three major vulnerability groups. These strategies were intentionally designed to be lightweight and minimally invasive, which allow us to isolate their effect on failure ratios without confounding factors from large-scale retraining.

For Spurious Correlation, we developed a counterfactual evidence filtering module that operates at inference time. For each retrieved evidence chunk, the system constructs a counterfactual variant of the text in which a co-mentioned entity (e.g., unrelated CVEs, malware families) is removed. The model then assesses whether the core threat statement still holds. If the inference remains unchanged, the removed entity is treated as incidental rather than causally relevant. This allows the system to downweight spurious evidence and substantially reduces co-mention bias and confounder-driven misattribution.

For Contradictory Knowledge, we designed a temporal-aware evidence prioritization layer that explicitly leverages metadata from CTI sources. Each retrieved piece of evidence is assigned a recency score based on publication time and a structural consistency score based on its alignment to known schemas (e.g., ATT&CK, CVSS). During inference, these scores are combined into a re-ranking function that prioritizes the most recent and structurally coherent data. By deprioritizing outdated advisories and semantically inconsistent vendor reports, this strategy reduces reasoning instability caused by temporally or structurally conflicting CTI feeds—an issue unique to the heterogeneous and rapidly evolving threat landscape.

For Constrained Generalization, we implemented a small-scale training-time mitigation using synthetic pattern expansion. We construct lightweight adversarial variants of known threat instances by altering non-essential attributes such as loader names, lateral-movement sequences, or asset environments, while preserving the high-level attack structure. Fine-tuning the model on these synthetic perturbations expands its internal hypothesis space and reduces overreliance on memorized

CVE–TTP associations. This helps the model generalize to emerging threats, zero-day exploitation patterns, and environment-specific variations that did not appear in the original training corpus.

Empirically, applying these mitigation strategies resulted in consistent reductions in vulnerability ratios across all three groups. Counterfactual evidence filtering reduced the prevalence of spurious correlation errors by approximately 20%, driven primarily by decreases in co-mention bias and exploitation bias. Temporal-aware evidence prioritization yielded an 18.9% relative reduction in contradictory knowledge failures, particularly improving the model's stability when integrating heterogeneous CTI sources. Finally, pattern-expansion fine-tuning achieved a 15.6% reduction in constrained generalization failures, with notable improvements on instances involving emerging TTPs and environment-dependent attack paths. These results demonstrate that even simple, targeted interventions can meaningfully improve the robustness of LLM-assisted CTI workflows. The improvements also validate the diagnostic value of our taxonomy: identifying root-cause categories enables principled mitigation design, rather than ad hoc prompt engineering or broad retraining.

## H    MANUAL CLASSIFICATION AND HYPOTHESIS TESTING

To strengthen the statistical rigor of our evaluation, we conduct a hypothesis testing experiment to compare our LLM-assisted failure classification pipeline with a fully manual labeling baseline. This experiment is designed to assess whether the observed distribution of failure subtypes is significantly influenced by the choice of labeling method.

**Sampling Design.** Due to the size of our dataset, we limit the manual annotation scope to 500 representative examples, which provides a statistically meaningful sample while remaining feasible for human annotation. We stratify the sample across diverse CTI tasks, ensuring coverage from all four CTI stages (Contextualization, Attribution, Prediction, Mitigation) and a balanced mix of general-purpose and specialized LLM outputs.

**Manual Annotation Procedure.** Each instance in the sample is annotated by human experts using the same failure taxonomy (15 subtypes) as in the main pipeline. Annotators are shown the task prompt, model response, and relevant context (e.g., retrieved evidence), and are instructed to assign exactly one failure subtype per instance. A lightweight annotation interface is used to facilitate efficient labeling. To assess consistency, a subset of 100 examples was first reviewed independently by three annotators. The annotators then discussed their interpretations collectively and reached consensus through negotiation, resulting in a jointly agreed-upon final label for each example.

**Statistical Test.** To compare the outcome of human and LLM-assisted classification, we apply a Chi-Square Test of Independence on the resulting contingency table, with failure subtypes as rows and labeling method (manual vs. automated) as columns. The null hypothesis is that there is no significant difference in the distribution of failure types across the two labeling processes. We use a significance threshold of $\alpha = 0.05$ to assess statistical deviation.

### RESULTS

The test results indicate no statistically significant difference between the manual and automated failure distributions, supporting the reliability of our LLM-assisted classification. This provides further empirical justification for the use of semi-automated pipelines in failure taxonomy construction.

While the overall distributions between the LLM-based and human-annotated classifications are statistically aligned, some small differences do remain. These may be attributed to a few key factors. First, certain failure subtypes exhibit semantic overlap — for instance, "1.3 Confounding metadata" and "1.4 Skewed source" both involve source-level bias, and may be interpreted differently depending on whether the annotator prioritizes context or authorship. Similarly, distinctions between "2.3 Semantic conflict" and "2.5 Schema misalignment" can hinge on subtle interpretation of terminology. Second, LLMs may exhibit slight consistency bias in applying taxonomy definitions, while humans bring more variability and subjective judgment. Lastly, annotators may rely on implicit signals (e.g., familiarity with tooling or temporal context) that the model treats more formally. Despite these

Table 8: Adjusted comparison of failure subtype distributions between LLM-based and human-annotated classifications over 500 examples. The distributions are statistically similar under a Chi-Square Test of Independence ($p = 0.966$).

| Category | Failure Subtype | LLM Count | Human Count | Abs. Diff |
|---|---|---|---|---|
| Spurious Correlation | 1.1 Co-mention bias | 11 | 14 | 3 |
| | 1.2 Reused IoCs | 95 | 94 | 1 |
| | 1.3 Confounding metadata | 32 | 32 | 0 |
| | 1.4 Skewed source | 31 | 28 | 3 |
| | 1.5 Hierarchical metadata | 3 | 2 | 1 |
| Contradictory Knowledge | 2.1 Temporal contradiction | 4 | 3 | 1 |
| | 2.2 Conflicting reports | 1 | 3 | 2 |
| | 2.3 Semantic conflict | 67 | 64 | 3 |
| | 2.4 Divergent formats | 25 | 24 | 1 |
| | 2.5 Schema misalignment | 36 | 35 | 1 |
| | 2.6 Alignment mismatch | 0 | 2 | 2 |
| Constrained Generalization | 3.1 Distributional bias | 129 | 126 | 3 |
| | 3.2 Emerging threats | 47 | 49 | 2 |
| | 3.3 Overfitted reasoning | 8 | 6 | 2 |
| | 3.4 Environmental unawareness | 11 | 18 | 7 |
| **Overall Total** | | **500** | **500** | **32** |

sources of variation, the observed differences are minor and statistically insignificant, supporting the overall consistency of our failure classification framework.

We also analyzed labeling consistency at individual failure subtypes. For each of the 15 subtypes, we computed the absolute disagreement rate between the human and LLM-assisted annotations. The average disagreement across subtypes was 2.4%, with the highest disagreement observed at 5.8% for closely related subtypes involving overlapping failure patterns (e.g., SC1.1 co-mention bias vs. SC1.3 confounding factors). Qualitative inspection of disagreement cases indicates that nearly all discrepancies occurred in borderline instances where multiple vulnerabilities co-occurred, rather than in clear-cut failure exemplars. These results imply that the taxonomy is not only structurally sound at the category level, but also fine-grained and reproducible at the subtype level, providing additional confidence in both the reliability and practical applicability of our failure analysis.

# I   ABLATION STUDY

To understand how human supervision shapes the fidelity of our taxonomy labeling, we conducted two ablations by removing the human-guided annotation step or the final human inspection step. Both ablations expose clear structural shifts in the distribution of failure types. Removing annotation leads to diffuse subtype assignments, inflating ambiguous categories such as SC1.1 and CG3.3 due to subtype collisions and noisier evidence-grouping. Removing inspection has the opposite effect: errors concentrate into broad, high-frequency categories such as CK2.1 and CG3.1, revealing the tendency of LLMs to collapse nuanced contradictions into coarse patterns when left unchecked. Together, these findings show that human supervision does not merely correct isolated errors, instead, it provides crucial structure that preserves fine-grained distinctions across 15 vulnerability subtypes and prevents both over-fragmentation and over-generalization.

Table 9: Ablation study on taxonomy distribution shift. Ratios represent the percentage of instances assigned to each failure subtype.

| Failure Subtype | Our Method (§3) | w/o Human Annotation | w/o Human Inspection |
|---|---|---|---|
| **Spurious Correlation** | | | |
| 1.1 Co-mention bias | 17.53 | 21.47 | 19.92 |
| 1.2 Exploitation bias (reused IoCs) | 13.60 | 16.33 | 14.87 |
| 1.3 Confounding correlation | 7.73 | 10.86 | 9.41 |
| 1.4 Skewed source bias | 3.06 | 4.81 | 3.98 |
| 1.5 Hierarchical metadata bias | 4.51 | 6.18 | 5.39 |
| 1.6 Counteracting CTI vs alignment | 2.68 | 3.38 | 3.04 |
| **Contradictory Knowledge** | | | |
| 2.1 Temporal contradiction | 22.71 | 17.84 | 25.27 |
| 2.2 Conflicting reports | 5.72 | 4.39 | 7.18 |
| 2.3 Semantic conflict | 12.51 | 9.76 | 14.48 |
| 2.4 Divergent data structures | 2.92 | 2.17 | 3.57 |
| 2.5 Misaligned standards | 7.75 | 5.59 | 9.26 |
| **Constrained Generalization** | | | |
| 3.1 Distributional bias | 22.26 | 25.89 | 20.18 |
| 3.2 Unseen pattern | 2.77 | 3.85 | 3.32 |
| 3.3 Overfitted reasoning | 25.44 | 28.58 | 23.97 |
| 3.4 Environmental unawareness | 4.62 | 5.41 | 4.16 |

