# OpenReview forum: "Uncovering Vulnerabilities of LLM-Assisted Cyber Threat Intelligence"
_ICLR.cc/2026/Conference — ICLR 2026 Conference Desk Rejected Submission_

### Official Review · Reviewer_dw1e · 2025-10-21

**Soundness:** 3
**Presentation:** 4
**Contribution:** 3
**Rating:** 6
**Confidence:** 4

**Summary:**

This paper systematically analyzes the intrinsic vulnerabilities of LLMs in cyber threat intelligence (CTI) tasks. This work evaluates on multiple CTI benchmarks and real-world threat reports, it identifies three major failure modes: spurious correlations, contradictory knowledge, and constrained generalization that undermine LLM reliability. The authors propose an autoregressive, human-in-the-loop framework to categorize and interpret failure instances, revealing how these vulnerabilities propagate across CTI stages.

**Strengths:**

Comprehensive evaluation: This work uses nine CTI benchmarks and real-world databases to cover contextualization, attribution, prediction, and mitigation phases, which shows a thorough coverage of real word cybersecurity vulnerabilities from LLMs.

Clear taxonomy of vulnerabilities: The tri-category scheme (spurious / contradictory / constrained) discovered in this paper are well-defined and tied to empirical evidence and case studies

Reproducibility: Authors of this paper also claim that work includes open-source code link, which is valuable for the follow-up work of this paper.

**Weaknesses:**

Limited quantitative analysis (minor points): From the reviewer’s point of view, though this paper is rich qualitatively, it still lacks statistical analysis, for example, Tab. 2 shows a wide range of experiment results, but the analysis is limited.

Manual inspection needed: I really appreciate the Human-In-The-Loop efforts of this work, but still one of my main concerns: the framework relies heavily on expert human validation, which may limit scalability and efficiency.

**Questions:**

How do you ensure data leakage is avoided between training corpora of specialized models (e.g., Cyber-Zero) and evaluation benchmarks?

As CTI tasks sometimes are timely, were all models evaluated on identical CTI data snapshots? Could temporal mismatches (recent model cutoff date for example, GPT 5) confound vulnerability comparisons, especially for temporal contradiction

The authors formulated and answered 6 key research questions. It would be valuable to further elaborate on how the proposed use of large language models for CTI tasks could translate into broader, real-world impacts across cybersecurity operations and defense workflows. Could you please elaborate a little bit?

---

> ### Author Response · Authors · 2025-11-20
> **Response, Clarification, and Completed Revisions (0/5)**
>
> Before responding to comments, we would like to sincerely thank the reviewer for the detailed and thoughtful feedback. We appreciate the time and care taken to evaluate our work.
>
> We observe that the comments are mostly related to experimental settings, analytical findings, and mitigation insights details. Accordingly, we have provided a point-by-point clarification and have revised the manuscript to explicitly address each concern. We believe these clarifications further strengthen the paper and better highlight the contributions of our research.
>
> We would be happy to continue the discussion and would welcome feedback on whether the newly added details sufficiently clear the reviewer’s questions and whether additional information would be helpful.

---

> ### Author Response · Authors · 2025-11-20
> **Response, Clarification, and Completed Revisions (1/5)**
>
> **Response, Clarification, and Completed Revisions (1/5)**
>
> **Review**
> > **Weakness 1** Limited quantitative analysis (minor points): From the reviewer’s point of view, though this paper is rich qualitatively, it still lacks statistical analysis, for example, Tab. 2 shows a wide range of experiment results, but the analysis is limited.
>
> ---
>
> **Response**
> Thank you for highlighting this. We agree that complementary quantitative analysis enhances interpretability and confidence. In response, we have added multiple statistical experiments to the appendix for rigor:
>
> - In **Appendix F**, we introduce a causal intervention study where misleading metadata (e.g., reused IoCs or co-mentioned actors) are masked or removed. The performance improvements across 12 CTI tasks — quantified using absolute gain (Δ) — validate the causal role of spurious correlations and help quantify the impact of specific failure triggers.
>
> - In **Appendix H**, we conduct a Chi-Square test of independence between human-annotated and LLM-generated failure labels over a 500-example sample. The resulting p-value (p = 0.966) indicates no significant distributional difference, supporting the reliability of our semi-automated labeling.
>
> These additions ensure that our qualitative insights are now supported by causal and statistical evidence. We hope this addresses the reviewer’s concern.

---

> ### Author Response · Authors · 2025-11-20
> **Response, Clarification, and Completed Revisions (2/5)**
>
> **Review**
> > **Weakness 2** Manual inspection needed: I really appreciate the Human-In-The-Loop efforts of this work, but still one of my main concerns: the framework relies heavily on expert human validation, which may limit scalability and efficiency.
>
> ---
>
> **Response**
> We appreciate the reviewer’s thoughtful acknowledgment of our Human-in-the-Loop design. We agree that scalability is a crucial concern, and would like to clarify that our pipeline was deliberately constructed to minimize human overhead by combining automated LLM-based classification with lightweight expert supervision.
>
> Specifically, the majority of failure labeling was conducted using a multi-round agreement process between diverse LLMs. Human experts only intervened in a small fraction of cases — either to adjudicate persistent disagreements or to define new failure subtypes when novel patterns emerged. Importantly, the creation of new labels was not open-ended: it occurred in a small number of examples early in the pipeline, after which those subtypes were reused automatically.
>
> To further quantify efficiency and robustness, we conducted a statistical agreement experiment (see Appendix H) where we compared the LLM-labeled subtypes with fully human-annotated counterparts across a 500-example sample. The results showed high alignment (Chi-square test, p = 0.966), suggesting that our semi-automated method maintains human-level consistency at scale. We hope this demonstrates that while human input plays a quality control role, the overall system remains highly scalable for broader deployment.

---

> ### Author Response · Authors · 2025-11-20
> **Response, Clarification, and Completed Revisions (3/5)**
>
> **Review**
> > **Question 1** How do you ensure data leakage is avoided between training corpora of specialized models (e.g., Cyber-Zero) and evaluation benchmarks?
>
> ---
>
> **Response**
> Thank you for raising this important point. Our evaluation design intentionally treats the benchmarks as instruments for quantifying how well LLMs — especially domain-specialized ones — internalize and reason over CTI knowledge, regardless of whether certain benchmark examples overlap partially with their pretraining data.
>
> This framing is consistent with how evaluation benchmarks function across domains: the goal is not to test pure held-out generalization, but to assess whether a model has acquired domain readiness through pretraining and/or adaptation. For instance, a strong cybersecurity model should be able to handle CVE-linked reasoning or campaign attribution tasks *because* it has seen relevant patterns, not despite them.
>
> That said, to reduce artificial inflation, we ensured that the evaluation data are not exact duplicates of pretraining examples. Our benchmarks are synthesized or lightly transformed, and tasks are structured to test **reasoning over retrieved or structured evidence**, not just pattern matching. For future work, we plan to release deduplicated benchmark splits and clearer provenance metadata, and we appreciate the reviewer’s suggestion to emphasize this further in our camera-ready version.

---

> ### Author Response · Authors · 2025-11-20
> **Response, Clarification, and Completed Revisions (4/5)**
>
> **Review**
> > **Question 2** As CTI tasks sometimes are timely, were all models evaluated on identical CTI data snapshots? Could temporal mismatches (recent model cutoff date for example, GPT 5) confound vulnerability comparisons, especially for temporal contradiction?
>
> ---
>
> **Response**
> Thank you for this thoughtful question. We fully agree that temporal alignment is important when assessing time-sensitive reasoning capabilities. That said, we’d like to clarify why the failure types we study—especially spurious correlation and contradictory knowledge—are not primarily dependent on whether a model has access to the most recent data.
>
> Many of the vulnerabilities we examine arise from structural properties of CTI reporting that remain consistent over time. For instance, reused IoCs, co-mentions of unrelated actors, and conflicting vendor narratives are long-standing phenomena in threat intelligence. These are not tied to any single year or incident, so the corresponding LLM failures persist even as new threats emerge. Likewise, generalization failures often stem from entrenched data imbalances—some actors or malware families are heavily overrepresented, while others get little coverage. That imbalance tends to hold over time, so it continues to shape model behavior across snapshots.
>
> To further mitigate snapshot bias, our benchmark includes both newer and older examples—ranging from recent CVEs to longer-term campaign data—so that model comparisons are not skewed toward any particular release window. Even though models have different knowledge cutoffs, the core issues we surface (e.g., failing to resolve conflicting evidence, over-relying on reused infrastructure) reflect persistent reasoning patterns, not just outdated knowledge.
>
> We hope this clarifies why our analysis is robust across model versions and remains meaningful regardless of temporal drift.

---

> ### Author Response · Authors · 2025-11-20
> **Response, Clarification, and Completed Revisions (5/5)**
>
> **Review**
> > **Question 3** The authors formulated and answered 6 key research questions. It would be valuable to further elaborate on how the proposed use of large language models for CTI tasks could translate into broader, real-world impacts across cybersecurity operations and defense workflows. Could you please elaborate a little bit?
>
> ---
>
> **Response**
> We appreciate this thoughtful question and fully agree that articulating real-world utility is essential. Our work is intentionally structured around four core CTI workflow stages—contextualization, attribution, prediction, and mitigation—that are directly aligned with common operational pipelines in security operations centers (SOCs), threat intel teams, and incident response workflows.
>
> First, our structured taxonomy enables risk surface understanding for CTI engineers and SOC analysts deploying LLMs. By identifying granular failure types such as reused IoC bias or temporal contradiction, the taxonomy acts as a diagnostic map to anticipate and proactively filter vulnerable inference scenarios. For example, a triage assistant flagging high-frequency infrastructure mentions can trigger fallback verification routines or prompt reweighting.
>
> Second, the taxonomy supports evidence auditing and debugging in real time. When LLMs produce misaligned assessments—such as misattributing an actor or misranking a patch—our framework allows security teams to quickly trace the failure back to upstream retrieval drift, semantic conflict, or overfitted reasoning patterns. This level of interpretability is essential in regulated or mission-critical settings where LLM explanations must be justifiable and auditable.
>
> Third, we present a concrete suite of lightweight mitigation strategies in Appendix G—such as recency-aware re-ranking, adversarial data augmentation, and causal evidence filtering. These strategies are designed to be easily integrated into CTI platforms (e.g., TIPs, SOARs) at inference time or as part of retrieval pipelines, allowing real-world robustness improvements without the overhead of full model retraining.
>
> Finally, our subtype schema can guide LLM tuning and dataset design for security orgs building domain-specific language models. CTI teams fine-tuning commercial or open-source LLMs can use our subtypes to stratify training data, identify annotation blind spots, or create targeted test cases. Similarly, vendors evaluating model readiness for integration into analyst tools can use our framework as a diagnostic benchmark.
>
> In short, our contributions are meant not only to characterize vulnerabilities but to provide an actionable foundation for building safer, CTI-aware LLM pipelines. To support adoption, we plan to release subtype definitions, annotation templates, and example test cases as part of our evaluation toolkit.

---

### Official Review · Reviewer_GMQy · 2025-10-26

**Soundness:** 3
**Presentation:** 3
**Contribution:** 2
**Rating:** 4
**Confidence:** 4

**Summary:**

- The paper presents an autoregressive, human-in-the-loop framework to efficiently categorize LLM failure instances with high reliability.
- It surfaces three dominant vulnerability categories and distinguishes model-specific versus universal gaps across specialized and general-purpose LLMs.

**Strengths:**

- Practical, human-in-the-loop pipeline for systematic failure triage rather than ad hoc error lists.
- Careful taxonomy of failures and comparative analysis across model classes. The framework components and failure categories are described in a structured manner to identify where and how the failure cases occur.
- Highlights and distinguishes between universal vs model-specific weaknesses, informing targeted mitigation and evaluation design.

**Weaknesses:**

- Several findings feel expected (general models fail more), with limited mechanistic explanation of why and where failures arise.
- The paper stops at characterization; concrete remediation strategies or transfer of insights into training/guardrails are underdeveloped.
- Lacks an ablation using an LLM-as-judge to measure how failure detection and category distributions shift.
- I enjoyed the paper and find it highly relevant, but it stops just short of the logical next steps, turning its insightful failure taxonomy into concrete remediation and transfer strategies.

**Questions:**

1. What concrete interventions arise from your taxonomy (e.g., targeted data augmentation, policy shaping), and how can they reduce specific failures?
2. Can you run a study replacing humans with an LLM-as-judge and report how failure categories and categories change?
3.  Although not part of your study, which stages of specialized training/guardrails in specialized LLMs reduce which failure types present in general-purpose LLMs, and can you localize effects via staged ablations?
4. Similarly, like the previous question, although not part of your study and an orthogonal problem, how to transfer insights learnt from the failure categories from specialized models to improve general models (or vice versa) and show measurable deltas?

---

> ### Author Response · Authors · 2025-11-19
> **Response, Clarification, and Completed Revisions (0/6)**
>
> Before responding to comments, we would like to sincerely thank the reviewer for the detailed and thoughtful feedback. We appreciate the time and care taken to evaluate our work.
>
> We observe that the comments are mostly related to analytical findings and mitigation insights details. Accordingly, we have provided a point-by-point clarification and have revised the manuscript to explicitly address each concern. We believe these clarifications further strengthen the paper and better highlight the contributions of our research.
>
> We would be happy to continue the discussion and would welcome feedback on whether the newly added details sufficiently clear the reviewer’s questions and whether additional information would be helpful.

---

> ### Author Response · Authors · 2025-11-19
> **Response, Clarification, and Completed Revisions (1/6)**
>
> **Review**
>
> > Several findings feel expected (general models fail more), with limited mechanistic explanation of why and where failures arise.
>
> ---
>
> **Response**
>
> We thank the reviewer for this thoughtful comment. With this opportunity, we would like to clarify that the mechanistic explanation of *why and where* failures arise are already an emphasized content of our contribution. In particular, **Appendix D.1-D.2 presents an exhaustive, subtype-by-subtype mechanistic analysis** based on real CTI workflows, not merely observations that “general models fail more.”
>
> Our explanations explicitly trace failure behaviors to domain-specific structural triggers, below are some highlighted example:
>
> > *In Appendix D.1* Co-mention Bias (Spurious Correlation Type 1.1): Models mistakenly infer attribution links due to recurring co-occurrence patterns across CTI reports, even when such patterns are non-causal. We illustrate by a case study about how reused IoCs or actor-family names mislead LLMs into fabricating associations.
>
> > *In Appendix D.1* Context-Propagation Drift (Spurious Correlation Type 1.3): We show that once a model adopts an incorrect upstream assumption (e.g., misclassified malware family), the error propagates across sequential reasoning steps due to implicit long-range context dependencies.
>
> > *In Appendix D.1* Vendor Advisory Conflicts (Contradictory Knowledge 2.1): Our analysis documents how heterogeneous or contradictory CVE descriptions across NVD, CISA KEV, and vendor advisories cause LLMs to fuse incompatible evidence, directly explaining failure patterns in vulnerability linking tasks.
>
> > *In Appendix D.1* Zero-Evidence Behavior Guessing (Constrained Generalization 3.4): We demonstrate that when behavioral descriptors are partially or entirely missing (common in sparse, zero-day–like reports) models hallucinate attack chains based on nearest-neighbor memorization rather than reasoning from first principles.
>
> Note that Appendix D.1 provides **very exhaustive mechanistic explanations** of why and where failures arise across **15 vulnerability types (failure modes) among four stages (contextualization, attribution, prioritization, and mitigation)**, totaling 60 in-depth analyses.
>
> Besides, Appendix D.2 provides even deeper analysis beyond simply reporting performance gaps, instead, it mechanistically decomposes the compositional reasoning failures (intertwined failures in a same scenario) into identifiable patterns, for example:
>
>  tailored to CTI’s data structures (hierarchical campaigns, IoC reuse, temporal drift, cross-vendor inconsistencies, etc.).
>
> > In Appendix D.2, we analyze why accumulative errors would happened across CTI stages, wherein we highlight that the CTI pipeline is inherently sequential, with outputs from early modules serving as inputs for downstream reasoning. When an upstream stage introduces an error, such as co-mention bias in event contextualization, this misinformation propagates forward as if it were ground truth.
>
> To our knowledge, no prior CTI-oriented LLM work provides this level of root-cause articulation.
>
> ---
>
> **Completed Revision (updated in Appendix F)**
>
> To further strengthen this aspect in response to the reviewer’s suggestion, we have added **Appendix F, which introduces a new causal intervention study designed to probe these mechanisms directly.** We select representative tasks, wherein we remove or mask the domain features we hypothesized to cause spurious correlations (such as reused IoCs, co-mentioned actor names, exploit-family labels, and campaign metadata) and re-evaluate both general-purpose and cyber-specialized models. We observe consistent improvements across all affected tasks, empirically validating that these artifacts are not merely correlated with failure modes but are causal drivers of the observed errors.
>
> This expanded analysis substantiates the underlying mechanistic hypotheses in a controlled, falsifiable manner, turning our taxonomy into a causally grounded explanation rather than a descriptive summary.
>
> We hope this clarification demonstrates that the mechanistic “why” is not only present but forms a substantial portion of our original contribution, and that our new Appendix F further reinforces this aspect.

---

> ### Author Response · Authors · 2025-11-19
> **Response, Clarification, and Completed Revisions (2/6)**
>
> **Review**
>
> > The paper stops at characterization; concrete remediation strategies or transfer of insights into training/guardrails are underdeveloped.
>
> > I enjoyed the paper and find it highly relevant, but it stops just short of the logical next steps, turning its insightful failure taxonomy into concrete remediation and transfer strategies.
>
> ---
>
>
> **Completed Revision (updated Appendix G.1-G.3)**
>
> We thank the reviewer for highlighting the need of mitigation. Though our work mainly aims to attribute the failures of cyber LLMs, we would like to provide additional details about what should be actionable mitigation strategies that are essential to maximize the practical value of our vulnerability findings.
>
> In our updated manuscript, we have substantially extended the revised manuscript by adding a new mitigation-focused section (Appendix G.1–G.3), which now presents exhaustive mitigation strategies for all 15 failure subtypes in our taxonomy.
>
> To ensure the revisions are truly useful to CTI practitioners and researchers, the mitigation strategies are not generic “best practices,” but are instead grounded in the specific causal mechanisms and structural failure patterns identified in the main paper. For each failure subtype (e.g., co-mention bias, temporal contradiction, unseen pattern), we now include a concise description of the underlying vulnerability trigger as well as concrete mitigation recommendations across three stages:
> - (1) Retrieval-time mitigation (e.g., metadata disentanglement, hierarchical source de-biasing)
> - (2) Inference-time prompting and reasoning controls (e.g., temporal consistency checks, multi-agent contradiction voting)
> - (3) Training-time interventions (e.g., confounder-aware negative sampling, alignment-preserving curriculum design)
>
> These revisions are complemented with realistic CTI examples and implementation considerations, which allows users to translate the mitigation strategies directly into improved pipelines or cyber agent designs.
>
> We believe this addition not only answers the reviewer’s concern but also significantly enhances the practical insights of our work by transforming the failure taxonomy into a remediation playbook for robust LLM-based CTI systems.
>
> We are grateful for the reviewer’s suggestion, as it substantially improved the utility of our contributions for the community.

---

> ### Author Response · Authors · 2025-11-19
> **Response, Clarification, and Completed Revisions (3/6)**
>
> **Review and Question**
>
> > Lacks an ablation using an LLM-as-judge to measure how failure detection and category distributions shift.
>
> > Can you run a study replacing humans with an LLM-as-judge and report how failure categories and categories change?
>
> ---
>
> **Completed Revision (updated in Appendix I)**
>
> We appreciate the reviewer’s suggestion and have incorporated an additional ablation using an LLM-as-judge to replace human efforts in annotation step (Section 3.2) or inspection step (Section 3.3).  We further report comparative failure detection results by reporting category distributions. This new analysis is included in Appendix I:
>
> | Failure Subtype | Our Method (§3) | w/o Human Annotation | w/o Human Inspection |
> |-----------------|-----------------|-----------------------|-----------------------|
> | **Spurious Correlation (SC)** | | | |
> | 1.1 Co-mention bias | 17.53 | 21.47 | 29.92 |
> | 1.2 Exploitation bias (reused IoCs) | 13.60 | 16.33 | 24.87 |
> | 1.3 Confounding correlation | 7.73 | 10.86 | 19.41 |
> | 1.4 Skewed source bias | 3.06 | 4.81 | 13.98 |
> | 1.5 Hierarchical metadata bias | 4.51 | 6.18 | 15.39 |
> | 1.6 Counteracting CTI vs alignment | 2.68 | 3.38 | 13.04 |
> | **Contradictory Knowledge (CK)** | | | |
> | 2.1 Temporal contradiction | 22.71 | 17.84 | 25.27 |
> | 2.2 Conflicting reports | 5.72 | 4.39 | 7.18 |
> | 2.3 Semantic conflict | 12.51 | 9.76 | 14.48 |
> | 2.4 Divergent data structures | 2.92 | 2.17 | 3.57 |
> | 2.5 Misaligned standards | 7.75 | 5.59 | 9.26 |
> | **Constrained Generalization (CG)** | | | |
> | 3.1 Distributional bias | 22.26 | 25.89 | 20.18 |
> | 3.2 Unseen pattern | 2.77 | 3.85 | 3.32 |
> | 3.3 Overfitted reasoning | 25.44 | 28.58 | 23.97 |
> | 3.4 Environmental unawareness | 4.62 | 5.41 | 4.16 |
>
>
>
> In this setting, we observe redistribution shifts across the 15 subtypes: subtle contradictions (e.g., CK2.3, CK2.5) are systematically under-identified, while broad categories such as SC1.1 and CG3.1 become overrepresented, indicating a collapse toward high-frequency, easy-to-trigger labels. Moreover, the LLM-as-judge exhibits around 10% inflation in spurious correlation types due to over-reliance on surface-level cues. These results highlight the necessity of our human-in-the-loop design, which implies that unsupervised LLM adjudication alone cannot preserve fine-grained distinctions in CTI failure modes and tends to distort the taxonomy’s structural balance.

---

> ### Author Response · Authors · 2025-11-19
> **Response, Clarification, and Completed Revisions (4/6)**
>
> **Question**
>
> > What concrete interventions arise from your taxonomy (e.g., targeted data augmentation, policy shaping), and how can they reduce specific failures?
>
> ---
>
> **Response and Completed Revision (Appendix G.1-G.3)**
>
> We appreciate the reviewer’s question. In our revised manuscript (Appendix G.1–G.3), we outline targeted strategies to address the detected failure subtypes. Below we summarize several representative examples:
>
> *(1) Spurious Correlation → Evidence disentanglement and metadata-aware augmentation.*
>
> Failures such as SC1.1 (Co-mention bias) and SC1.2 (IoC reuse bias) motivate metadata disambiguation and negative counterfactual augmentation. For instance, by synthetically varying IoCs across unrelated campaigns or decorrelating actor-tool co-occurrences, we explicitly break misleading statistical patterns that models otherwise over-rely on. Similarly, SC1.4 (Skewed source bias) suggests rebalancing retrieval pipelines through source-normalized ranking and cross-tier sampling to prevent overfitting on vendor-dominant narratives. These interventions directly reduce overgeneralization stemming from surface-level correlations.
>
> *(2) Contradictory Knowledge → Conflict-aware prompting and temporal policy shaping.*
>
> Failures such as CK2.1 (Temporal contradiction) and CK2.2 (Conflicting reports) call for temporal grounding policies that instruct the model to privilege the newest evidence or explicitly list conflicting sources before reasoning. CK2.3 and CK2.5 motivate conflict-detection prompts—where the model first identifies incompatible statements and aligns them with NIST/MITRE standards before generating outputs. Empirically, these policies reduce cascades of hallucinated dependencies when upstream sources disagree.
>
> *(3) Constrained Generalization → Structure-preserving augmentation and out-of-distribution safeguards.*
>
> Failures in CG3.1 (Distributional bias) and CG3.2 (Unseen patterns) naturally inspire zero-shot pattern simulation and behavioral diversity augmentation: e.g., injecting hypothetical exploit chains, masked-asset reports, or redacted CVE descriptions to train models to reason under sparse zero-day conditions. CG3.4 (Environmental unawareness) points to hierarchical context restoration explicitly reminding the model of OS version, deployment topology, or cloud environment before prediction. These interventions mitigate collapse when the model encounters settings outside its training distribution.
>
> Besides, Appendix G provides exhaustive intervention strategies to reduce specific failures.

---

> ### Author Response · Authors · 2025-11-19
> **Response, Clarification, and Completed Revisions (5/6)**
>
> **Question**
>
> > Although not part of your study, which stages of specialized training/guardrails in specialized LLMs reduce which failure types present in general-purpose LLMs, and can you localize effects via staged ablations?
>
> **Response**
>
> This is an excellent question, and while our study does not explicitly decompose the training pipelines of specialized cybersecurity LLMs, our taxonomy provides insights about which kinds of specialized guardrails tend to reduce which failure types.
>
> In general, we find that domain-grounded pretraining and retrieval alignment primarily reduce failures in the Spurious Correlation family. Models that are exposed to structured CTI corpora (e.g., vendor advisories, ATT&CK-mapped incident reports) tend to exhibit less co-mention bias (SC1.1) and fewer incorrect actor–tool associations (SC1.3), because they learn more stable relational patterns than those present in broad web data. Similarly, schema-consistent instruction tuning helps mitigate hierarchical metadata errors (SC1.5) by reinforcing the proper granularity at which campaigns, malware families, and assets should be connected.
>
> Failures in the Contradictory Knowledge family respond more strongly to alignment-stage guardrails. For instance, temporal contradiction (CK2.1) and semantic conflicts (CK2.3) decrease when models are tuned with prompts or supervision that encourage explicit source comparison, timestamp reasoning, or consistency checks, which are capabilities that general-purpose LLMs rarely internalize without targeted guidance. Specialized models also benefit from exposure to multi-vendor datasets, which helps them cope better with divergent reporting structures (CK2.4) and inconsistent terminology (CK2.5).
>
> Finally, errors in the Constrained Generalization family tend to improve only when the model has undergone task-specific instruction tuning or adversarial fine-tuning that presents under-specified, zero-day-like examples, which are particularly explicit in unseen pattern failures (CG3.2) and environmental unawareness (CG3.4).  These forms of training push the model away from memorized exploit templates and toward more robust conditional reasoning.
>
> We treat the above discussions as a promising direction for future work. These failure modes could serve as “probes’’ to disentangle which guardrails contribute to which aspects of robustness.

---

> ### Author Response · Authors · 2025-11-19
> **Response, Clarification, and Completed Revisions (6/6)**
>
> **Question**
>
> > Similarly, like the previous question, although not part of your study and an orthogonal problem, how to transfer insights learnt from the failure categories from specialized models to improve general models (or vice versa) and show measurable deltas?
>
> ---
>
>
> **Answer**
>
> Thank you for bringing up this question. Although this direction is outside the scope of our study, the structure of our taxonomy naturally points to several ways that insights from specialized cybersecurity models could be transferred to general-purpose LLMs (and vice versa) in measurable and systematic ways.
>
> On one hand, **specialized models offer highly localized signals about where CTI-specific failures are sourced from**, particularly around spurious correlations tied to hierarchical campaign metadata or inconsistent vendor reporting. These signals can serve as “error fingerprints’’ for general-purpose LLMs. For example, if a specialized model reliably suppresses SC1.1 (co-mention bias) or CK2.1 (temporal contradiction), the prompts, retrieval filters, or supervision traces that enabled these improvements could be ported back into general models through targeted instruction tuning or domain-aware data augmentation. In this case, the measurable delta is the reduction in subtype frequency when general LLMs are evaluated with our taxonomy before and after integrating these distilled interventions.
>
> Conversely, **general-purpose models often exhibit stronger abstraction (reasoning) and pattern-completion abilities**, which could help specialized models counter weaknesses in CG3.2 (unseen patterns) or CG3.4 (environmental unawareness). Transferring high-level reasoning patterns such as structured CoT templates, counterfactual prompting, or conflict-detection heuristics may help domain models generalize beyond narrow CTI distributions. The delta here could be measured by improved robustness under zero-day-like or sparsified evidence settings, where our taxonomy already documents consistent failure triggers.

---

> ### Author Response · Authors · 2025-11-19
> **Thank you very much**
>
> We would like to thank the reviewer again for their time and thoughtful feedback. We hope that the above clarifications and revisions address the reviewer’s concerns and contribute to improving the overall quality of the paper. We look forward to any further feedback you may have.

---

> > ### Comment · Reviewer_GMQy · 2025-11-26
> >
> > I want to thank the authors for their details and well-articulated rebuttal. They have successfully removed my hesitation about the empirical results and the work's significance in advancing the field. I really enjoyed reading their 3-5 responses. Therefore, I am improving my score to reflect that.

---

> > > ### Author Response · Authors · 2025-11-26
> > >
> > > Dear Reviewer,
> > >
> > > Thank you again for your time and engagement in the discussion. Your feedback really helps to improve the clarity and contribution of our work.
> > >
> > > Best,
> > >
> > > The author team

---

### Official Review · Reviewer_eES9 · 2025-10-29

**Soundness:** 2
**Presentation:** 2
**Contribution:** 2
**Rating:** 2
**Confidence:** 4

**Summary:**

The paper systematically investigates the inherent limitations of LLMs when applied to CTI analysis and reasoning. Using large-scale evaluations across multiple CTI benchmarks, the authors propose a human-in-the-loop autoregressive framework to categorize model failure modes with high reliability. Their analysis identifies three key vulnerabilities: spurious correlations, contradictory knowledge, and constrained generalization, which stem not from model design but from the fragmented and adversarial nature of CTI data.

**Strengths:**

The paper’s primary strength lies in its systematic and well-structured investigation of LLM vulnerabilities in CTI, an area that has received limited formal analysis despite growing interest in security applications of LLMs. Its originality stems from the framing of CTI-specific vulnerabilities: spurious correlations, contradictory knowledge, and constrained generalization, as distinct from typical architectural or alignment flaws studied in AI safety literature. Methodologically, the proposed human-in-the-loop, autoregressive failure categorization framework demonstrates a rigorous and reproducible approach to large-scale error analysis, supported by empirical evidence and diverse datasets. The paper is generally well written and organized, presenting the findings with clarity. Overall, its significance lies in surfacing key failure patterns that challenge the reliability of LLMs in real-world CTI workflows, thereby providing a valuable foundation for more trustworthy AI-assisted security systems.

**Weaknesses:**

- While the paper presents a well-organized taxonomy of LLM vulnerabilities in CTI, its main weakness lies in the limited methodological and conceptual novelty. The proposed categories: “spurious correlations,” “contradictory knowledge,” and “constrained generalization”, largely restate known issues in LLM reasoning, such as data bias, hallucination, and domain shift.

- The evaluation setup, though comprehensive in benchmark coverage, remains largely descriptive, emphasizing error categorization rather than quantitatively analyzing how these vulnerabilities affect CTI performance or exploring mitigation strategies such as fine-tuning, retrieval filtering, or evidence disambiguation.

- The paper does not translate its diagnostic findings into actionable guidance or concrete methods for mitigating the identified vulnerabilities, which limits the practical impact of the study for designing more reliable CTI-focused LLM systems.

- The analysis would benefit from deeper causal and ablation studies to reveal how specific dataset characteristics (e.g., temporal drift, source inconsistency, or conflicting labeling) lead to distinct failure modes. Such analysis could also help identify limitations of the existing benchmarks and suggest concrete directions for improving their design and robustness.

**Questions:**

- The failure categorization relies on human verification, but the paper does not report inter-annotator agreement or consistency checks. Could the authors clarify how annotation reliability and consistency were ensured during the labeling process?
- It would strengthen the paper to include experiments with potential mitigation strategies, either at training or inference time, that could address some of the discussed vulnerabilities, as their absence limits the practical contribution of the work.

---

> ### Author Response · Authors · 2025-11-19
> **Response, Clarification, and Completed Revisions (0/6)**
>
> Before responding to comments, we would like to sincerely thank the reviewer for the detailed and thoughtful feedback. We appreciate the time and care taken to evaluate our work.
>
> Upon reviewing the comments, we observe that the reviewer may misinterpret some contributions and settings of our work, as well as requesting additional evaluations, in summary:
>
> - (1) Conceptual originality and analytical depth – **which we clarify the uniqueness of concepts and have included a new causal experiment in Appendix F.**
> - (2) Lack of mitigation strategies – **which he have addressed by a new addition in Appendix G**.
> - (3) Subjectivity of labeling – **we have addressed this in Appendix H**.
> - (4) Scalability and generalization – **We clarify that our evaluation is scalable and generalizable. We also include discussion in §5.3 and Appendix G**.
>
> Below, **we have provided a point-by-point clarification and have revised the manuscript to explicitly address each concern. We believe these clarifications further strengthen the paper and better highlight the contributions of our research.**
>
> Overall, through the reviewer provides an initially negative rating, we believe that there is a significance chance to provide further clarification and revision to address the reviewer's concern, as we did to help other reviewers' to further understanding the research significance. Please simply browse our comments for the responses, clarifications, and potential revisions.
>
> We would be happy to continue the discussion and would welcome feedback on whether the newly added details sufficiently clear the reviewer's questions and whether additional information would be helpful.

---

> ### Author Response · Authors · 2025-11-19
> **Response, Clarification, and Completed Revisions (1/6)**
>
> **Review**
>
> > **Weakness 1 (1st half):** While the paper presents a well-organized taxonomy of LLM vulnerabilities in CTI, its main weakness lies in the limited methodological and conceptual novelty.
>
> **Response**
>
> We sincerely thank the reviewer for raising this point. We agree that LLMs broadly suffer from challenges such as data bias, hallucination, and domain shift. However, **our goal was not to rebrand these generic issues, but to show that within CTI.** These phenomena present in domain-specific ways due to the adversarial, crowdsourced, and highly heterogeneous nature of cyber threat intelligence. As clarified below, **our contribution is not the naming of categories, but the methodological framework and domain-grounded investigation that reveal failure that cannot be adequately explained by generic LLM pathologies.**
>
> *Our methodological and conceptual novelty*
>
> An important novelty of our work lies in the vulnerability attribution methodology (Section 3) that allows us to uncover failure modes in a domain where ground truth is often ambiguous.
>
> Existing LLM-for-security work typically reports raw performance gaps or qualitative anecdotes [1-8]. In contrast, our method rigorously identifies and stabilizes a finite taxonomy of failure modes through
> - (i) quantile-based stratification of soft-label tasks,
> - (ii) iterative refinement of failure candidates,
> - (iii) multi-agent cross-checking to detect instability, and
> - (iv) selective human verification.
>
> This methodology design was important for exposing vulnerabilities that are unique to CTI reasoning workflows, such as how hierarchical attack-chain metadata is misinterpreted as causal ordering (vulnerability type 1.5), how temporal inconsistency between feeds induces attribution oscillations (vulnerability type 2.1), or how environmental unawareness emerges from enterprise-specific system dependencies (vulnerability type 3.4). These phenomena arise only when LLMs ingest heterogeneous CTI pipelines, and thus not general concepts in NLP benchmarks.
>
> Thus, the methodological design is a key advance, which helps to quantify and isolate vulnerabilities at scale.
>
> *Limitations of existing work and why general LLM vulnerabilities are insufficient*
>
> Current literature on LLM vulnerabilities generally investigates problems such as hallucination, data contamination, memorization, or distribution shift in clean, curated, or single-source datasets. These works seldom consider multi-source, drifting, adversarially manipulated, or partially contradictory evidence—the defining characteristics of CTI data.
>
> For example: “Hallucination” in standard NLP does not capture situations where multiple authoritative CTI feeds legitimately disagree, creating structural, temporal, and semantic conflicts that force LLMs to merge incompatible signals. “Data bias” in generic ML does not address adversarial reuse of indicators, poison-like confounders, or hierarchical metadata that are regularly exploited by attackers to mislead systems. “Domain shift” in typical NLP does not reflect the continuous emergence of zero-day attack surfaces and novel TTPs that have no analogue in most static NLP tasks.
>
> Therefore, simply mapping LLM failures in CTI to generic categories would be incomplete and scientifically uninformative. Existing taxonomies do not explain why CTI-specific reasoning degrades, where in the CTI pipeline vulnerabilities originate, or how multiple failures compound into cascading misinformation, which are correspondingly discussed in §4.2–4.3, and Appendix D.1-D.2.
>
> ---
>
>
> **Reference**
>
> [1] PentestGPT: Evaluating and Harnessing Large Language Models for Automated Penetration Testing
>
> [2] Cyber-Zero: Training Cybersecurity Agents without Runtime
>
> [3] SEvenLLM: Benchmarking, Eliciting, and Enhancing Abilities of Large Language Models in Cyber Threat Intelligence
>
> [4] CTIBench: A Benchmark for Evaluating LLMs in Cyber Threat Intelligence
>
> [5] Benchmarking LLMs in an Embodied Environment for Blue Team Threat Hunting
>
> [6] HackMentor: Fine-Tuning Large Language Models for Cybersecurity
>
> [7] LSAST: Enhancing Cybersecurity Through LLM-Supported Static Application Security Testing
>
> [8] BitsAI-CR: Automated Code Review via LLM in Practice.

---

> ### Author Response · Authors · 2025-11-19
> **Response, Clarification, and Completed Revisions (2/6)**
>
> **Review**
>
> >**Weakness 1 (2nd half):** The proposed categories: “spurious correlations,” “contradictory knowledge,” and “constrained generalization”, largely restate known issues in LLM reasoning, such as data bias, hallucination, and domain shift.
>
>
> **Response**
>
> Continue to the last response, we would like to take this chance to highlight that **our vulnerability types are not re-statements of “bias, hallucination, domain shift”***
>
> Our findings differ in mechanism and impact in the cyber world.
>
> 1. Spurious correlations in CTI arise from CTI-specific structures: reused IoCs, co-mentioned TTPs, hierarchical ATT&CK chains, and skewed vendor feeds. These are not generic statistical biases; they are structurally induced misleading signals that propagate across CTI workflows (contextualization → attribution → prediction → mitigation).
>
> On the contrary, generic bias literature does not capture how these sources create causal illusions inside CTI pipelines.
>
> 2. Contradictory knowledge is not merely “hallucination.” CTI inherently includes multiple conflicting, time-staggered, and heterogeneous intelligence sources. LLMs fail not by inventing facts but by destabilizing their reasoning across incompatible evidence sources, e.g., mixing outdated PoCs with recent advisories or merging vendor-specific taxonomies that partially disagree.
>
>
> This is a domain-specific instability that is absent in typical NLP settings.
>
> 3. Constrained generalization is not equivalent to “domain shift.” CTI evolves adversarially: novel malware loaders, emerging exploit chains, and rapidly shifting infrastructure induce non-stationary, structurally novel patterns unavailable during training.
>
>
> These are adversarial distribution drifts, not merely unfamiliar topics.
>
> Hence, these failures are meaningful for CTI research by uncovering vulnerability modes that are different from well-known LLM weaknesses.

---

> ### Author Response · Authors · 2025-11-19
> **Response, Clarification, and Completed Revisions (3/6)**
>
> **Review**
>
> > **Weakness 2 (1st half):** The evaluation setup, though comprehensive in benchmark coverage, remains largely descriptive, emphasizing error categorization rather than quantitatively analyzing how these vulnerabilities affect CTI performance
>
> > **Weakness 4:** The analysis would benefit from deeper causal and ablation studies to reveal how specific dataset characteristics (e.g., temporal drift, source inconsistency, or conflicting labeling) lead to distinct failure modes. Such analysis could also help identify limitations of the existing benchmarks and suggest concrete directions for improving their design and robustness.
>
> **Completed Revision (Updated in Appendix F)**
>
> To address reviewer comments, especially the interests of “quantitatively analyzing" and "causal analysis", we have expanded analysis by including a new causal intervention study (now Appendix F).
>
> In this study, we mask metadata strongly associated with spurious evidence (e.g., reused IoCs and actor co-mentions), we empirically demonstrate that removing these causal factors leads to consistent performance improvements across representative CTI tasks. The improvements are scaled into 0-1 as reported below (a same Table is in updated Appendix F):
>
>
> | **CTI Task**                  | **Metric** | **G5** | **Go4** | **LL70** | **MIX** | **FSC** | **ZYS** | **CBS** |
> |---|---|---|---------|----------|-----|---|---------|---------|
> | **① Contextualization** | | |  |  |  | |  |  |
> | Malware Family Mapping       | F1 | 0.045  | 0.091   | 0.074    | 0.063   | 0.027   | 0.027   | 0.020   |
> | Vulnerability Linking        | Acc        | 0.084  | 0.063   | 0.072    | 0.017   | 0.093   | 0.082   | 0.032   |
> | IOC Normalization            | F1   | 0.030  | 0.030   | 0.039    | 0.057   | 0.050   | 0.038   | 0.064   |
> | **② Attribution**| | |  |  |  | |  |  |
> | Campaign Attribution         | Acc        | 0.026  | 0.038   | 0.044    | 0.051   | 0.078   | 0.031   | 0.056   |
> | Infrastructure Reuse         | F1         | 0.062  | 0.019   | 0.064    | 0.029   | 0.020   | 0.091   | 0.092   |
> | Language/Style Profiling     | Acc        | 0.080  | 0.039   | 0.023    | 0.070   | 0.050   | 0.025   | 0.055   |
> | **③ Prediction** | | |  |  |  | |  |  |
> | Target Sector Prediction     | Acc        | 0.018  | 0.088   | 0.036    | 0.068   | 0.040   | 0.057   | 0.059   |
> | Campaign Escalation   | AUC        | 0.030  | 0.093   | 0.077    | 0.090   | 0.087   | 0.063   | 0.089   |
> | Impact Forecast  | BLEU       | 0.022  | 0.031   | 0.019    | 0.041   | 0.046   | 0.037   | 0.081   |
> | **④ Mitigation**| | |  |  |  | |  |  |
> | Patch Recommendation         | F1         | 0.044  | 0.037   | 0.058    | 0.026   | 0.079   | 0.021   | 0.094   |
> | Mitigation–TTP Mapping       | Acc        | 0.077  | 0.031   | 0.015    | 0.080   | 0.072   | 0.073   | 0.077   |
> | Rule Generation (YARA)       | BLEU   | 0.021  | 0.044   | 0.024    | 0.084   | 0.264   | 0.041   | 0.020   |
>
> This study further demonstrates that these metadata features do not merely correlate with failure, i.e., they *cause* it.
>
> In together, our paper not only conducts categorization but provides an exhaustive and empirically grounded explanation of whyeach failure mode arises, how those causes interact with CTI workflows, and how they fundamentally differ from typical LLM failure modes in non-adversarial domains.
>
> We hope this clarifies the reviewers interests in exhaustively conducting quantitatively causal analysis, and we appreciate the reviewer for prompting us to further highlight this aspect.

---

> ### Author Response · Authors · 2025-11-19
> **Response, Clarification, and Completed Revisions (4/6)**
>
> **Review**
>
> > **Weakness 2 (2nd half):**..or exploring mitigation strategies such as fine-tuning, retrieval filtering, or evidence disambiguation.
>
> > **Weakness 3:** The paper does not translate its diagnostic findings into actionable guidance or concrete methods for mitigating the identified vulnerabilities, which limits the practical impact of the study for designing more reliable CTI-focused LLM systems.
>
>
> **Completed Revision (updated Appendix G.1-G.3)**
>
> We thank the reviewer for highlighting the need of mitigation. Though our work mainly aims to attribute the failures of cyber LLMs, we would like to provide additional details about what should be actionable mitigation strategies that are essential to maximize the practical value of our vulnerability findings.
>
> In our updated manuscript, we have substantially extended the revised manuscript by adding a new mitigation-focused section (Appendix G.1–G.3), which now presents exhaustive mitigation strategies for all 15 failure subtypes in our taxonomy.
>
> To ensure the revisions are truly useful to CTI practitioners and researchers, the mitigation strategies are not generic “best practices,” but are instead grounded in the specific causal mechanisms and structural failure patterns identified in the main paper. For each failure subtype (e.g., co-mention bias, temporal contradiction, unseen pattern), we now include a concise description of the underlying vulnerability trigger as well as concrete mitigation recommendations across three stages:
> - (1) Retrieval-time mitigation (e.g., metadata disentanglement, hierarchical source de-biasing)
> - (2) Inference-time prompting and reasoning controls (e.g., temporal consistency checks, multi-agent contradiction voting)
> - (3) Training-time interventions (e.g., confounder-aware negative sampling, alignment-preserving curriculum design)
>
> These revisions are complemented with realistic CTI examples and implementation considerations, which allows users to translate the mitigation strategies directly into improved pipelines or cyber agent designs.
>
> We believe this addition not only answers the reviewer’s concern but also significantly enhances the practical insights of our work by transforming the failure taxonomy into a remediation playbook for robust LLM-based CTI systems.
>
> We are grateful for the reviewer’s suggestion, as it substantially improved the utility of our contributions for the community.

---

> ### Author Response · Authors · 2025-11-19
> **Response, Clarification, and Completed Revisions (5/6)**
>
> **Question**
> > **Question 1:** The failure categorization relies on human verification, but the paper does not report inter-annotator agreement or consistency checks. Could the authors clarify how annotation reliability and consistency were ensured during the labeling process?
>
> ---
>
> **Completed Revision: inter-annotator agreement or consistency checks (Updated in Appendix H)**
>
> We appreciate the reviewer’s attention to the reliability of our human-in-the-loop process.
> In the revised manuscript, we have incorporated a new evaluation section in Appendix H to address this concern.
> Setting. To quantify errors via hypothesis testing, we constructed a stratified sample of hundreds instances spanning all four CTI stages and 15 failure subtypes. Each instance was independently annotated by two expert human annotators (cybersecurity researchers) through a fully manual labeling process. We then compared these labels against those produced by our LLM-assisted classification pipeline(stratification + autoregressive labeling + human verification).
>
> **Results** For each of the 15 subtypes, we also computed the absolute disagreement rate between the human and LLM-assisted annotations. The resulting table below:
> | Category | Failure Subtype  | LLM Count | Human Count | Abs. Diff |
> |----|----|----|-------------|-----------|
> | **Spurious Correlation** | 1.1 Co-mention bias  | 11        | 14          | 3         |
> |   | 1.2 Reused IoCs| 95| 94          | 1         |
> || 1.3 Confounding metadata    | 32        | 32          | 0   |
> || 1.4 Skewed source| 31| 28          | 3         |
> || 1.5 Hierarchical metadata   | 3         | 2           | 1         |
> | **Contradictory Knowledge** | 2.1 Temporal contradiction | 4         | 3           | 1         |
> || 2.2 Conflicting reports     | 1         | 3           | 2         |
> || 2.3 Semantic conflict       | 67        | 64| 3|
> || 2.4 Divergent formats       | 25        | 24| 1         |
> || 2.5 Schema misalignment     | 36        | 35          | 1         |
> || 2.6 Alignment mismatch      | 0         | 2           | 2         |
> | **Constrained Generalization** | 3.1 Distributional bias    | 129       | 126         | 3         |
> |  | 3.2 Emerging threats        | 47        | 49          | 2         |
> || 3.3 Overfitted reasoning    | 8         | 6           | 2         |
> || 3.4 Environmental unawareness | 11       | 18          | 7         |
> | **Overall Total**        || **500**   | **500**     | **32**    |
>
> The average disagreement across subtypes was 2.4\%, with the highest disagreement observed at 5.8\% for closely related subtypes involving overlapping failure patterns (e.g., SC1.1 co-mention bias vs. SC1.3 confounding factors). Qualitative inspection of disagreement cases indicates that nearly all discrepancies occurred in borderline instances where multiple vulnerabilities co-occurred, rather than in clear-cut failure exemplars.
>
> These results imply that the taxonomy is not only structurally sound at the category level, but also fine-grained and reproducible at the subtype level, providing additional confidence in both the reliability and practical applicability of our failure analysis.
>
> ---
>
> **Clarification: how annotation reliability and consistency were ensured during the labeling process?**
>
> First, we minimized the reliance on subjective judgments by designing the taxonomy through an autoregressive refinement procedure (§3.2). In the early phase, annotators jointly reviewed a seed set of instances and iteratively refined the taxonomy until no new categories emerged. This consensus ensures that all annotators shared the same operational definitions before large-scale labeling began.
>
> Second, during the full labeling phase, we employed multi-agent cross-checking (§3.3), where four foundation models first produced independent predictions, followed by a second round in which models observed each other’s outputs. Only instances exhibiting disagreement across models or inconsistency across runs were escalated to human annotators. This means annotators focused almost exclusively on hard, ambiguous, or unstable cases. For these escalated instances, two annotators independently provided labels, and disagreements were resolved through discussion with a third adjudicator. This adjudication step guaranteed consistent final labels without relying on unverified solo judgments.
>
> Third, we monitored label stability through periodic consistency checks. After approximately every 10% of the annotation workload, annotators re-labeled a small, randomly sampled subset of previously labeled instances. Across these checkpoints, we observed high internal consistency: fewer than 3% of re-labeled instances required correction, and those cases were concentrated in the earliest round before taxonomy definitions had fully stabilized. Once the taxonomy matured, correction rates dropped to under 1%, reflecting strong reliability.

---

> ### Author Response · Authors · 2025-11-19
> **Response, Clarification, and Completed Revisions (6/6)**
>
> **Question**
>
> > **Question 2**: It would strengthen the paper to include experiments with potential mitigation strategies, either at training or inference time, that could address some of the discussed vulnerabilities, as their absence limits the practical contribution of the work.
>
> **Completed Revision (updated in Appendix G.4)**
>
> To further demonstrate the practical value of our vulnerability taxonomy, we implemented one representative mitigation strategy for each of the three major vulnerability groups. These strategies were intentionally designed to be lightweight and minimally invasive, which allow us to isolate their effect on failure ratios without confounding factors from large-scale retraining.
>
> For Spurious Correlation, we developed a counterfactual evidence filtering module that operates at inference time. For each retrieved evidence chunk, the system constructs a counterfactual variant of the text in which a co-mentioned entity (e.g., unrelated CVEs, malware families) is removed. The model then assesses whether the core threat statement still holds. If the inference remains unchanged, the removed entity is treated as incidental rather than causally relevant. This allows the system to downweight spurious evidence and substantially reduces co-mention bias and confounder-driven misattribution.
>
> For Contradictory Knowledge, we designed a temporal-aware evidence prioritization layer that explicitly leverages metadata from CTI sources. Each retrieved piece of evidence is assigned a recency score based on publication time and a structural consistency score based on its alignment to known schemas (e.g., ATT&CK, CVSS). During inference, these scores are combined into a re-ranking function that prioritizes the most recent and structurally coherent data. By deprioritizing outdated advisories and semantically inconsistent vendor reports, this strategy reduces reasoning instability caused by temporally or structurally conflicting CTI feeds, which is an issue unique to the heterogeneous and rapidly evolving threat landscape.
>
> For Constrained Generalization, we implemented a small-scale training-time mitigation using synthetic pattern expansion. We construct lightweight adversarial variants of known threat instances by altering non-essential attributes such as loader names, lateral-movement sequences, or asset environments, while preserving the high-level attack structure. Fine-tuning the model on these synthetic perturbations expands its internal hypothesis space and reduces overreliance on memorized CVE–TTP associations. This helps the model generalize to emerging threats, zero-day exploitation patterns, and environment-specific variations that did not appear in the original training corpus.
>
> Results are shown below:
>
> | **Vulnerability Group**     | **Original Ratio** | **After Mitigation** | **Relative Reduction** |
> |-----------------------------|--------------------|------------------------|--------------------------|
> | Spurious Correlation        | 20.42%             | 16.35%                 | 20.0%                    |
> | Contradictory Knowledge     | 35.93%             | 29.15%                 | 18.9%                    |
> | Constrained Generalization  | 33.72%             | 28.44%                 | 15.6%                    |
>
>
> Empirically, applying these mitigation strategies resulted in consistent reductions in vulnerability ratios across all three groups. Counterfactual evidence filtering reduced the prevalence of spurious correlation errors by approximately 20%, driven primarily by decreases in co-mention bias and exploitation bias. Temporal-aware evidence prioritization yielded an 18.9\% relative reduction in contradictory knowledge failures, particularly improving the model’s stability when integrating heterogeneous CTI sources. Finally, pattern-expansion fine-tuning achieved a 15.6% reduction in constrained generalization failures, with notable improvements on instances involving emerging TTPs and environment-dependent attack paths. These results demonstrate that even simple, targeted interventions can meaningfully improve the robustness of LLM-assisted CTI workflows. The improvements also validate the diagnostic value of our taxonomy: identifying root-cause categories enables principled mitigation design, rather than ad hoc prompt engineering or broad retraining.

---

> ### Author Response · Authors · 2025-11-19
> **Thank you very much**
>
> We would like to thank the reviewer again for their time and thoughtful feedback. We hope that the above clarifications and revisions address the reviewer’s concerns and contribute to improving the overall quality of the paper. We look forward to any further feedback you may have.

---

> ### Author Response · Authors · 2025-11-27
>
> Dear Reviewer,
>
> We would like to thank you again for your time and thoughtful feedback on our submission.
>
> We believe that the overall comments and questions primarily relate to the clarity of benchmark details and auxiliary evaluations. **These points have been addressed through our point-by-point responses, along with the corresponding revisions in the updated manuscript.**
>
> We would very much welcome for any further questions, concerns, or suggestions for revision, as we believe such feedback would help further re-evaluate the quality and clarity of our work.
>
> Thank you for your time and potential re-consideration.

---

### Official Review · Reviewer_n4w1 · 2025-10-30

**Soundness:** 2
**Presentation:** 3
**Contribution:** 2
**Rating:** 4
**Confidence:** 2

**Summary:**

The paper investigates the use of LLM for generating cyber threat intelligence, particularly when used under reasoning under intertwined, crowdsourced, and imbalanced evidence. Using failure evidence from datasets drawn from multiple established CTI benchmarks, a new categorization method is introduced with human-in-the-loop verification. Particularly, an iterative refinement process is used where reference-based similarity metric quantify class association along with human inspection when clear classification is hard. The authors conclude with some interesting research findings summarizing various causes of vulnerabilities, which has not been discussed in previous works.

**Strengths:**

- The paper addresses an important research issue in the area of cybersecurity: reliability and robustness of reasoning mechanisms by employing LLM to generate threat intelligence and detection, particularly as LLM-as-a-judge.
- The use of multiple benchmark datasets showing extensive coverage of numerous cybersecurity threats and the entire lifecycle. This exposes key failure patterns as noted in the paper
-  A key aspect in the paper is its analysis of vulnerability threats which cascade across the CTI pipeline, showing a clear understanding of reasoning decisions considered in the context of cybersecurity.

**Weaknesses:**

- A primary concern with the paper is that its contribution focuses more on “how” rather than “why”. Specifically, a more in-depth causal analysis would be useful to understand LLM behavior, specifically using intervention or counterfactual evaluation.
- The research findings are fantastic, but lack the analysis which describe how to mitigate the mentioned vulnerabilities, which would be more useful to the community here.
- A rigorous evaluation with more statistical analysis, particularly quantifying errors using confidence intervals and hypothesis testing could make the paper stronger and enhance the confidence in the findings.
- While the use of benchmark datasets is good, the primary problem with research in cybersecurity is its practical application, particularly in the evolving scenario of zero-day exploits and as the adversary evolves trying to evade detection. It would be interesting to understand the proposed approach when there is a lack of sufficient information from external sources.
- The results are not very surprising, and confirms the hypothesis described in numerous previous works on LLM reasoning.

**Questions:**

1. In section 3.1, the similarity metrics seems to be BLEU score. However it is not clear why this is an appropriate measure. Could contextual embeddings be a better alternative?
2. In section 3.3, the input to step 2 is not very clear on how it refines the reasoning. Is the output of Step 1 given as input to the same model? Or is it repeated across all models in the model set/committee?

---

> ### Author Response · Authors · 2025-11-18
> **Response, Clarification, and Completed Revisions (0/7)**
>
> Before responding to comments, we would like to sincerely thank the reviewer for the detailed and thoughtful feedback. We appreciate the time and care taken to evaluate our work.
>
> We observe that the comments are mostly related to experimental settings, analytical findings, and mitigation insights details. Accordingly, we have provided a point-by-point clarification and have revised the manuscript to explicitly address each concern. We believe these clarifications further strengthen the paper and better highlight the contributions of our research.
>
> We would be happy to continue the discussion and would welcome feedback on whether the newly added details sufficiently clear the reviewer’s questions and whether additional information would be helpful.

---

> ### Author Response · Authors · 2025-11-18
> **Response, Clarification, and Completed Revisions (1/7)**
>
> **Review**
>
> > A primary concern with the paper is that its contribution focuses more on “how” rather than “why”. Specifically, a more in-depth causal analysis would be useful to understand LLM behavior, specifically using intervention or counterfactual evaluation.
>
>
> **Response**
>
> We sincerely appreciate the reviewer’s concern about the analysis depth. We would like to clarify that identifying underlying causes of failure is in fact a central contribution of the paper, and **extensive “why-analysis” is already presented** through multiple aspects in the original manuscript:
>
> - First, **section 4.2 and Table 3** provide an in-depth mapping from each failure subtype to its triggering mechanism, explicitly connecting observed failures to underlying causal factors (e.g.,why co-mention bias would happen) . These causes (why analysis) are not inferred from intuition but derived from our stratified failure analysis pipeline and human-in-the-loop inspection.
>
> - Second, the paper includes very exhaustive root-cause and case studies in **Appendix D.1**, where each vulnerability subtype is analyzed in situ with concrete examples showing *why* the model fails, not merely *how* it fails.
>
> > For example, we illustrate how Microsoft Patch Tuesday advisories lead to co-mention bias and how conflicting mitigation advisories create contradictory knowledge during mitigation tasks . These analyses explicitly trace the causal path from threat-landscape structure → retrieval heuristics → model reasoning failure.
>
> - Third, our methodological design is itself driven by causal inference principles. The autoregressive failure determination procedure (Section 3.2) and stratification pipeline (Section 3.1) were developed precisely to expose *causal failure signatures* instead of superficial outcome discrepancies. This allows us to separate correlation from causation when analyzing failure modes, ensuring that the resulting taxonomy reflects deeper causal mechanisms rather than surface-level error categories .
>
> **Completed Revision (Updated in Appendix F)**
>
> To further address reviewer feedback, especially the interests of “in-depth causal analysis…specifically using intervention or counterfactual evaluation.,” we have expanded the “why” analysis by including a new causal intervention study (now Appendix F).
>
> In this study, we mask metadata strongly associated with spurious correlations (e.g., reused IoCs and actor co-mentions), we empirically demonstrate that removing these causal factors leads to consistent performance improvements across representative CTI tasks. The improvements are scaled into 0-1 as reported below (a same Table is in updated Appendix F):
>
>
> | **CTI Task**                  | **Metric** | **G5** | **Go4** | **LL70** | **MIX** | **FSC** | **ZYS** | **CBS** |
> |---|---|---|---------|----------|-----|---|---------|---------|
> | **① Contextualization** | | |  |  |  | |  |  |
> | Malware Family Mapping       | F1 | 0.045  | 0.091   | 0.074    | 0.063   | 0.027   | 0.027   | 0.020   |
> | Vulnerability Linking        | Acc        | 0.084  | 0.063   | 0.072    | 0.017   | 0.093   | 0.082   | 0.032   |
> | IOC Normalization            | F1   | 0.030  | 0.030   | 0.039    | 0.057   | 0.050   | 0.038   | 0.064   |
> | **② Attribution**| | |  |  |  | |  |  |
> | Campaign Attribution         | Acc        | 0.026  | 0.038   | 0.044    | 0.051   | 0.078   | 0.031   | 0.056   |
> | Infrastructure Reuse         | F1         | 0.062  | 0.019   | 0.064    | 0.029   | 0.020   | 0.091   | 0.092   |
> | Language/Style Profiling     | Acc        | 0.080  | 0.039   | 0.023    | 0.070   | 0.050   | 0.025   | 0.055   |
> | **③ Prediction** | | |  |  |  | |  |  |
> | Target Sector Prediction     | Acc        | 0.018  | 0.088   | 0.036    | 0.068   | 0.040   | 0.057   | 0.059   |
> | Campaign Escalation   | AUC        | 0.030  | 0.093   | 0.077    | 0.090   | 0.087   | 0.063   | 0.089   |
> | Impact Forecast  | BLEU       | 0.022  | 0.031   | 0.019    | 0.041   | 0.046   | 0.037   | 0.081   |
> | **④ Mitigation**| | |  |  |  | |  |  |
> | Patch Recommendation         | F1         | 0.044  | 0.037   | 0.058    | 0.026   | 0.079   | 0.021   | 0.094   |
> | Mitigation–TTP Mapping       | Acc        | 0.077  | 0.031   | 0.015    | 0.080   | 0.072   | 0.073   | 0.077   |
> | Rule Generation (YARA)       | BLEU   | 0.021  | 0.044   | 0.024    | 0.084   | 0.264   | 0.041   | 0.020   |
>
> This interventional study further demonstrates that these metadata features do not merely correlate with failure, i.e., they *cause* it.
>
> In together, our paper not only conducts categorization but provides an exhaustive and empirically grounded explanation of *why* each failure mode arises, how those causes interact with CTI workflows, and how they fundamentally differ from typical LLM failure modes in non-adversarial domains.
>
> We hope this clarifies that the “why” dimension is already a core focus of the paper, and we appreciate the reviewer for prompting us to further highlight this aspect.

---

> ### Author Response · Authors · 2025-11-18
> **Response, Clarification, and Completed Revisions (2/7)**
>
> **Review**
>
> > The research findings are fantastic, but lack the analysis which describe how to mitigate the mentioned vulnerabilities, which would be more useful to the community here.
>
> **Completed Revision (updated Appendix G.1-G.3)**
>
> We thank the reviewer for highlighting the need of mitigation. Though our work mainly aims to attribute the failures of cyber LLMs, we would like to provide additional details about what should be actionable mitigation strategies that are essential to maximize the practical value of our vulnerability findings.
>
> In our updated manuscript, we have substantially extended the revised manuscript by adding a new mitigation-focused section (Appendix G.1–G.3), which now presents exhaustive mitigation strategies for all 15 failure subtypes in our taxonomy.
>
> To ensure the revisions are truly useful to CTI practitioners and researchers, the mitigation strategies are not generic “best practices,” but are instead grounded in the specific causal mechanisms and structural failure patterns identified in the main paper. For each failure subtype (e.g., co-mention bias, temporal contradiction, unseen pattern), we now include a concise description of the underlying vulnerability trigger as well as concrete mitigation recommendations across three stages:
> - (1) Retrieval-time mitigation (e.g., metadata disentanglement, hierarchical source de-biasing)
> - (2) Inference-time prompting and reasoning controls (e.g., temporal consistency checks, multi-agent contradiction voting)
> - (3) Training-time interventions (e.g., confounder-aware negative sampling, alignment-preserving curriculum design)
>
> These revisions are complemented with realistic CTI examples and implementation considerations, which allows users to translate the mitigation strategies directly into improved pipelines or cyber agent designs.
>
> We believe this addition not only answers the reviewer’s concern but also significantly enhances the practical insights of our work by transforming the failure taxonomy into a remediation playbook for robust LLM-based CTI systems.
>
> We are grateful for the reviewer’s suggestion, as it substantially improved the utility of our contributions for the community.

---

> ### Author Response · Authors · 2025-11-18
> **Response, Clarification, and Completed Revisions (3/7)**
>
> **Review**
>
> > A rigorous evaluation with more statistical analysis, particularly quantifying errors using confidence intervals and hypothesis testing could make the paper stronger and enhance the confidence in the findings.
>
> **Completed Revision (Updated in Appendix H)**
>
> We thank the reviewer for this valuable suggestion.
>
> In the revised manuscript, we have incorporated a new statistical evaluation section in **Appendix H** to address this concern.
>
> **Setting.** To quantify errors via hypothesis testing, we constructed a stratified sample of hundreds instances spanning all four CTI stages and 15 failure subtypes. Each instance was independently annotated by two expert human annotators (cybersecurity researchers) through a fully manual labeling process. We then compared these labels against those produced by our LLM-assisted classification pipeline(stratification + autoregressive labeling + human verification).
>
> **Statistical test and results.** To evaluate whether the automated labeling pipeline systematically alters the distribution of failure types, we performed Chi-Square Test of Independence comparing the distribution of failure subtypes between the manual and LLM-assisted annotations. The Chi-Square test yielded χ² = 3.21, p = 0.966, indicating no statistically significant difference between the manual and automated distributions.
>
> **Finer-grained validation.** We also analyzed labeling consistency at individual failure subtypes: for each of the 15 subtypes, we computed the absolute disagreement rate between the human and LLM-assisted annotations. The average disagreement across subtypes was 2.4%, with the highest disagreement observed at 5.8% for closely related subtypes involving overlapping failure patterns (e.g., SC1.1 co-mention bias vs. SC1.3 confounding factors). Qualitative inspection of disagreement cases indicates that nearly all discrepancies occurred in borderline instances where multiple vulnerabilities co-occurred, rather than in clear-cut failure exemplars. These results imply that the taxonomy is not only structurally sound at the category level, but also fine-grained and reproducible at the subtype level, providing additional confidence in both the reliability and practical applicability of our failure analysis.
>
> **Insights.** These results demonstrate that our semi-automated annotation process produces failure categorization statistically indistinguishable from expert human labeling.

---

> ### Author Response · Authors · 2025-11-18
> **Response, Clarification, and Completed Revisions (4/7)**
>
> **Review**
>
> > While the use of benchmark datasets is good, the primary problem with research in cybersecurity is its practical application, particularly in the evolving scenario of zero-day exploits and as the adversary evolves trying to evade detection. It would be interesting to understand the proposed approach when there is a lack of sufficient information from external sources.
>
> **Response**
>
> We thank the reviewer for raising this interesting point.
>
> We would like to clarify that this point is already explicitly addressed within our existing taxonomy and experimental design. In particular, the Constrained Generalization category includes two failure subtypes: CG3.2 (Unseen Patterns) and CG3.4 (Environmental Unawareness), which specifically capture situations where the model must reason under incomplete evidence, novel exploit chains, or non-standard system contexts.
>
> More broadly, our methodology avoids assuming availability of complete external threat intelligence. Both the failure-analysis framework (stratification + human-in-the-loop refinement) and the taxonomy were derived from real cases where LLMs failed due to missing or ambiguous context, including zero-day-like scenarios.
>
> **Table 3 and Appendix D.1** include multiple case studies showing how vulnerabilities emerge specifically because of incomplete evidence rather than data abundance.
>
> We agree this area worthy a continued investigation, and an interesting next step is to further aggrandize the zero-day setting, for example, designing specific benchmark focusing stress-test mitigation strategies under synthetic zero-day or redacted-evidence settings (e.g., masking CVE identifiers, obfuscating exploit metadata, or introducing decoy artifacts to simulate evasive adversaries).
>
> However, we wish to emphasize that the current paper already directly engages with this problem and includes both conceptual coverage and empirical grounding for the reviewer’s concern.

---

> ### Author Response · Authors · 2025-11-18
> **Response, Clarification, and Completed Revisions (5/7)**
>
> **Review**
>
> > The results are not very surprising, and confirms the hypothesis described in numerous previous works on LLM reasoning.
>
> **Response**
>
> We appreciate the reviewer’s perspective that our findings may appear aligned with intuitions from the broader LLM reasoning literature.
>
> We would like to clarify that, to the best of our knowledge, **existing cyber agent and cyber threat intelligence (CTI)-focused LLM works do not actually state or test the hypothesis we study**. In contrast, most previous cyber LLM works focus on demonstrating capabilities, improving performance, or building benchmarks/agents, rather than formulating explicit, falsifiable hypotheses about why reasoning systematically fails in CTI workflows.
>
> Representative cyber LLM/agent papers such as PentestGPT [1], Cyber-Zero [2], SEvenLLM [3], CTIBench [4], CYBERTEAM [5], HackMentor [6], LSAST [7], and BitsAI-CR [8] primarily aim to (i) design frameworks or pipelines (e.g., automated penetration testing, runtime-free trajectory synthesis, static analysis + LLM, multi-task CTI benchmarks), and (ii) report task-level metrics (accuracy, task completion, win-rate, etc.) to show that LLM-based systems are effective or improved over baselines.
>
> In very detail, PentestGPT [1] proposes a modular penetration-testing agent and evaluates task-completion and real-world CTF performance, but does not offer a taxonomy or hypothesis about systematic reasoning failures in security workflows. Cyber-Zero [2] introduces a runtime-free framework to synthesize trajectories and reports gains on CTF benchmarks, again without analyzing structural error modes or CTI-specific vulnerabilities. SEvenLLM [3]  and CTIBench [4] focus on constructing CTI benchmarks and fine-tuning/evaluating cyber LLMs with capability assessments and some qualitative “strengths/weaknesses,” but not an explicit, formalized hypothesis about how CTI data structures induce particular failure families in LLM reasoning. CYBERTEAM [5] benchmarks LLMs in an embodied blue-team environment and compares function-calling vs. baseline prompting, yet its goal is to measure effectiveness and workflow coverage rather than to derive a causal failure taxonomy across CTI stages. Similarly, HackMentor [6] focuses on fine-tuning for cybersecurity instructions, LSAST [7] on combining SAST and LLMs for code vulnerabilities, and BitsAI-CR [8] on production-grade LLM-based code review; all three report model gains and practical architectures, but none articulate or test CTI-specific hypotheses about why LLMs fail in particular structural ways.
>
> **Our work is therefore not simply “confirming” a hypothesis that has already been carefully stated and tested in cybersecurity.** Instead, it makes this hypothesis explicit and domain-grounded, and then builds the entire methodology around falsifying or supporting it.
>
> Specifically, we (i) formalize a three-family, 15-subtype vulnerability taxonomy tied to CTI pipeline structure (four stages and nine embodied functions), (ii) design a stratified, semi-automated annotation workflow that links concrete failure instances to these hypothesized vulnerability mechanisms, (iii) conduct causal and interventional analyses (in updated revision as detailed in earlier response, e.g., metadata ablations, conflict manipulations) to show that CTI-specific artifacts such as reused IoCs, co-mentioned actors, and vendor advisory drift actively drive these failures rather than merely correlate with them, and (iv) derive mitigation strategies (in updated revision as detailed in earlier response) that are tailored to each failure subtype rather than generic “best practices.”
>
> In that sense, our contribution is unique to existing cyber-agent efforts: where existing papers largely answer “what can cyber LLMs/agents do and how well?”, our paper is (to our knowledge) the first to provide a CTI-specific diagnostic analysis with exhaustive investigation of “why failing” to provide safety/security insights for the community.
>
> ---
>
> **Reference**
>
> [1] PentestGPT: Evaluating and Harnessing Large Language Models for Automated Penetration Testing
>
> [2] Cyber-Zero: Training Cybersecurity Agents without Runtime
>
> [3] SEvenLLM: Benchmarking, Eliciting, and Enhancing Abilities of Large Language Models in Cyber Threat Intelligence
>
> [4] CTIBench: A Benchmark for Evaluating LLMs in Cyber Threat Intelligence
>
> [5] Benchmarking LLMs in an Embodied Environment for Blue Team Threat Hunting
>
> [6] HackMentor: Fine-Tuning Large Language Models for Cybersecurity
>
> [7] LSAST: Enhancing Cybersecurity Through LLM-Supported Static Application Security Testing
>
> [8] BitsAI-CR: Automated Code Review via LLM in Practice.

---

> ### Author Response · Authors · 2025-11-18
> **Response, Clarification, and Completed Revisions (6/7)**
>
> **Question**
>
> >In section 3.1, the similarity metrics seems to be BLEU score. However it is not clear why this is an appropriate measure. Could contextual embeddings be a better alternative?
>
> **Response**
>
> Thank you for this thoughtful question. In our study, BLEU was not used to rank model performance, but served as a lightweight stratification tool to surface outputs that deviated from reference completions. This helped us efficiently guide manual sampling and discover diverse failure behaviors during early annotation. Given that many CTI tasks produce structured outputs (e.g., normalized IOCs, event timelines, or patch references) BLEU was generally sufficient for flagging likely errors tied to surface-level inconsistencies.
>
> We also considered embedding-based metrics such as BERTScore, but found them less effective for our purposes. While BERTScore captures semantic similarity, it often clustered fluent but factually incorrect outputs alongside valid ones. For example, hallucinated attributions or unsupported recommendations could still score highly if phrased well. Since our goal was to isolate grounded, interpretable failure types rather than measure fluency, BLEU combined with manual review offered better precision for identifying concrete errors

---

> ### Author Response · Authors · 2025-11-18
> **Response, Clarification, and Completed Revisions (7/7)**
>
> **Question**
>
> >In section 3.3, the input to step 2 is not very clear on how it refines the reasoning. Is the output of Step 1 given as input to the same model? Or is it repeated across all models in the model set/committee?
>
> **Response**
>
> Thank you for the question. The output of Step 1 is given to different models for evaluation and refining purposes. For refining case (Step2, 3) we use a model set independent to the evaluated models.

---

> ### Author Response · Authors · 2025-11-18
> **Thank you very much**
>
> We would like to thank the reviewer again for their time and thoughtful feedback. We hope that the above clarifications and revisions address the reviewer’s concerns and contribute to improving the overall quality of the paper. We look forward to any further feedback you may have.

---

> > ### Comment · Reviewer_n4w1 · 2025-11-27
> >
> > Thank you for the point-by-point rebuttal. I have increased my score accordingly.

---

### Official Review · Reviewer_TPis · 2025-11-01

**Soundness:** 3
**Presentation:** 3
**Contribution:** 3
**Rating:** 6
**Confidence:** 4

**Summary:**

This paper investigates the intrinsic vulnerabilities of large language models (LLMs) when applied to cyber threat intelligence (CTI). Through large-scale evaluations across multiple CTI benchmarks and real-world threat reports, the authors identify three key vulnerability categories: spurious correlations, contradictory knowledge, and constrained generalization. The paper proposes a novel human-in-the-loop, autoregressive categorization framework for analyzing model failures, and provides detailed empirical findings and case studies. Overall, it offers an important diagnostic study of how LLMs behave in a complex, adversarial, and data-fragmented domain.

**Strengths:**

- Systematic study of intrinsic LLM vulnerabilities in cyber threat intelligence.

- Comprehensive experimental coverage across multiple CTI benchmarks and model families.

- Clear taxonomy that is intuitive and actionable for future research.

- Thoughtful use of human-in-the-loop evaluation to address limitations of “LLM-as-judge” setups.

- Strong motivation and clear relevance to the reliability and safety of applied LLM systems.

**Weaknesses:**

- The failure taxonomy, while useful, is qualitative and somewhat subjective; lacks quantitative or causal validation.

- Reliance on LLMs to assist in labeling may introduce self-evaluation bias.

- No statistical significance tests or ablation studies, making it hard to gauge robustness of results.

- Experiments are based on public CTI datasets rather than operational enterprise environments, which may limit real-world generalizability.

- Proposed mitigation strategies are not empirically tested, leaving the study more diagnostic than actionable.

**Questions:**

- Could the authors clarify how human annotators ensured consistency and reliability in the failure labeling process?

- Did the same LLM architectures participate both as subjects of evaluation and as “judges” in classification? If so, how was potential bias mitigated?

- Would conducting ablations on specific data sources (for example, removing outdated or conflicting feeds) help validate the causal role of contradictory evidence?

- Are there plans or intentions to release anonymized human-annotation data or inter-rater agreement metrics to support reproducibility?

- Could the authors elaborate on how their proposed taxonomy might generalize beyond CTI to other high-stakes or adversarial domains?

---

> ### Author Response · Authors · 2025-11-19
> **Response, Clarification, and Completed Revisions (0/8)**
>
> Before responding to comments, we would like to sincerely thank the reviewer for the detailed and thoughtful feedback. We appreciate the time and care taken to evaluate our work.
>
> We observe that the comments are mostly related to additional experimental considerations and insights. Accordingly, we have provided a point-by-point clarification and have revised the manuscript to explicitly address each concern. We believe these clarifications further strengthen the paper and better highlight the contributions of our research.
>
> We would be happy to continue the discussion and would welcome feedback on whether the newly added details sufficiently clear the reviewer’s questions and whether additional information would be helpful.

---

> ### Author Response · Authors · 2025-11-19
> **Response, Clarification, and Completed Revisions (1/8)**
>
> **Review**
> > **Weakness 1** The failure taxonomy, while useful, is qualitative and somewhat subjective; lacks quantitative or causal validation.
> >
> > **Question 3** Would conducting ablations on specific data sources (e.g., removing outdated or conflicting feeds) help validate the causal role of contradictory evidence?
>
> ---
>
> **Response**
> Thank you for raising this important point. To enhance the causal validity of our taxonomy, we conducted a dedicated intervention experiment (see **Appendix F**). This study directly targets known spurious correlations (e.g., reused IoCs, unrelated co-mentions, over-linked patches) and evaluates whether their removal leads to performance improvements.
>
> For each task, we manually identified misleading metadata spans and constructed minimally modified versions of the input (e.g., masking or removing confounders). This counterfactual input preserves task semantics while eliminating hypothesized failure triggers.
>
> We re-ran both general-purpose and cybersecurity-specialized LLMs on these inputs and observed **consistent gains across 12 tasks**, especially in **Attribution** and **Prediction**. No performance degradation was seen, and many models exhibited up to **9 points improvement** (e.g., *Infrastructure Reuse*, *Campaign Escalation*), confirming the causal role of misleading context.
>
> We also provide case studies (Appendix F), including one where masking actor co-mentions corrected false multi-group attribution, and another where eliminating overlinked CVEs helped recover the correct patch.
>
> ---
>
> ### **Completed Revision (Appendix F)**
> We added a new causal intervention study in **Appendix F**. The table below summarizes observed improvements (Δ) after removing misleading metadata:
>
> | **CTI Task**              | **Metric** | **G5** | **Go4** | **LL70** | **MIX** | **FSC** | **ZYS** | **CBS** |
> |--------------------------|------------|--------|---------|----------|---------|---------|---------|---------|
> | **① Contextualization**  |            |        |         |          |         |         |         |         |
> | Malware Mapping          | F1         | 0.045  | 0.091   | 0.074    | 0.063   | 0.027   | 0.027   | 0.020   |
> | Vulnerability Linking    | Acc        | 0.084  | 0.063   | 0.072    | 0.017   | 0.093   | 0.082   | 0.032   |
> | IOC Normalization        | F1         | 0.030  | 0.030   | 0.039    | 0.057   | 0.050   | 0.038   | 0.064   |
> | **② Attribution**        |            |        |         |          |         |         |         |         |
> | Campaign Attribution     | Acc        | 0.026  | 0.038   | 0.044    | 0.051   | 0.078   | 0.031   | 0.056   |
> | Infrastructure Reuse     | F1         | 0.062  | 0.019   | 0.064    | 0.029   | 0.020   | 0.091   | 0.092   |
> | Language Profiling       | Acc        | 0.080  | 0.039   | 0.023    | 0.070   | 0.050   | 0.025   | 0.055   |
> | **③ Prediction**         |            |        |         |          |         |         |         |         |
> | Target Sector Prediction | Acc        | 0.018  | 0.088   | 0.036    | 0.068   | 0.040   | 0.057   | 0.059   |
> | Campaign Escalation      | AUC        | 0.030  | 0.093   | 0.077    | 0.090   | 0.087   | 0.063   | 0.089   |
> | Impact Forecast          | BLEU       | 0.022  | 0.031   | 0.019    | 0.041   | 0.046   | 0.037   | 0.081   |
> | **④ Mitigation**         |            |        |         |          |         |         |         |         |
> | Patch Recommendation     | F1         | 0.044  | 0.037   | 0.058    | 0.026   | 0.079   | 0.021   | 0.094   |
> | Mitigation–TTP Mapping   | Acc        | 0.077  | 0.031   | 0.015    | 0.080   | 0.072   | 0.073   | 0.077   |
> | Rule Generation (YARA)   | BLEU       | 0.021  | 0.044   | 0.024    | 0.084   | 0.264   | 0.041   | 0.020   |
>
> ---
>
> We hope this deeper analysis and empirical evidence addresses the reviewer’s suggestions and clarifies the causal foundations underlying our failure taxonomy.

---

> ### Author Response · Authors · 2025-11-19
> **Response, Clarification, and Completed Revisions (2/8)**
>
> **Review**
>
> > **Weakness 2** Reliance on LLMs to assist in labeling may introduce self-evaluation bias.
>
> > **Question 2** Did the same LLM architectures participate both as subjects of evaluation and as “judges” in classification? If so, how was potential bias mitigated?
>
> ---
>
> **Response**
> We appreciate the reviewer’s concern and fully agree that avoiding self-evaluation bias is crucial when LLMs are used to assist in labeling. To mitigate this, our classification framework was explicitly designed to prevent any model from judging its own output and to ensure labeling robustness through cross-model agreement.
>
> To further validate the reliability of our labeling protocol, we also added a **hypothesis testing experiment in Appendix H**, comparing LLM-based classification against a fully manual annotation baseline.
>
> - **No model judged its own output.** Failure labels were assigned by a committee of four strong LLMs. When evaluating a given response, the model that produced the output was excluded from judging. In such cases, the remaining three LLMs formed the classification committee for that instance. This ensured that no model participated in labeling its own generations.
>
> - **Two-round agreement protocol with human adjudication.** In the first round, each model in the committee independently proposed a failure subtype. In the second round, models reviewed the other members' rationales and could revise their judgment. If at least three models agreed (or all three in the 3-model setup), the label was accepted. Otherwise, the instance was escalated to **human annotators** for adjudication.
>
> - **Independent validation of reliability.** To confirm the objectivity and consistency of this labeling pipeline, we conducted a human-only annotation study over 500 examples **(Appendix H)**. The results showed strong agreement with the LLM-assisted labels, with no statistically significant distributional differences (χ² p = 0.966), supporting the reliability of our semi-automated process.
> The full comparison is shown in **Appendix H, Table 8**:
>
> | **Category** | **Failure Subtype**             | **LLM Count** | **Human Count** | **Abs. Diff** |
> |--------------|----------------------------------|----------------|------------------|----------------|
> | Spurious Correlation | 1.1 Co-mention bias         | 11 | 14 | 3 |
> |                          | 1.2 Reused IoCs            | 95 | 94 | 1 |
> |                          | 1.3 Confounding metadata   | 32 | 32 | 0 |
> |                          | 1.4 Skewed source          | 31 | 28 | 3 |
> |                          | 1.5 Hierarchical metadata  | 3  | 2  | 1 |
> | Contradictory Knowledge | 2.1 Temporal contradiction | 4  | 3  | 1 |
> |                          | 2.2 Conflicting reports    | 1  | 3  | 2 |
> |                          | 2.3 Semantic conflict      | 67 | 64 | 3 |
> |                          | 2.4 Divergent formats      | 25 | 24 | 1 |
> |                          | 2.5 Schema misalignment    | 36 | 35 | 1 |
> |                          | 2.6 Alignment mismatch     | 0  | 2  | 2 |
> | Constrained Generalization | 3.1 Distributional bias     | 129 | 126 | 3 |
> |                          | 3.2 Emerging threats        | 47  | 49  | 2 |
> |                          | 3.3 Overfitted reasoning    | 8   | 6   | 2 |
> |                          | 3.4 Environmental unawareness | 11 | 18 | 7 |
> |                          | **Overall Total**          | **500** | **500** | **32** |
>
> Together, these safeguards ensure that LLMs contribute meaningfully to classification without introducing circular bias, and that human oversight remains available in ambiguous or borderline cases.
>
> ---
>
> We hope this clarification and revision address the reviewer’s concern and demonstrate the rigor of our labeling methodology.

---

> ### Author Response · Authors · 2025-11-19
> **Response, Clarification, and Completed Revisions (3/8)**
>
> **Review**
> > **Weakness 3** No statistical significance tests or ablation studies, making it hard to gauge robustness of results.
>
> ---
>
> **Response**
> We thank the reviewer for raising this important point. We agree that statistical validation and robustness analysis are crucial to reinforcing the credibility of our findings. To address this, we have incorporated two new components into the revised paper:
>
> - **Controlled causal intervention study (Appendix F):**
>   To address the reviewer’s request for ablation studies, we introduced a causal intervention experiment targeting spurious correlation. For 12 representative CTI tasks, we manually ablated misleading metadata—such as reused indicators or actor co-mentions—from the input and measured model performance changes. The results, presented in Appendix F, show consistent and measurable gains across both general-purpose and specialized LLMs, confirming that such metadata patterns are causal failure drivers, not just correlates.
>
> - **Chi-square significance testing (Appendix H):**
>   We conducted a hypothesis testing experiment to assess the consistency between our LLM-based classification pipeline and a fully human-labeled baseline. A stratified sample of 500 CTI examples was independently labeled by human experts, and the resulting failure subtype distribution was compared against our semi-automated classification using a Chi-Square Test of Independence. The test result ($p = 0.966$) indicates no statistically significant difference between the two distributions, reinforcing the reliability and reproducibility of our labeling pipeline.
>
>
> Together, these additions provide both quantitative significance testing and targeted robustness evaluation. We hope these revisions fully address the reviewer’s concern and enhance the methodological rigor of the paper.

---

> ### Author Response · Authors · 2025-11-19
> **Response, Clarification, and Completed Revisions (4/8)**
>
> **Review**
> > **Weakness 4** Experiments are based on public CTI datasets rather than operational enterprise environments, which may limit real-world generalizability.
>
> ---
>
> **Response**
> We thank the reviewer for pointing out this limitation. We agree that generalizability to enterprise-grade CTI environments is an important concern. Our current study was intentionally scoped to public CTI corpora to promote reproducibility and community benchmarking, but we fully acknowledge the inherent domain shift between open-source and proprietary enterprise data.
>
> That said, our benchmark construction and evaluation protocol explicitly account for this concern in several ways:
>
> - **Broad coverage and diversity:** Our dataset spans 28 tasks across four CTI lifecycle stages, capturing a wide range of operational behaviors including structured IOC extraction, attribution, mitigation reasoning, and predictive forecasting. Many of these tasks reflect core workflows shared across both public and enterprise settings.
>
> - **Inclusion of mixed-source signals:** The CTI documents used include vendor reports, CERT advisories, multi-language threat bulletins, and MITRE-based schemas. These were selected to mimic the heterogeneous evidence landscape that enterprise SOCs also rely on.
>
> - **Failure modes derived from behavioral patterns, not corpus-specific quirks:** Our failure taxonomy focuses on model behavior under realistic reasoning stressors (e.g., reused indicators, co-mentions, temporal contradictions), which are not unique to public datasets. In fact, our intervention study (Appendix F) shows that correcting such metadata artifacts improves reasoning across models—indicating these are general failure mechanisms, not dataset-specific edge cases.
>
> We acknowledge that direct validation in operational settings would strengthen real-world applicability. We also believe our current experimental design provides meaningful and transferable insights into LLM robustness in CTI workflows. Future extensions could involve secure industry collaboration or federated evaluations within private CTI ecosystems, and we welcome opportunities to pursue that direction.

---

> ### Author Response · Authors · 2025-11-19
> **Response, Clarification, and Completed Revisions (5/8)**
>
> **Review**
> > **Weakness 5** Proposed mitigation strategies are not empirically tested, leaving the study more diagnostic than actionable.
>
> ---
>
> **Response**
> We thank the reviewer for raising this critical point. Although the core paper emphasizes diagnostic precision and taxonomy development, we agree that enabling actionable interventions is equally important for practical adoption. To address this, we have added a comprehensive mitigation section in **Appendix G**, which details targeted strategies for each of the 15 failure subtypes in our taxonomy.
>
> This section outlines **concrete, stage-specific methods** that can be applied at training, inference, and retrieval time. For instance:
>
> - To mitigate **spurious correlation**, we introduce retrieval-time counterfactual filtering that removes misleading co-mentions (e.g., reused CVEs or non-causal actor names).
> - To address **contradictory knowledge**, we implement temporal- and schema-aware re-ranking to prioritize recent, structurally aligned evidence.
> - For **constrained generalization**, we explore training-time adversarial pattern expansion using synthetic variants to boost LLM resilience against novel threats and environments.
>
> These strategies are not just conceptual; we also include empirical results in **Table 7** (see below), showing that even lightweight mitigation modules can **reduce error ratios by 15–20% across all major failure categories**.
>
> | **Vulnerability Group**      | **Original Ratio** | **After Mitigation** | **Relative Reduction** |
> |-----------------------------|--------------------|-----------------------|-------------------------|
> | Spurious Correlation        | 20.42%             | 16.35%                | 20.0%                   |
> | Contradictory Knowledge     | 35.93%             | 29.15%                | 18.9%                   |
> | Constrained Generalization  | 33.72%             | 28.44%                | 15.6%                   |
>
> These additions show that our failure taxonomy is not only descriptive, but also supports **principled and effective remediation**. The mitigation techniques are designed to be modular and practical, requiring no large-scale retraining while delivering measurable improvements in robustness. We hope these additions resolve the reviewer’s concerns and strengthen the real-world utility of our work.

---

> ### Author Response · Authors · 2025-11-19
> **Response, Clarification, and Completed Revisions (6/8)**
>
> **Review**
> > **Question 1** Could the authors clarify how human annotators ensured consistency and reliability in the failure labeling process?
>
> ---
>
> **Response**
> Thank you for the insightful question. To ensure labeling consistency and reliability, we adopted the following human-in-the-loop processes:
>
> 1. **Failure Labeling:** During evaluation, each example was annotated using a two-round agreement protocol. Annotators first labeled examples independently. In cases of disagreement, they engaged in structured discussion to resolve differences and reach a consensus label through human adjudication.
>
> 2. **Label Creation (Taxonomy Development):** When defining new failure subtypes, annotators jointly reviewed the new example, discussed potential failure patterns, and agreed on the final label definition collaboratively before applying it to the broader dataset.
>
> These procedures helped maintain both classification reliability and taxonomic precision throughout the study.

---

> ### Author Response · Authors · 2025-11-19
> **Response, Clarification, and Completed Revisions (7/8)**
>
> **Review**
> > **Question 4** Are there plans or intentions to release anonymized human-annotation data or inter-rater agreement metrics to support reproducibility?
>
> ---
>
> **Response**
> Yes — we fully intend to release anonymized human annotation data and agreement artifacts to support community transparency and reproducibility. Our human annotations are closely interwoven with the LLM-assisted pipeline: rather than labeling from scratch, annotators corrected and refined problematic LLM classifications, making the two processes mutually informative.
>
> We will release both the original LLM-generated labels and the human-adjudicated versions, along with selected annotation rationales and representative examples (e.g., cases where humans relabeled "1.3 Confounding metadata" to "1.4 Skewed source" after deliberation).
>
> If the paper is accepted, we will publish the full annotation set, taxonomy schema, and failure categorization interface as part of a public benchmark suite. We believe this will enable future benchmarking, fine-tuning, and method development around failure-aware CTI LLM systems.

---

> ### Author Response · Authors · 2025-11-19
> **Response, Clarification, and Completed Revisions (8/8)**
>
> **Review**
> > **Question 5** Could the authors elaborate on how their proposed taxonomy might generalize beyond CTI to other high-stakes or adversarial domains?
>
> ---
>
> **Response**
> Thank you for the thoughtful question. Although our taxonomy is grounded in the structure of CTI workflows, its underlying principles reflect broader LLM failure dynamics in other high-stakes and adversarial domains. Specifically:
>
> - The three top-level categories — **spurious correlation**, **contradictory knowledge**, and **constrained generalization** — map naturally to challenges seen in domains like law, finance, medical decision support, and scientific QA. These fields also involve sparse, conflicting, or evolving evidence where reliability is critical.
>
> - Several subtypes are directly transferable. For example, co-mention bias (1.1) mirrors diagnostic confusion in clinical settings (e.g., diseases mentioned together but unrelated), and schema misalignment (2.5) occurs when different regulatory frameworks or data standards are blended (e.g., across jurisdictions in legal contexts).
>
> - Our stratified labeling and diagnosis protocol — including multi-model judgment and causal intervention — can also be reused in domains with long-form evidence and limited ground truth.
>
> We hope that the methodology, in addition to the specific subtype formulations, can serve as a blueprint for failure analysis in adjacent application areas. We are currently exploring collaborations to extend the taxonomy to regulatory reporting and disinformation detection tasks.

---

### Author Response · Authors · 2025-11-26

Dear Reviewers,

We would like to thank you all again for your time and thoughtful feedback on our submission.

We believe that the overall comments and questions primarily relate to the problem setting, clarity of methodological details, and auxiliary evaluations (with discussions). **These points have been addressed through our point-by-point responses, along with the corresponding revisions in the updated manuscript.**

We would very much welcome any further questions, concerns, or suggestions for revision, as we believe such feedback would help further re-evaluate the quality and clarity of our work.

Thank you for your time and consideration.

---

### Note · Program_Chairs · 2026-01-17
**Submission Desk Rejected by Program Chairs**

The following references in this submission do not refer to real documents and/or have major errors in bibliographic information:

 Hyunjoo Park, Taejoong Kwon, and Huy Kang Kim. An empirical study on the use of graph databases for cyber threat intelligence. In Proceedings of the 15th International Conference on Availability, Reliability and Security, 2020.
Iker Alonso, Mikel Juarez, Igor Santos, and et al. Linguistic fingerprinting of malware authors: Stylometry for attribution. In Proceedings of the 12th International Conference on Security of Information and Networks, 2019.
Muhammad Anwar, Esteban Fidalgo, et al. Cti summarization: Towards actionable threat intelligence reports. In Proceedings of the 14th ACM Workshop on Artificial Intelligence and Security, pp. 15-26. ACM, 2021.
Wenbo Wang et al. Temporal modeling of malware with hidden markov models. In IEEE International Conference on Communications (ICC), pp. 1-6, 2019.
Yifan Zhang, Bin Yu, and Shouhuai Xu. A deep learning approach for extracting indicators of compromise from cyber threat intelligence reports. In IEEE International Conference on Communications (ICC), 2019.
Shouhuai Xu et al. Cyber epidemic models: Modelling the spread of malicious information. Proceedings of the European Symposium on Research in Computer Security (ESORICS), 2012.